**Decadal changes in atmospheric ammonia and dry deposition across China inferred from space-ground measurements and model simulations**

Fan Sun[1], Yu Cui[1,2], Jiayin Su[3], Mark W. Shephard[4], Shailesh K. Kharol[4,5], Yifan Zhang[1,2],

Xuejing Shi[1,2], Junqing Zhang[1,2], Huili Liu[1,2], Qitao Xiao[6], Xiao Lu[3], Zhao-Cheng Zeng[7],

Timothy J. Griffis[8], Cheng Hu[1*]

[1] College of Ecology and Environment, Joint Center for sustainable Forestry in Southern China, Nanjing Forestry University, Nanjing 210037, China

[2] Yale-NUIST Center on Atmospheric Environment, Collaborative Innovation Center on Forecast and Evaluation of Meteorological Disasters (CIC-FEMD), Nanjing University of Information Science & Technology, Nanjing, 210044, China

[3] School of Atmospheric Sciences, Sun Yat-sen University, Zhuhai, 519082, China.

[4] Environment and Climate Change Canada (ECCC), Toronto, Ontario, M3H 5T4, Canada

[5] AtmoAnalytics Inc., Brampton, Ontario, L6S 6L2, Canada

[6] Key Laboratory of Lake and Watershed Science for Water Security, Nanjing Institute of Geography and Limnology, Chinese Academy of Sciences

[7] School of Earth and Space Sciences, Peking University, Beijing 100871, China

[8] Department of Soil, Water, and Climate, University of Minnesota-Twin Cities, Minneapolis, MN, USA

*Corresponding author: Cheng Hu (chenghu@njfu.edu.cn or nihaohucheng@163.com)

Submitted to: *Atmospheric Chemistry and Physics*

**Abstract:** Ammonia ($NH_3$), a key alkaline gas in the atmosphere, significantly influences ecosystem nitrogen cycling and the formation of fine particulate matter ($PM_{2.5}$). However, limited ground-based monitoring hinders understanding of $NH_3$'s spatial and temporal dynamics and its dry deposition across China, which is ranked as one of the largest global $NH_3$ emission hotspots. This study integrated 2013-2023 satellite-derived $NH_3$ column concentrations from the Cross-track Infrared Sounder (CrIS) with adjustments from approximately five years ground in-situ ground observations to derive spatial-temporal variation in ground-level $NH_3$ concentrations across China. We also used the GEOS-Chem transport model and a random forest algorithm by using emission inventories and reanalysis meteorological fields to simulate $NH_3$ dry deposition velocity and fluxes, and explore the mechanisms driving observed trends. The CrIS observations results show that column-averaged (averages from ground to ~1 km) $NH_3$ concentrations were the highest in the North China Plain (>10 ppb), with notable annual and seasonal increasing trends. $NH_3$ concentrations in 2023 were 13.8%-30.6% higher than in 2013. CrIS retrievals aligned well with in-situ data, though were generally about twice as high. After applying the regression equation between ground in-situ observations and CrIS column-averaged $NH_3$ concentrations, we derive the spatial-temporal ground-level (1~1.5 m) $NH_3$ concentrations and dry deposition fluxes from 2013 to 2023. The $NH_3$ dry deposition fluxes exhibited a clear east-west gradient, with maxima in the North China Plain, and another hotpot region is also observed in the Sichuan Basin, southwestern China. Increases in ground-level $NH_3$ concentrations and deposition were most pronounced in urban, cropland, and forest regions, with urban areas experiencing the fastest growth and grasslands the highest total deposition. The national mean ground-level $NH_3$ concentration and dry deposition flux were 4.98 ppb and 0.51 g $NH_3$ $m^{-2}$ $yr^{-1}$, respectively. Anthropogenic emissions explained 77.4% of the variability in ground-level $NH_3$ concentration trend, and meteorological factors accounted for the remainder. Besides, 72.6%-81.2% of the $NH_3$ dry deposition trend was governed by $NH_3$ concentration changes. This study identifies the underlying cause of increasing ammonia pollution, which can be used to better inform nitrogen management strategies in China.

Keywords: $NH_3$ concentration, dry deposition, satellite-based observation, random forest model

**1 Introduction**

Ammonia ($NH_3$), as the most abundant alkaline gas in the atmosphere, readily reacts with acidic species such as nitric acid and sulfuric acid to form secondary inorganic aerosols. These aerosols contribute significantly to fine particulate matter ($PM_{2.5}$), thereby adversely affecting human health, air quality, and atmospheric visibility (Na et al., 2007; Hauglustaine et al., 2014; He et al., 2001). Reducing $NH_3$ emissions has been identified as a cost-effective strategy for mitigating air pollution (Pinder et al., 2007; Wu et al., 2016). In addition, excessive atmospheric $NH_3$ can also deposit onto terrestrial and aquatic ecosystems through dry and wet processes, leading to soil acidification, eutrophication, and biodiversity loss (Hernández et al., 2016; Fu et al., 2017; Hu et al., 2021). Therefore, monitoring and quantifying atmospheric $NH_3$ concentrations and deposition rates within different land cover types, especially at global emission hotspots, are critical for informing nitrogen management strategies and protecting air, soil, and water resources, as well as human health (Liu et al., 2017a; Griffis et al., 2019).

As the world's largest agricultural country in terms of total crop yield, China is also among the top $NH_3$ emitters globally. In 2018, the global $NH_3$ emissions from rice, wheat and corn fields were $4.3 \pm 1.0$ Tg N $yr^{-1}$, of which China's emissions per unit area were as high as 19.7 kg N $ha^{-1}$ $yr^{-1}$, which was much higher than that of the United States (9.1 kg N $ha^{-1}$ $yr^{-1}$) and India (10.8 kg N $ha^{-1}$ $yr^{-1}$) (Zhan et al., 2021; Luo et al., 2022). From global inventories such as EDGAR and CEDS, China's $NH_3$ emissions accounted for 19.8% of the global total in 2013. In 2022, this proportion had declined to about 14.5% (Crippa et al., 2024). In recent years, the proportion of $NH_3$ deposition to total nitrogen (N) deposition has increased steadily, accounting for approximately 67.0% in China in 2020 (Liu et al., 2024c). This upward trend is expected to continue, driven by declining $NO_x$ and $SO_2$ emissions due to pollution control policies and rising $NH_3$ emissions associated with global agricultural intensification (Erisman et al., 2008; Goldberg et al., 2021; Pinder et al., 2008).

$NH_3$ deposition in China is nearly double that of the EU (Liu et al., 2024c), mainly due to excessive nitrogen fertilizer application. In 2014, agricultural $NH_3$ volatilization accounted for

12 Tg N yr$^{-1}$ globally, with China contributing about 34% (Ma et al., 2020). Anthropogenic
activities have nearly doubled NH$_3$ emission over the past few decades, with cropland and
livestock sources making up around 80% of the global total emissions. Non-agricultural
sources—such as wildfire biomass burning, wastewater treatment, human excreta, and
transportation—remain relatively minor (Behera et al., 2013; Zhu et al., 2015; Van Damme et
al., 2018; Lutsch et al., 2019). Although the growth rate of both agricultural and non-agricultural
NH$_3$ emissions in China has slowed in recent years, the absolute emissions continue to rise
(Chen J et al., 2023).

Atmospheric NH$_3$ concentration serves as a key indicator of emission intensity due to its
relatively short atmospheric lifetimes, typically the order of hours in the atmospheric boundary
layer (hereafter ABL) (Evangeliou et al., 2021). Therefore, accurately quantifying its
spatiotemporal variations and identifying the underlying drivers is essential for constraining
NH$_3$ emission estimates, evaluating the ecological and environmental impacts and informing
effective mitigation strategies. Due to its high reactivity and predominant agricultural sources,
NH$_3$ exhibits pronounced temporal and spatial variability. To date, China operates two national
observation networks dedicated to monitoring NH$_3$ concentrations and deposition: The National
Nitrogen Deposition Monitoring Network (NNDMN, established in 2004) and the Ammonia
Monitoring Network of China (AMoN-China, established in 2015). While these networks
provide high-quality measurements, their sparse spatial coverage limits their ability to
characterize regional patterns for China (Liu et al., 2017a; b). Additionally, few sites offer long-
term (>10 years) continuous data records (Wang et al., 2023), posing challenges for trend
analysis across China. The limited availability of NH$_3$ monitoring data impedes our
understanding of its spatial-temporal patterns and impacts on air quality, climate, and
ecosystems.

In addition to surface monitoring, the chemical transport models (CTMs, i.e. GEOS-Chem,
WRF-Chem) are widely used to simulate NH$_3$ concentrations and dry deposition, as they
incorporate processes such as emission, transport, deposition, and chemical transformation (Hu
et al., 2020; 2021; Lu et al., 2020). However, their accuracy is constrained by uncertainties in
emission inventories and model parameterizations (e.g. bi-directional flux), where the bias in
both $NH_3$ emissions and other species (e.g. $NO_x$ and $SO_2$) can lead to considerable uncertainty
in simulating $NH_3$ concentration and corresponding deposition to ground (Kharol et al., 2018;
Van Der Graaf et al., 2022; Liu et al., 2024d). $NH_3$ emission estimates remain highly uncertain
due to outdated activity data, poorly constrained emission factors, and underrepresented sources
such as cities (Chang et al., 2021). Compared to most other air pollutants, $NH_3$ exhibits greater
variability and uncertainty in different inventories and models, particularly because of its
diverse agricultural sources and large influence from meteorological factors and human
activities (Beusen et al., 2008; Behera et al., 2013).

Recent advances in satellite remote sensing offer new opportunities to monitor boundary layer
atmospheric $NH_3$, which was first demonstrated by Beer et. al., (2008) with NASA's
Tropospheric Emission Spectrometer (TES) observations. The first global $NH_3$ distribution map
was derived in 2009 using data from the Infrared Atmospheric Sounding Interferometer (IASI)
onboard the MetOp-A satellite (Clarisse et al., 2009). Since then, other hyperspectral infrared
instruments have been used to map $NH_3$ concentrations over large regions, such as NASA TES
sensor, NASA/NOAA Cross-track Infrared Sounder (CrIS), the NASA Atmospheric Infrared
Sounder (AIRS), JAXA Greenhouse Gases Observing Satellite (GOSAT), and the
Geostationary Interferometric Infrared Sounder (GIIRS) on board China's FengYun-4B satellite
(Shephard et al., 2011; Shephard et al., 2015; Someya et al., 2020; Chen J et al. 2023; Zeng et
al., 2023). Satellite observations provide wide spatial coverage and continuous temporal
resolution, helping to fill spatial-temporal observation gaps by ground networks. Satellite-
derived $NH_3$ retrievals contain approximately 1 independent piece of information driven by
peak sensitivity (averaging kernel) in the ABL (~1-3 km) (Shephard et al., 2011; Shephard et
al., 2020) that can be represented as profiles with limited vertical resolution or integrated
column-averaged values. Therefore, column-averaged satellite retrievals cannot directly
replace ground-level (1~1.5 m) concentrations but provide complementary information that
helps fill in monitoring gaps.
Despite these limitations, satellite observations have been increasingly used to constrain $NH_3$
emissions, assess deposition flux, and identify trends (Chen et al., 2021; Kharol et al., 2018;
Van Damme et al., 2021). For instance, Liu et al. (2019a) estimated global surface $NH_3$
concentrations from IASI data and identified high concentrations (>6 μg N m$^{-3}$) in the North
China Plain and northern India. Linear trend analysis from 2008 to 2016 revealed strong
increases in eastern China (>0.2 μg N m$^{-3}$ yr$^{-1}$). More recently, satellite data have been used to
investigate urban $NH_3$ concentrations globally, showing a significant rise (1.2% yr$^{-1}$) in 2008-
2019 (Liu et al., 2024d). These studies demonstrate the utility of satellite retrievals in
characterizing $NH_3$ pollution and its spatiotemporal evolution, especially in regions lacking
surface monitoring. In addition to these near surface ammonia concentration observations (from
either in-situ surface or satellite observations), the dry deposition estimations also depend on
deposition velocities (Lei et al., 2021; Liu et al., 2024d). Therefore, an alternative and reliable
approach is to combine model simulated dry deposition, ground-level $NH_3$ concentration from
sites and satellite-based column-averaged observations, which can make full use of
corresponding advantages and eliminate the large uncertainty from emission inventories of
different pollution species.

Therefore, accurate estimation of $NH_3$ dry deposition and its driving factors are becoming
increasingly critical. Kharol et al. (2018) reported that $NH_3$ contributed more than $NO_2$ to dry
N fluxes over much of North America in the warm season. Liu et al. (2019a) used satellite-
derived data to estimate global $NH_3$ dry deposition during 2008-2016, with results broadly
consistent with ground measurements, highlighting the potential for satellite-based $NH_3$
observations to fill spatial-temporal gaps in $NH_3$ deposition assessment. In China, satellite
observations indicate that elevated $NH_3$ concentrations are predominantly observed in the North
China Plain, Northeast China, and the Sichuan Basin, whereas lower concentrations are found
on the Tibetan Plateau (Liu et al., 2017b). Despite the prominent $NH_3$ pollution identified in
several regions of China, there remains a lack of comprehensive long-term studies that examine
the spatiotemporal variations of $NH_3$ concentrations and dry deposition. The key drivers behind
these variations—impacted by rapid urbanization, land-use changes, climate change, and shifts

in fertilizer application practices—have not been sufficiently quantified. While observational studies conducted over a ten-year period cannot fully address the data gap, they offer valuable insights into the medium- and long-term trends in $NH_3$ concentrations and deposition patterns. To robustly constrain and quantify the spatiotemporal variations in column-averaged near surface level (average from ground to ~1 km), ground-level (1~1.5 m) $NH_3$ concentrations and dry deposition over the past decade, we integrated multiple data sources and analytical approaches. These included high-resolution satellite-derived $NH_3$ retrievals from 2013 to 2023, ground-based observational datasets, simulations from the GEOS-Chem chemical transport model, and dry deposition velocity estimates derived using a random forest algorithm. This study aims to address the following key scientific questions: (1) What are the spatial and temporal patterns of near surface level and ground-level $NH_3$ concentrations across different land cover types in China over the past decade from 2013 to 2023? (2) What are the temporal trends in $NH_3$ dry deposition across China during this period, and what are the primary driving factors? (3) What are the $NH_3$ concentrations and dry deposition fluxes in China compared to those in other regions globally? By addressing these questions, this study seeks to advance understanding of the nitrogen cycle in China and provide a scientific foundation for evaluating ecological impacts and informing targeted strategies for nitrogen management and sustainable agriculture.

## 2 Materials and Methods

### 2.1 Satellite-based atmospheric $NH_3$ concentration

The CrIS (version 1.6.4) satellite-based atmospheric $NH_3$ concentration used in this study. The CrIS is a hyperspectral infrared sounder onboard the Suomi National Polar-orbiting Partnership (Suomi NPP), NOAA-20, and NOAA-21 satellites (Shephard et al., 2020). Operating in a sun-synchronous orbit at an altitude of approximately 824 km, CrIS provides global coverage twice daily, with local overpass time around 13:30 (daytime) and 01:30 (nighttime). The instrument has a swath width of up to 2200 km, with a nadir spatial resolution of approximately 14 km, and excellent signal-to-noise ratio (Zavyalov et al., 2013). The CrIS fast physical retrieval (CFPR) algorithm (Shephard and Cady-Pereira, 2015) produces $NH_3$ retrievals using CrIS

onboard Suomi NPP from May 2012 to May 2021, and CrIS onboard NOAA-20 since March

8, 2019.

In this study, the near surface level of CrIS-derived atmospheric $NH_3$ retrieved profile

concentrations was utilized, which are strongly correlated with ABL values around 900 hPa (~1

km) and can represent column average $NH_3$ concentration from ground to ~1 km. To avoid

misunderstanding, we define near surface level in this study as the lowest level of CrIS-derived

$NH_3$ retrieved profile (average from ground to ~1km), and the ground-level as height of 1~1.5

m, which is the typical height of site-based observations. As this study focuses on China, we

used $NH_3$ data over regions of 73°-136°E and 3°-54°N and extracted $NH_3$ concentration within

China. To ensure data reliability, only high-quality retrievals were included, filtered using a

Quality Flag (QF) $\geq$ 3 and Cloud_Flag = 0. Non-detects (Cloud_Flag = 3) that account for

values below the detection limit of the sensor were not included in this study (White et al., 2023;

Shephard et al., 2025), but are not expected to have a significant impact in source regions found

in China. The analysis period spans from 2013 to 2023, covering both the SNPP and NOAA-

20 satellite missions, and provides an 11-year, near-continuous time series of atmospheric $NH_3$

observations over China. To assess the consistency between the two satellite missions, a

regression analysis was performed using monthly averaged $NH_3$ concentrations from the

overlapping period (2019-2021), revealing strong agreement and consistency across China

(Figure S1, *SI*). For subsequent analyses, the original satellite retrievals were resampled to a

uniform spatial resolution of 0.1° × 0.1°.

**2.2 Ground-based observations of atmospheric $NH_3$ concentration**

The dry deposition of $NH_3$ is the product of ground-level (usually calculated by site-based

observations of 1~1.5 m height) $NH_3$ concentration and modeled dry deposition velocity. Our

previous observation and modeling study in the U.S. Corn Belt found significant vertical

gradients within ABL height (~1-2 km) in years of 2017-2019 (Griffis et al., 2019; Hu et al.,

2020; 2021). Therefore, the coarse vertical resolution regional satellite mixing ratio values in

the lower boundary $NH_3$ concentration should be converted to better represent local ground

level values at 1~1.5 m, which will further be used to derive $NH_3$ dry deposition flux. To validate and adjust the regional satellite-derived $NH_3$ concentrations to better represent surface level sampling observations, we used measurements from the National Nitrogen Deposition Monitoring Network (NNDMN), which was established since 2010 and comprises 43 monitoring sites across China, encompassing different land cover types especially for urban, rural (cropland), and background (coastal, forest, and grassland) regions. The network provides high-quality observations of atmospheric reactive nitrogen (Nr) species in gas, particulate, and precipitation phases, including measurements of both wet and dry nitrogen deposition by using simulated dry deposition velocities (Xu et al., 2015).

NNDMN employs two monitoring methods: the long-term active denuder for long-term atmospheric sampling (DELTA) and the low-cost, passive Active Leading Passive High Absorption (ALPHA) sampler (Flechard et al., 2011). Monthly surface $NH_3$ concentrations are primarily monitored using DELTA, with a few sites utilizing ALPHA. Xu et al. (2015) demonstrated that these two methods yield statistically consistent $NH_3$ measurements. The observation periods for most sites range from 2010 to 2015, with detailed site information, including site names, locations, land cover types, and observation periods, provided in Table S1 (*SI*). Given that the satellite data selected for this study spans from 2013 to 2023, the analysis is limited to the period corresponding to the satellite data coverage. For sites where the observation period does not overlap with the satellite research period, and considering the typically low $NH_3$ concentrations at background sites, this study selected 24 representative urban and rural stations for adjustment to improve the reliability of subsequent $NH_3$ dry deposition estimates. The locations of monitoring sites and land cover types across China are also shown in Figure. 1a.

As noted above, the calculation of $NH_3$ dry deposition flux depends on ground-level $NH_3$ concentrations, although tens of site-based $NH_3$ concentration observations are available, they cannot provide long term spatial-temporal resolved $NH_3$ distributions especially in regions with high spatial heterogeneity within China. Therefore, we combined the advantage of ground-

based NH$_3$ observations of which can represent heights of 1~1.5 m, and satellite based spatial-
temporal NH$_3$ distributions. A linear relationship was constructed by comparing both datasets
at the same location and period (Hu et al., 2017; Liu et al., 2024b), where the regression
equation was used to adjust the lower boundary layer satellite mixing ratio observations to
ground-level of 1~1.5 m.

**2.3 Estimation of NH$_3$ dry deposition**
Dry deposition flux of atmospheric NH$_3$ was estimated by multiplying the observed ground-
level NH$_3$ concentration with the modeled dry deposition velocity, following the equation:

$$F = C \times V_\mathrm{d} \qquad (1)$$

Here, $F$ denotes the dry deposition flux, $C$ is the ground-level NH$_3$ concentration (ppb) obtained
from satellite retrievals and subsequently adjusted using ground-based measurements, and $V_d$
is the dry deposition velocity (cm s$^{-1}$), which is highly variable in space and time due to its
sensitivity to land surface characteristics and meteorological conditions.

The most widely used approach to derive $V_d$ is by model simulation. Here we first used the
GEOS-Chem chemical transport model to simulate spatial-temporal varied $V_d$ across China in
2015, with spatial resolution of 0.5° × 0.625° at hourly scale. However, considering (1) the
spatial resolution of 0.5° × 0.625° will lead to aggregation errors when quantifying NH$_3$
concentration and dry deposition from different land cover types within the same grid cell, and
(2) the GEOS-Chem model requires substantial computational resources for one decade, and to
further improve spatial resolution and computational efficiency (Figure S2, *SI*), a random forest
machine learning algorithm was also applied to simulate dry deposition velocities from 2013 to
2023 based on output from GEOS-Chem model (see more details in Section 2.4), where the
spatial resolution can improve to 0.25°, see more details in Section 2.4.

**2.4 Simulation of NH$_3$ dry deposition velocity (V$_d$)**
**2.4.1 Simulation of V$_d$ by using GEOS-Chem model**
We applied a hybrid modeling approach that combines the GEOS-Chem model with a random

forest regression algorithm to estimate $NH_3$ dry deposition velocities across China. GEOS-Chem is a global 3-D chemical transport model driven by meteorological inputs from NASA's Goddard Earth Observing System (GEOS), developed for simulating atmospheric composition and chemistry (Eastham et al., 2014). In this study, we used GEOS-Chem v13.3.1 to simulate $NH_3$ dry deposition velocity over China for the year 2015. The model was driven by assimilated meteorological data from NASA's MERRA-2 reanalysis. Simulations were conducted on a nested horizontal grid of $0.5° \times 0.625°$ covering the domain of 60°E-149.375°E and 11°S-54.5°N (Lu et al., 2025).

**2.4.2 Simulation of $V_d$ by using random forest machine learning algorithm**

To improve the spatial resolution and model efficiency, we used the GEOS-Chem model based $V_d$ simulations to train a random forest model that can predict dry deposition velocities under various meteorological and land surface conditions and with finer spatial resolution for the entire study period. This data-driven approach enables downscaling to a 0.25° resolution and extends predictions to the entire study period from 2013 to 2023 by using ERA5 reanalysis data.

The random forest (RF) algorithm is a widely adopted ensemble machine learning method that integrates multiple decision trees using the bagging strategy to capture complex nonlinear relationships between predictors and response variables. Overall, the RF model was used for two purposes, (1) for simulating dry deposition velocity ($V_d$) across 2013-2023, which is displayed in this Section; and (2) to simulate $NH_3$ concentration and identify key drivers of atmospheric $NH_3$ changes as illustrated in Section 2.6.1. This RF model has been widely used in atmospheric environment assessments, nitrogen management in agriculture, and model validation studies, providing a robust framework for evaluating the ecological impacts of $NH_3$ deposition (Asadi et al., 2021; Ai et al., 2024; Zhang et al., 2024). As shown in Figure. S2, the RF model was trained on multiple bootstrapped datasets and evaluated by aggregating outputs from multiple trees to obtain stable and accurate predictions. We selected five meteorological and hydrological variables from ERA5 reanalysis data as predictors: planetary boundary layer height, 10 m wind speed, volumetric soil water of surface layer, surface temperature, and total

precipitation. The dataset was randomly split into a training set (60%) and a validation set (40%),
the comparisons of $V_d$ simulation by using GEOS-Chem and RF model are evaluated in Section

322    3.4.1.


**2.5 Geographical division in China and other supporting data**

To investigate spatial heterogeneity in interannual trends, China was divided into nine
subregions based on the classification system from the Resource and Environmental Science
Data Center (Figure 1b). These regions include: Northeast China Plain, Yunnan-Guizhou
Plateau, Northern Arid and Semi-Arid Region, Southern China, Sichuan Basin and Surrounding
Areas, Middle-Lower Yangtze Plain, Qinghai-Tibet Plateau, Loess Plateau, and Huang-Huai-
Hai Plain. Table S2 summarizes the dominant land cover types and their proportional areas
within each subregion, and the provinces contained in each region are listed in Table S3 (*SI*),
the details of main land cover categories and corresponding proportions in each region are also
displayed in Figure 1b and Text S2 (*SI*).

To clarify the characteristics of atmospheric $NH_3$ concentrations and dry deposition flux across
different land cover types, we utilized the 30-meter resolution China annual Land Cover Dataset
(CLCD) to classify surface types. The CLCD is the first annual land cover product for China
derived from Landsat imagery, covering the period from 1985 to 2022 (Yang et al., 2021). The
dataset categorizes land cover into nine classes: cropland, forest, shrubland, grassland, water
bodies, snow/ice, barren land, impervious surfaces, and wetlands. Based on this classification,
we conducted a systematic analysis of the spatial variation and temporal trends in $NH_3$
concentrations and dry deposition fluxes across different land surface types.

In this study, multiple emission inventories of $SO_2$, $NO_x$, and $NH_3$ were utilized to investigate
the drivers behind changes in atmospheric $NH_3$ concentrations and to assess potential future
trends. The reason of using multiple emission inventories instead of only EDGAR is based on
the fact that many previous studies have concluded large potential bias in using a single
inventory caused by highly uncertain emission factors and activity data discrepancies (Crippa

et al., 2019; Liu et al., 2024a). Therefore, we make full use of all available inventories from different data sources to provide robust evaluation of their emission changes. The emission inventories for $SO_2$ and $NO_x$ include: (1) the Inversed Emission Inventory for Chinese Air Quality (CAQIEI, https://www.scidb.cn/en/detail?dataSetId=81cc0de9c68b4a4981e2f295ac612fbf); (2) the Multi-resolution Emission Inventory for China (MEIC, http://meicmodel.org.cn/?page_id=560); (3) the Air Benefit and Cost and Attainment Assessment System - Emission Inventory (ABaCAS, https://abacas-dss.com/abacasChinese/Default.aspx); (4) the Community Emissions Data System (CEDS, https://github.com/JGCRI/CEDS/); and (5) the Emissions Database for Global Atmospheric Research (EDGAR, https://edgar.jrc.ec.europa.eu/dataset_ap81#p3). Due to the relatively late development of ammonia ($NH_3$) research and the limited availability of comprehensive emission inventories, this study employed only two datasets—EDGAR v8.1 and MEIC—for $NH_3$ emission analysis. In addition, the Dynamic Projection model for Emissions in China (DPEC, http://meicmodel.org.cn/?page_id=1917), developed by Tsinghua University, was used to project future emission trends. Further details on all six emission inventories are provided in Text S3 and Table S4-S5 (*SI*). Note the emissions from EDGAR will be used in this study to simulate spatial-temporal patterns of $NH_3$ concentration. Note the EDGAR does not include biomass burning. However, we also extracted emissions from biomass burning from the MEIC inventory for 2013-2020, the total emissions of $SO_2$, $NO_x$, and $NH_3$ during this period in China, as well as the average annual emissions and their proportions from biomass burning were displayed in Table S6 (*SI*). And the contribution of biomass burning to these three gases was less than 3%, indicating relatively small influence of biomass burning in simulating $NH_3$ concentrations.

**2.6 Quantification of influencing factors to annual trend of $NH_3$ concentration and dry deposition**

**2.6.1 Simulation of ground $NH_3$ concentration by using random forest model**

To assess the contributions of meteorological conditions and emissions to $NH_3$ concentrations

over the study period, we constructed another RF model to simulate ground-level $NH_3$ concentration. Here the CrIS-retrieved $NH_3$ concentrations for 2022 were used to train this RF model considering the most updated emission inventory is available for 2022, and input parameters included five ERA5-derived meteorological and hydrological variables (ABL height, wind speed, soil moisture, temperature, and precipitation) and three emission datasets from the EDGAR inventory ($SO_2$, $NO_x$, and $NH_3$ emissions). To isolate the effects of emissions and meteorological variables, we conducted a few sensitivity experiments using the 2022-trained model as the baseline. By holding emissions constant or regressing meteorological data back to 2013 (and vice versa), we simulated $NH_3$ concentrations attributable solely to changes in meteorology or emissions (for all or each of $NH_3$, $SO_2$ and $NO_x$). The contributions of each factor were then normalized to calculate the percentage influence on $NH_3$ concentration changes. Note previous modeling results (i.e. $PM_{2.5}$) always suffers from bias in 1/3 of modeling days and it's better to choose days with good predictions. And in this study for $NH_3$ observations, they were measured by passive sampler, representing averages of one week instead of hourly or daily scales. Therefore, to avoid the random errors from observations and simulations, monthly average was conducted for $NH_3$ concentration for machine learning.

**2.6.2 Quantification of influencing factors to annual trends of $NH_3$ concentration**

We further used the logarithmic differentiation method to decompose the relative contributions of $NH_3$ concentration and dry deposition velocity to the overall change in dry deposition flux. The logarithmic form allows the multiplicative relationship to be transformed into an additive form, making it suitable for quantifying variable impacts, particularly when concentration and velocity change in opposite directions. The decomposition is based on the following:

$$\Delta \ln F = \Delta \ln C + \Delta \ln V_d \qquad (2)$$

The respective contributions of concentration ($\Delta \ln C$) and deposition velocity ($\Delta \ln V_d$) are calculated as:

$$\eta_C = \left| \frac{\Delta \ln C}{\Delta \ln F} \right| \qquad (3)$$

$$\eta_{V_d} = \left| \frac{\Delta \ln V_d}{\Delta \ln F} \right| \qquad (4)$$

where $\Delta$ln denotes the change in the natural logarithm, $\eta_C$ and $\eta_{Vd}$ represent relative contributions from $NH_3$ concentration and dry deposition velocity to dry deposition of F, respectively. These contributions were normalized to provide intuitive percentage values. This method is particularly effective in quantifying dynamic and opposing changes and does not assume linear relationship, offering a more robust analysis than traditional linear regression. Additionally, the Mann-Kendall (MK) trend test was employed to statistically evaluate the temporal trends in $NH_3$ concentrations over the study period (Text S1, *SI*).

**3 Results and Discussions**

**3.1 Spatial patterns of near surface satellite $NH_3$ concentration and its trend analysis**

Using CrIS satellite-derived near surface $NH_3$ concentrations (representing average between ground to ~1 km) from 2013 to 2023, a high-resolution (0.1° × 0.1°) monthly averaged $NH_3$ concentration dataset across China over an 11-year period was generated. The observation from the near surface layer can reflect the impact of human activities and natural source emissions on the near-Earth atmospheric environment. We first displayed the annual averaged spatial patterns and its trend from 2013-2023 at both the national scale and within specific subregions, followed by an analysis of seasonal variations (Figures 2a-j and Figures S3-S7, *SI*). The results of the annual average indicate that the North China Plain (also known as the Huang-Huai-Hai Plain) consistently exhibited the highest $NH_3$ concentrations (>10 ppb) during the study period (Figure 2a). This region is recognized as one of China's most intensive agricultural zones, accounting for approximately 25% of China's total arable land area and grain production (Song et al., 2024), and is thus subject to frequent fertilizer application, contributing significantly to elevated $NH_3$ emissions and corresponding concentration.

The secondary $NH_3$ concentration hotspots were observed in the Guanzhong Plain in Shaanxi Province and the southeastern margin of the Tibetan Plateau. The Guanzhong Plain region is another major agricultural production area in western China, with cultivated land accounting

for 49.4% of Shaanxi Province's total arable area. Intensive fertilizer application and related activities are the main sources of $NH_3$ emissions in this region. The elevated $NH_3$ concentrations in southeastern Tibet are likely attributed to emissions from extensive livestock farming, particularly yak and sheep husbandry. In addition to these agricultural and pastoral regions, relatively high $NH_3$ concentrations were also observed in arid zones such as Xinjiang and Inner Mongolia. However, these apparent $NH_3$ enhancements are likely artifacts of satellite retrievals potentially influenced by surface radiative properties or dust. Higher accuracy is typically associated with higher thermal contrast; conversely, lower thermal contrast would lead to higher uncertainties in $NH_3$ retrievals, leading to overestimation of $NH_3$ concentrations due to limitations in retrieval algorithms and thermal contrast biases (Liu et al., 2020b).

To further explore spatial patterns in temporal change, the pixel-wise trend analysis of annual $NH_3$ concentrations was also conducted (Figure 2b). Significant positive trends ($>0.4$ ppb yr$^{-1}$) were found in the central and eastern parts of China, particularly in major agricultural zones with intensive crop fertilization. These results are consistent with findings by Warner et al. (2017), who reported a substantial increase in $NH_3$ concentrations over eastern China using AIRS data from 2002 to 2016. Our study extends this trend through 2023, indicating that $NH_3$ concentrations in these regions have continued to rise significantly in recent years. In contrast, western China generally showed stable or declining trends. Although northern Xinjiang exhibited moderate $NH_3$ increases in areas where the trend passed significance testing, other parts of the west demonstrated declining trends. This pattern may be associated with grassland restructuring policies implemented by the Chinese government to reduce overgrazing and restore degraded ecosystems. These measures have significantly alleviated the ecological pressure on grasslands and fostered the transformation and upgrading of grassland animal husbandry, as well as environmental optimization. Therefore, with policy support, they contribute to reducing environmental pollution from animal husbandry in grassland areas, thereby lowering $NH_3$ emissions.

The spatial patterns of $NH_3$ concentration increases correspond closely to regions of high

population density and agricultural land cover types, such as the North China Plain and Sichuan
Basin. These areas are also hotspots for reductions in $SO_2$ and $NO_x$ emissions due to stringent
air pollution control measures as displayed in Figures S8-S9 (*SI*). The decline in acid gases may
reduce atmospheric neutralization capacity, thereby enhancing the lifetime and apparent
abundance of $NH_3$ in the atmosphere (Dong et al., 2023), contributing to the pronounced
upward trends observed in these regions.

We also displayed the seasonal variations and its trend during 2013-2023, clear seasonal
differences in $NH_3$ spatial distribution were observed during the whole study period (Figures
2c-j, Figures S4-S7, *SI*). In spring, the $NH_3$ distribution resembled the annual pattern but
exhibited concentrations approximately 13.9% higher. The Huang-Huai-Hai Plain showed
especially concentrated and elevated values, likely due to extensive fertilizer use during spring
planting. In contrast, the northwest exhibited little seasonal deviation from annual averages, as
emissions are more influenced by pastoral activities than by seasonal patterns of fertilization in
agricultural regions. In autumn, $NH_3$ levels declined sharply, despite localized fertilizer
application, primarily due to reduced emissions and cooler temperatures. High concentrations
remained in Shandong Province and adjacent regions. Winter concentrations were the lowest,
reflecting widespread agricultural dormancy and low temperatures, although lower thermal
contrast and reduced $NH_3$ signal strength increase retrieval uncertainties.

In summer, $NH_3$ concentrations peaked across China, with higher concentration regions
expanding westward into semi-arid areas. This peak seasonality contrasts with trends in Europe
and the U.S., where springtime peak is also more typical. In China, summer fertilization is
applied for the key agricultural crops as rice paddy, maize, corn and wheat—often involving
both mineral and organic fertilizers—contributes to the observed summer peak (Paulot et al.,
2014; Luo et al., 2025). Elevated temperatures further enhance volatilization from manure of
agricultural area and urban waste in cities, intensifying atmospheric $NH_3$ concentration.
Although urbanization has increased over the past decade, many system-scale farms continue
to be used for agricultural production. As reported by Liu et al. (2024d) that temperature

increases accounted for up to 20.0% of urban $NH_3$ increases between 2008 and 2019. Notably, elevated $NH_3$ levels were also observed along the Yangtze River basin, corresponding to fertilizer use in rice paddies.

The spatial distributions of the 11-years trend analyses for each season are also displayed (Figures 2d, f, h and j), they show significant increases across eastern China, particularly during summer and autumn. Overall, these results indicate the annual trend of surface $NH_3$ concentration occurred throughout each season. Winter trends were the weakest in magnitude and spatial extent. Consistent with annual patterns, the North China Plain and Sichuan Basin showed the most pronounced increases. There was no significant change in trend in most parts of western China. There was a slight increasing trend in summer and autumn in northern Xinjiang. Other regions exhibiting a significant trend were decreasing.

**3.2 Temporal variation of near surface satellite $NH_3$ concentrations for different regions**

In this section, we continue to present the spatiotemporal near-surface $NH_3$ concentrations derived from CrIS lower ABL mixing ratio values. The temporal variation of annual $NH_3$ concentrations and across different seasons from 2013 to 2023 is displayed in Figure 3a. Over this period, the annual mean $NH_3$ concentration in China increased by 22.5%, with seasonal increases of 13.8% in spring, 30.6% in summer, 26.4% in autumn, and 18.1% in winter, respectively. Among these seasons, summer exhibited the highest mean concentration (3.60 ppb), followed by spring (3.28 ppb), with annual, autumn, and winter means recorded at 2.88 ppb, 2.63 ppb, and 2.00 ppb, respectively (Table 1). The Mann-Kendall trend test results (Table 1) indicated statistically significant upward trends for spring, summer, autumn, and annual mean concentrations ($p < 0.05$). Although winter showed a positive trend ($Z > 0$), it did not reach statistical significance. The seasonal rates of increase, in descending order, were: summer (0.065 ppb yr$^{-1}$), autumn (0.050 ppb yr$^{-1}$), annual (0.045 ppb yr$^{-1}$), spring (0.039 ppb yr$^{-1}$), and winter (0.023 ppb yr$^{-1}$). The most pronounced increase during summer from 2013 to 2023 also aligns with previous findings by Liu et al. (2018), which only analyze the North China Plain region from 2008 to 2016. However, their trend is slightly lower than our results, the

comparisons reveal a significant increase in $NH_3$ concentrations after 2016, which could
potentially be attributed to enhanced $NH_3$ emissions, favorable climatic conditions, or a
decrease in $NO_x/SO_2$ emissions, as discussed and quantified below.

The increasing summer trend of atmospheric $NH_3$ is likely related to global warming in study
period (Figure S10, *SI*). The summer temperatures in China rose by 0.3°C from 2013 to 2023.
As reported in our previous study on the U.S. Corn Belt, $NH_3$ emissions are projected to increase
by a factor of 2.5 for every 10°C rise in summer temperatures (Hu et al., 2020; 2021). Other
studies also showed that over 40% of fertilizer application and approximately 25% of livestock
emissions occur during the summer months (Xu et al., 2015; Kang et al., 2016), which enhances
$NH_3$ volatilization from ground to atmosphere. The slower rate of increase in spring may be
associated with China's national fertilizer reduction policies, such as the "Action Plan for
Fertilizer Reduction by 2025". Fertilizer use increased until peaking in 2015 and subsequently
declined for eight consecutive years, resulting in a 15.1% reduction from 2013 to 2023, with
the national application totaling 50.22 million tons in 2023 (Figure S11, *SI*).

The decrease in chemical fertilizer use, combined with the adoption of organic fertilizers, has
contributed to a gradual slowdown in the rise of $NH_3$ concentrations. By 2024, the nitrogen use
efficiency (NUE) for rice, maize, and wheat reached 42.6%, helping to reduce fertilizer input
without compromising yields and mitigating $NH_3$ emissions and nutrient pollution. Zhan et al.
(2021) identified improving NUE as the most effective and cost-efficient strategy for $NH_3$
mitigation in agriculture, a finding supported by cost-benefit assessments. Autumn also showed
a substantial increase in $NH_3$ concentrations, second only to summer. Current emission
reduction efforts have primarily focused on spring and summer, reflecting crop planting cycles,
while autumn has often been overlooked, contributing to this seasonal gap in mitigation. These
findings highlight the need for seasonally and crop-specific emission control strategies in future
$NH_3$ management efforts.

Significant spatial heterogeneity was observed in the interannual variation of $NH_3$

concentrations across different regions. Figure 3b illustrates long-term trends in $NH_3$ concentrations for nine subregions. Most regions exhibited increasing trends, with the Huang-Huai-Hai Plain standing out for its consistently elevated concentrations—approximately twice as high as the national average (Table 2). This region is China's primary agricultural zone, characterized by high population density and intensive agricultural activity, both of which contribute to substantial $NH_3$ emissions. Additionally, it has been a focal area for $SO_2$ and $NO_x$ emission reductions, and the combined effects of high emissions and reduced atmospheric neutralization capacity have led to persistent $NH_3$ accumulation.

The trend analysis further revealed statistically significant upward trends in the Huang-Huai-Hai Plain, the Northern Arid and Semi-Arid Region, the Loess Plateau, the Middle-Lower Yangtze Plain, South China, the Northeast China Plain, and the Sichuan Basin and its surrounding regions. We used compound annual growth rate (CAGR) method to calculate the annual growth rate of $NH_3$ concentration across the country and in the Huang-Huai-Hai Plain region. The Huang-Huai-Hai Plain showed the steepest increase, with an average annual rise of 0.24 ppb, corresponding to a 6.0% per year growth rate—3 times the national average of 2.0% (Manisha et al., 2023). The primary driver of this sharp increase is the marked reduction in atmospheric $SO_2$, which has disrupted the $NH_3$-acid gas neutralization balance (Xu et al., 2019a). The Loess Plateau ranked second, with an average increase of 0.14 ppb per year. In contrast, the Yunnan-Guizhou Plateau exhibited a mild, non-significant increase, with relatively stable concentrations. The Tibetan Plateau showed a slight downward trend, which also lacked statistical significance ($p > 0.05$), indicating a relatively stable $NH_3$ regime in this high-altitude, low-emission region.

**3.3 Comparison between satellite and ground-based $NH_3$ observations and adjustment from surface level to ground-level $NH_3$ concentration**

As stated in Section 2.1, although satellite-based observations provide extensive spatial coverage and long-term data for atmospheric $NH_3$ studies, they have limited vertical profile resolution of mixing ratio values near the surface that often cannot capture the reported fine

scale vertical gradient in the lower ABL created from the reactive nature of ammonia and its role in chemical transformation processes (Hu et al., 2020; 2021; Griffis et al., 2019). Further, the dry deposition of $NH_3$ is the product of ground-level (usually calculated by site-based observations of 1~1.5 m height) $NH_3$ concentration and dry deposition velocity. Therefore, to enable accurate estimation of $NH_3$ dry deposition, we conducted a comparative analysis between satellite-derived and multiple years of observations at 24 ground-based $NH_3$ sites, and their relationship will be used to adjust the lower vertical resolution satellite observations to ground-based surface observations.

As shown in Figure 4a, the scatter plots of monthly averaged site-based ground-level $NH_3$ concentrations and corresponding satellite-based observations exhibit a strong correlation with a coefficient of determination ($R^2$) of 0.62 and a root mean square error (RMSE) of 3.56 ppb. Note, to minimize the random error, each plot in Figure 4a represents averages of all observations at urban or rural sites during each overlap month. Overall, it illustrates that the ground-level measurements are, on average, approximately twice as high as those retrieved by satellite. This discrepancy can be attributed to the vertical gradient of $NH_3$ in the atmosphere: ground-based sensors typically local point source observations operate at heights of 1-1.5 m, while satellite observations are regional (14 km) with low vertical resolution (~1km or more), which is shown from the averaging kernels (Shephard et al.,2011, Shephard et al., 2020). Many pioneer studies have demonstrated that when the land surface acts as an $NH_3$ source, its vertical distribution decreases logarithmically with height (Hu et al., 2020; 2021; Shephard et al., 2011; Shephard et al., 2020). For example, our previous studies of tall tower observations in the United States reported an $NH_3$ mixing ratio gradient of $-0.27$ ppb per 100 m, with modeled gradients ranging from $-0.21$ to $-0.84$ ppb per 100 m (Hu et al., 2020; 2021; Griffis et al., 2019), showing good agreement between observations and simulations of the vertical profiles within boundary layer. When using the gradient of above reported values, the average of 0-1000 m column $NH_3$ concentration should be around 1~ 4 ppb lower than ground-level, this pronounced vertical gradient is a major reason for the systematic underestimation of $NH_3$ by satellites when compared with ground-level observations.

To address this inconsistency, we used the regression relationship derived from Figure 4a to
adjust the satellite retrievals. After correction, a new regression (Figure 4b) shows a nearly 1:1
agreement between satellite and ground-based measurements, with the RMSE reduced from
3.56 ppb to 1.69 ppb. The purpose of the linear regression equation is to adjust the column-
averaged $NH_3$ concentration to the ground-level at 1.5 m, as described in Section 2.2. The
approach we used is applying an additive shift (bias correction), where $R^2$ remain almost the
same (Figures 4a-b ), it's also based on "K-theory" (gradient diffusion theory) with the well-
mixed assumption in the ABL. This method assumes that transport flux can be represented
analogously to molecular diffusion, where fluxes are proportional to the mean gradient of the
transported quantity. This adjustment enables the derivation of $NH_3$ dry deposition, which can
then be compared with global observations. The reason that the $R^2$ value remained unchanged
is that the same equation, y=0.35+0.16, was applied to all scatter plots. This theoretically affects
only the RMSE and does not influence the $R^2$ value. The reduction in RMSE further indicates
that this approach effectively adjusts the column-averaged $NH_3$ concentration to the ground-
level at 1.5 m. The conversion is given by x=(y-0.16)/0.45=2.22y-0.36 , where y represents the
CrIS satellite-based column-averaged $NH_3$ concentration (from ground to 1 km), and x denotes
the $NH_3$ concentration after adjustment to 1.5 m. This approach is conceptually similar to using
a simple multiplicative (or additive) conversion factor. It is important to acknowledge that
spatial-temporal uncertainties or potential systematic biases may exist in the relationship
between ground-based and satellite-derived $NH_3$ observations across different regions and
under varying thermal contrast and boundary-layer conditions. As demonstrated in the scatter
plots of Figures 4a-b, which exhibit significant variability. Nevertheless, the regression slope
and associated uncertainty were 0.45 ± 0.04, indicating that the potential systematic biases
mentioned above could result in an error of approximately 9% when deriving ground-level $NH_3$
concentrations and dry deposition rates. This error was calculated by dividing the uncertainty
extent (0.04) by the regression slope (0.45). These uncertainties can be mitigated by increasing
the number of ground-based $NH_3$ observations in diverse regions in future studies.

To further assess the adjustment effectiveness, we selected the year 2015—when both satellite

and ground data are available—for analysis. As shown in Figure 4c, the adjusted satellite-based NH$_3$ concentrations closely match ground observations across almost all sites, confirming the reliability of using the adjustment approach. This adjustment function was then applied to the full 2013-2023 satellite dataset to improve the reliability of NH$_3$ dry deposition estimates. Table 2 illustrated the adjusted average of ground-level NH$_3$ concentrations across different regions, with the Huang-Huai-Hai Plain exhibiting the highest value of 11.36 ppb. This was followed by the Northern Arid and Semi-Arid Region (6.93 ppb), the Qinghai-Tibet Plateau (6.48 ppb), and the Loess Plateau (6.05 ppb). Although the height-corrected NH$_3$ concentrations in these regions ranked immediately after the Huang-Huai-Hai Plain, their values were approximately two times lower than those observed in the Huang-Huai-Hai Plain.

### 3.4 Estimation of spatiotemporal variations of NH$_3$ dry deposition across China

### 3.4.1 Simulation of spatiotemporal dry deposition velocities

As illustrated in Method Section 2.4, to estimate NH$_3$ dry deposition flux across China, we first used the GEOS-Chem model to simulate NH$_3$ dry deposition velocities for the year 2015 (Figure 5a). Considering the high computational cost, limited temporal flexibility and spatial resolution of the GEOS-Chem model, we adopted a hybrid modeling approach by training a random forest (RF) machine learning model on the GEOS-Chem model-based simulation results. This approach allowed us to extend the simulation to the full 2013-2023 period, while improving both computational efficiency and spatial resolution from $0.5° × 0.625°$ to $0.25° × 0.25°$.

The resulting RF-predicted dry deposition velocities for 2015 show high spatial agreement with the GEOS-Chem outputs (Figure 5b and Figure S12, *SI*). Both models identify southern China as a hotspot for dry deposition velocity, likely due to the region's warm and humid conditions that facilitate gaseous NH$_3$ deposited onto ground surface. Additionally, southern China is a major rice-producing region where surface resistance in paddy fields is lower than in dryland fields, further enhancing dry deposition rates. Figure 5c shows the differences between the two model outputs, with over 99% of grid cells having discrepancies less than 0.1 cm s$^{-1}$, indicating

strong consistency and validating the reliability of the RF model for long-term simulations.
Using this trained model, we further simulated $NH_3$ dry deposition velocities from 2013 to 2023
at monthly averages.

**3.4.2 The spatiotemporal variations of $NH_3$ dry deposition in China**

With the adjusted spatiotemporal ground-level $NH_3$ concentrations and simulated deposition
velocities from 2013 to 2023, we derived the monthly grid-level $NH_3$ dry deposition flux for
China. These were further aggregated to estimate average $NH_3$ dry deposition flux and total
deposition over different land cover types (Figure 6; Figure S13, *SI*). Figure 6a illustrates the
spatial distribution of $NH_3$ dry deposition flux average from 2013 to 2023. Distinct spatial
differences are evident, where the eastern coastal regions exhibited significantly higher
deposition flux than inland areas, with values higher than 1.8 g $NH_3$ $m^{-2}$ $yr^{-1}$. Notably, the
Huang-Huai-Hai Plain and the southwestern region of the Qinghai-Tibet Plateau emerged as
prominent hotspots of $NH_3$ dry deposition, highlighting the substantial impact of intensive
agricultural activities and industrial emissions. Elevated deposition rates were also observed in
the southern Tibetan Plateau, driven by locally high $NH_3$ concentrations.

A trend analysis of dry deposition over the 11-year period (Figure 6b) shows statistically
significant increases in deposition flux in eastern coastal areas (> 0.1 g $m^{-2}$ $yr^{-1}$), likely
reflecting rising $NH_3$ concentrations in these regions. In contrast, western China shows minimal
change, with some areas even exhibiting slight declines. Unlike the $NH_3$ concentration trends,
there is no region in western China that displayed a statistically significant increase in dry
deposition flux, which was caused by the trend of $V_d$ in this region, emphasizing the spatial
decoupling between emission intensity and deposition patterns in less industrialized regions.

The interannual variation of $NH_3$ dry deposition also exhibited significant spatial heterogeneity
at the regional scale (Figure 6c and Table 3). The Huang-Huai-Hai Plain, characterized by
persistently high $NH_3$ concentrations, recorded the highest area-specific dry deposition flux,
reaching 1.06 g $m^{-2}$ $yr^{-1}$—approximately twice the levels observed in other regions. MK trend
analysis indicated a significant increasing trend in dry deposition flux across all regions except
the Tibetan Plateau, where a weak downward trend was observed but was not statistically
significant. The most pronounced increase was found in the Huang-Huai-Hai Plain, with an
average annual increment of 0.05 g m$^{-2}$ yr$^{-1}$, followed by the middle and lower reaches of the
Yangtze River, at 0.03 g m$^{-2}$ yr$^{-1}$, detailed numbers are displayed in Table 3.

**3.4.3 Comparisons of ground-level NH$_3$ concentration, dry deposition velocity and flux in**
**different land cover types**
In addition to meteorological factors, land cover types play a pivotal role in regulating dry
deposition processes. In this section, we annually extracted and compared ground-level NH$_3$
concentrations, dry deposition velocities, and dry deposition fluxes across different land cover
categories. The analysis focused on four representative land-use types—urban, cropland, forest,
and grassland—selected based on their distinct NH$_3$ emission characteristics (Figure 7; Table
S7, *SI*). The average NH$_3$ concentrations, ranked from highest to lowest, were: urban (8.76 ppb),
cropland (6.27 ppb), national average (6.01 ppb), grassland (5.72 ppb), and forest (3.76 ppb)
(Figure 7a). Urban areas exhibited both the highest concentrations and the largest interannual
variability, with a statistically significant upward trend ($p < 0.05$, $Z > 1.96$), increasing at an
average rate of 0.39 ppb yr$^{-1}$. This trend is primarily attributed to anthropogenic sources such
as vehicular emissions, as well as the urban heat island effect, which raises urban temperatures
by 1-3°C—and occasionally by over 10°C—relative to surrounding rural areas (Santamouris et
al., 2013; Cao et al., 2016; Chang et al., 2021). These elevated temperatures, further amplified
by global warming, facilitate enhanced NH$_3$ volatilization within cities.

While ground-level NH$_3$ concentrations over grassland areas remained relatively stable
throughout the study period, cropland regions exhibited a continuous upward trend, with the
two trends intersecting in 2016 (Figure 7a), after which NH$_3$ concentrations in croplands
exceeded those in grasslands. NH$_3$ emissions in grassland ecosystems are predominantly
associated with livestock grazing, and the stabilization observed is likely attributable to the
implementation of grazing restrictions and ecological restoration policies. In contrast, despite
the introduction of fertilizer reduction policies in some agricultural areas, rising food demand
driven by population growth has sustained or even increased fertilizer application, thereby
contributing to the observed increase in cropland $NH_3$ concentrations. At the national scale,
$NH_3$ concentrations exhibited a statistically significant upward trend, with an average increase
of 0.10 ppb $yr^{-1}$ (equivalent to an annual growth rate of 2.2%). Forested regions, which are
minimally impacted by anthropogenic sources such as synthetic fertilizers and livestock
emissions, maintained the lowest and most stable $NH_3$ concentrations, showing only a slight
upward trend that may be linked to climate warming (Figure 7a; Figure 8).

Dry deposition velocities exhibited limited interannual variability across different land cover
types. Forested areas recorded the highest average deposition velocity, likely attributable to
greater surface roughness and enhanced canopy-induced turbulence, followed by urban and
cropland regions (Figure 7b; Figure 8). The mean $NH_3$ dry deposition velocities for forest, urban,
cropland, grassland, and the national average were 0.43, 0.42, 0.40, 0.32, and 0.36 cm $s^{-1}$,
respectively. Mann-Kendall trend analysis revealed statistically significant increasing trends in
urban and cropland areas, with annual rates of 0.0013 and 0.0012 cm $s^{-1}$ $yr^{-1}$, respectively.
Although forests maintained the highest mean velocity and exhibited a positive trend, the
change was not statistically significant. At the national scale, deposition velocity showed a weak
but consistent upward trend. In contrast, grassland areas experienced a slight decline in
deposition velocity over the 11-year period, though this trend was not statistically significant.

Area-specific $NH_3$ dry deposition fluxes closely followed the spatial distribution of atmospheric
concentrations across different land cover types (Figure 7c; Figure 8). Urban regions exhibited
the highest deposition flux (0.88 g $m^{-2}$ $yr^{-1}$), followed by cropland areas (0.61 g $m^{-2}$ $yr^{-1}$). Both
urban and national average fluxes demonstrated statistically significant upward trends over the
study period. The steepest increase was observed in urban areas, with a rate of 0.04 g $m^{-2}$ $yr^{-1}$—
approximately four times the national average—followed by croplands at 0.03 g $m^{-2}$ $yr^{-1}$. Our
findings also agree the previous study by Chen P et al. (2023), which conducted that, although
fertilizer application has been partially reduced under agricultural emission control policies,

non-agricultural sources—such as industrial processes and transportation—have become the predominant contributors to $NH_3$ emissions in China, particularly concentrated in urban areas. This shift has contributed to elevated $NH_3$ concentrations and enhanced dry deposition fluxes in cities.

In contrast, forests and grasslands showed relatively stable fluxes, likely due to lower levels of anthropogenic disturbance. Nevertheless, a statistically significant increasing trend in forest deposition flux was detected, which may have important ecological implications. Sustained increases in $NH_3$ deposition could lead to adverse effects such as plant nutrient imbalances, biodiversity loss, and eutrophication of adjacent aquatic systems, potentially compromising forest health and long-term ecosystem stability. Furthermore, interannual variability in dry deposition was more pronounced in urban areas, reflecting the dynamic nature of urban development and emission variability, whereas cropland fluxes exhibited a more gradual trend in response to evolving fertilizer management practices.

Trends in total $NH_3$ dry deposition across different land cover types generally mirrored those of area-specific fluxes; however, total dry deposition values of $NH_3$ were modulated by the area of each land cover type. Grasslands accounted for the largest share of annual total $NH_3$ dry deposition (1.23 Tg), followed by croplands (1.15 Tg), forests (0.92 Tg), urban areas (0.21 Tg), and a national total of 4.85 Tg. Over the 11-year study period, statistically significant upward trends in total dry deposition were observed at the national scale, as well as in cropland, forest, and urban areas, with annual increases of 0.10, 0.05, 0.03, and 0.01 Tg yr$^{-1}$, respectively. Although grasslands also exhibited an increasing trend, it was not statistically significant. Changes in annual total $NH_3$ dry deposition are driven not only by atmospheric concentrations and deposition velocities but also by land-use dynamics (Figure 7d; Figure 8). In particular, the continuous expansion of urban areas from 2013 to 2023 contributed substantially to the increasing trend in total urban $NH_3$ deposition (Figure S14, *SI*). These findings highlight the importance of considering both biogeochemical processes and anthropogenic land-use changes in assessing long-term trends in reactive nitrogen deposition.

**3.5 Simulation of ground-level NH$_3$ concentration and contribution factors analysis to both NH$_3$ concentration and deposition flux**

In this section, we quantified and partitioning the contributions influencing the trends in the NH$_3$ concentration and dry deposition flux, and further investigated the key drivers of atmospheric NH$_3$ concentrations using the Random Forest (RF) regression model. Model performance was evaluated by comparing simulated NH$_3$ concentrations with observations for the period 2013-2023, showing good agreement (Figure 9). The RF model effectively captured the spatial variability of NH$_3$ concentrations, with deviations generally within ±0.1 ppb, indicating robust predictive capability. The input variables were categorized into two major groups: meteorological factors and anthropogenic emissions, including NH$_3$ emissions as well as SO$_2$ and NO$_x$ emissions. The feature-importance ranking figure illustrates the relative importance of eight driving factors in predicting NH$_3$ concentrations using the Random Forest model (Figure S15, *SI*). Among the emission and meteorological-hydrological factors, the latter plays a more prominent role in explaining the spatial and temporal variability of NH$_3$ concentrations. Within the meteorological-hydrological factors, the 10-meter wind speed (20.3%), 2-meter temperature (14.9%), and boundary layer height (13.1%) are the most influential variables affecting the NH$_3$ concentration simulation. These variables collectively reflect the role of atmospheric diffusion capacity and volatilization conditions in regulating the distribution of NH$_3$ concentrations. Total precipitation (11.0%) and surface soil moisture content (13.6%) contribute to the removal of NH$_3$ from the atmosphere, though their relative importance is lower. Among the emission factors, NH$_3$ emissions (16.4%) are the most significant, followed by NO$_x$ (11.0%) and SO$_2$ (5.1%) emissions. This suggests that, in addition to direct emissions, precursor chemical processes also have an indirect influence on the distribution of NH$_3$ concentrations.

To quantify the contribution of emissions and meteorological factors to changes in NH$_3$ concentrations, we used a random forest model to simulate NH$_3$ concentration with different sensitivity test by replacing single factor, and the difference between them can be treated as contributions from corresponding factor. Figure 10a shows the adjusted ground-level NH$_3$

concentration in 2022 and the simulation results under three different meteorological and emission scenarios. The simulated concentrations are 3.08 ppb, 3.14 ppb, 3.10 ppb. Both meteorological and emission contributions are calculated from the simulation results. Simulation results from the random forest model showed that anthropogenic emissions were the main driver, accounting for approximately 77.4% of the $NH_3$ concentration changes, while meteorological conditions accounted for the remaining 22.6% (Figure 10a).

The above relative contributions are calculated by the method that using the emissions and meteorological-hydrological factors from 2013 as the baseline (more details in Method Section 2.6.1), we first simulated the $NH_3$ concentration for 2022 using the emissions and meteorological-hydrological factors from that year. The simulated concentration was 3.08 ppb, which was consistent with the satellite-observed concentration for 2022, yielding a relative error of 0.1%. Subsequently, we replaced the emissions data with those from 2013 while keeping the 2022 meteorological-hydrological factors constant, resulting in a simulated concentration of 3.14 ppb. We then replaced the meteorological-hydrological factors with those from 2013 while keeping the 2022 emissions constant, leading to a simulated concentration of 3.10 ppb. By subtracting the two simulated concentrations from the 2022 $NH_3$ concentration simulation, we quantified the effects of changes in emissions and meteorological-hydrological factors on $NH_3$ concentration. Finally, the results were normalized, revealing that the relative contributions of emissions and meteorological-hydrological factors to the concentration changes were 77.4% and 22.6%, respectively.

Among meteorological parameters, air temperature emerged as the most influential factor, whereas other variables (e.g., relative humidity, wind speed) exhibited minimal interannual variation and lower predictive importance. Analysis of ERA5 reanalysis data revealed a persistent warming trend over the past decade, with the annual mean surface temperature in 2023 being 8.4% higher than in 2013 (Figure S10, *SI*). Previous studies, such as Hu et al. (2020), reported an exponential relationship between $NH_3$ mixing ratios and temperature, with $NH_3$ concentrations increasing from 4 ppb to 19 ppb as temperature increased from 0°C to 10°C.

The regional temperature sensitivity ($Q_{10}$) of $NH_3$ emissions was estimated to be approximately
2.5, indicating that continued warming will likely enhance $NH_3$ volatilization. This may further
exacerbate nitrogen loss from agricultural systems and elevate $NH_3$ dry deposition to downwind
natural ecosystems, potentially intensifying ecological risks such as eutrophication and
biodiversity loss.

Figure S16 (*SI*) illustrates multi-year emission trends of $SO_2$, $NO_x$, and $NH_3$ derived from
multiple emission inventories, including EDGAR and MEIC; considering the potential
uncertainty of pollution emission inventories, the comparisons of different inventories can
provide robust results of emission trends. Although observed atmospheric $NH_3$ concentrations
have increased over the period 2013-2023, all inventories consistently indicate a slight decline
in $NH_3$ emissions. This apparent contradiction suggests that the observed rise in $NH_3$
concentrations may be primarily driven by reduced emissions of acidifying species—namely
$SO_2$ and $NO_x$—which typically enhance $NH_3$ partitioning into the particulate phase. The
reductions in $SO_2$ and $NO_x$ emissions may have suppressed their atmospheric reactions with
$NH_3$, thereby decreasing the formation of particulate ammonium and leaving a greater fraction
of $NH_3$ un-neutralized in the gas phase. This shift likely contributed to elevated ambient $NH_3$
concentrations, as reported in previous studies (Xu et al., 2019a; Liu et al., 2018; Liu et al.,
2017a).

We also investigated the temporal changes of agricultural fertilizer application and livestock
farming in China from 2013 to 2023, which are treated as the dominating source of $NH_3$
emissions in China (Figures S17-S18, *SI*). During the study period, the application rate of
agricultural fertilizers in China showed a trend of first increasing and then decreasing, reaching
a peak in 2015, and then continuing to decline until 2023. In order to reveal the changing
characteristics of different regions more clearly, we examined the change of agricultural
fertilizer amount in each region, and the results indicated that all regions showed a downward
trend. At the same time, the total amount of livestock breeding in China first decreased and then
rose during the same period.

Furthermore, it is important to note that, although satellite based observations from 2013 to 2023 reveal a clear upward trend in $NH_3$ concentrations at both column-averaged near surface level and ground-level, emission inventories from EDGAR, MEIC, and previous bottom-up estimates suggest that $NH_3$ emissions in China have stabilized or declined gradually in recent years (Liao et al., 2022; Zheng et al., 2018). This discrepancy is not only evident in the current study but has also been observed in other research, where some satellite-based $NH_3$ inversion studies show varying degrees of increasing trends (Zhang et al., 2017; Evangeliou et al., 2021; Luo et al., 2022). The difference may stem from the inherent contrasts between "bottom-up" and "top-down" estimation methods as displayed in Figure 13c. Several top-down studies indicate that the observed rise in $NH_3$ emissions could be partially explained by the neglect of $SO_2$ and $NO_x$ column concentration changes. For instance, Luo (2022) estimated global $NH_3$ emissions from 2008 to 2018 using a top-down approach and found that $NH_3$ emissions in eastern China increased by 61% per decade (6.6 Tg $a^{-1}$ per decade), particularly after 2013, driven primarily by the rise in IASI $NH_3$ column concentrations. However, when the model incorporated the decreasing $SO_2$ and $NO_x$ column concentrations, $NH_3$ emissions in eastern China were found to decrease by 19% per decade, with the decline becoming more pronounced after 2013 (28% per decade), aligning more closely with inventory results. This suggests that $SO_2$ and $NO_x$ concentrations play a significant role in mitigating atmospheric $NH_3$ levels. Additionally, both $SO_2$ and $NO_x$ emissions are negatively correlated with $NH_3$ concentrations to some extent (Deng et al., 2022). In summary, there are large differences in the estimation of $NH_3$ emissions by different methods, so it is necessary to further strengthen the comprehensive analysis and mutual verification of various methods (such as emission factor method, satellite observation inversion method and field observation method) to improve the accuracy and reliability of estimation results (Chen P et al., 2023).

According to EDGAR data, national $SO_2$ and $NO_x$ emissions declined by approximately 20.0% from 2013 to 2022, following the implementation of the Air Pollution Prevention and Control Action Plan in 2013, which led to substantial reductions in these precursor gases. It is important to note that our Random Forest model does not account for atmospheric chemical processes involving the formation and partitioning of secondary inorganic aerosols, such as nitrate ($NO_3^-$),

sulfate ($SO_4^{2-}$), and ammonium ($NH_4^+$). Therefore, for future investigations aiming to quantify
the role of atmospheric chemistry in modulating $NH_3$ concentrations and deposition, the use of
comprehensive atmospheric chemical transport models such as WRF-Chem or GEOS-Chem is
strongly recommended. These models are capable of resolving multiphase chemical reactions
and the thermodynamic partitioning of $NH_3$ into the aerosol phase, thereby offering a more
mechanistic understanding of $NH_3$ dynamics in response to co-emitted precursor changes.

To further elucidate the drivers of $NH_3$ dry deposition trends, we employed the method
described in Section 2.6.2 to decompose the relative contributions of changes in $NH_3$
concentrations and deposition velocities across different land cover types (Figure 10b; Table
S8, *SI*). All variables were normalized to facilitate comparison of relative contributions. The
results show that the change of $NH_3$ dry deposition was mainly driven by the change of
atmospheric $NH_3$ concentration, which accounted for 72.6%-81.2% of the total contribution in
China and four land cover types. Among them, the concentration changes in urban areas
contributed the least (72.6%), and the dry deposition rate change contributed the most (27.4%),
likely reflecting the more complex aerodynamic and surface resistance conditions in urban
environments. In contrast, forested areas showed the highest concentration-driven contribution
(81.2%), consistent with their relatively stable surface characteristics and low anthropogenic
disturbance.
To quantify the individual contribution from $SO_2$ and $NO_x$, we also applied the constructed RF
model with the method introduced in Section 2.6.1. Taking 2013 as the benchmark, the $SO_2$ and
$NO_x$ emissions in 2022 are simulated back to the level of 2013, and the results are normalized
to calculate the relative contribution. The results show that the contribution of $SO_2$ is 27.1%
and that of $NO_x$ is 72.9%. The contribution of $NO_x$ is significantly higher than that of $SO_2$,
which is closely related to the earlier start of $SO_2$ emission reduction. Long-term $SO_2$ emission
reduction has changed the composition of acid gases in the atmosphere, causing the relative
concentration of $NO_x$ to rise, gradually becoming the main acid gas reacting with $NH_3$ (Liu et
al., 2024d).
Considering the neutralization effect of $SO_2$ and $NO_x$ acid gases on $NH_3$, we analyzed the
changes of the three emissions (Table S9, *SI*). The data in Table S9 shows that the relative
annual reduction rates and total reduction rates of the three are similar, with values around 2.5%
and 20.5%. However, in terms of the average annual reduction, the reduction scale of $SO_2$ is
about 3 times that of $NH_3$, and that of $NO_x$ is about 2.4 times that of $NH_3$. Since the reduction
of $SO_2$ and $NO_x$ is larger, more $NH_3$ is distributed in the free state in the atmosphere. In addition,
$SO_2$ and $NO_x$, as acid gases, can react with $NH_3$ in the atmosphere, and they have a synergistic
effect in consuming $NH_3$. Therefore, although the relative annual reduction rates of the three
are similar, the contribution of acid gas as a whole to emission reduction is more significant.
From the perspective of chemical reaction measurement relationship, the equation for the
reaction between $SO_2$ and $NH_3$ to generate ammonium sulfate is: $2SO_2 + 4NH_3 + 2H_2O + O_2$
$\rightarrow 2\ (NH_4)\ _2SO_4$. In this reaction, 1 molecule of $SO_2$ can consume 2 molecules of $NH_3$; The
equation for the reaction between $NO_x$ and $NH_3$ to generate ammonium nitrate is: $NH_3 + HNO_3$
$\rightleftharpoons NH_4NO_3$. This reaction is a 1: 1 measurement relationship and is a reversible reaction. It will
re-decompose and release $NH_3$ under higher temperature or lower concentration conditions.
With the intensification of global warming, $NH_4NO_3$ in the atmosphere will also decompose
and release $NH_3$. Therefore, although the emissions of $SO_2$, $NO_x$ and $NH_3$ have all decreased
by about 20.5% from 2013 to 2025, the massive emission reduction of $SO_2$ and $NO_x$ has
weakened the consumption capacity of $NH_3$, resulting in a relative surplus of $NH_3$ that should
have been neutralized, causing $NH_3$ in the atmosphere. The concentration continues to rise, and
the increase of $NH_3$ concentration also promotes the increase of $NH_3$ dry deposition.
In summary, the observed increase in atmospheric $NH_3$ concentrations across China is largely
attributable to the substantial reductions in $SO_2$ and $NO_x$ emissions. Concurrently, changes in
$NH_3$ dry deposition fluxes are primarily driven by rising $NH_3$ concentrations, which are
indirectly influenced by declining $SO_2$ and $NO_x$ emissions. This inference is supported by
consistent evidence from both satellite and ground-based monitoring networks, which
document a marked decrease in $SO_2$ concentrations (Liu et al., 2019b; Xi et al., 2021), alongside
improvements in acid rain conditions.

Previous studies have indicated that optimizing fertilizer application and adjusting protein content in animal feed could potentially reduce $NH_3$ emissions by up to 30% without compromising agricultural yields or incurring additional costs (Zhang et al., 2020). In contrast, regulation of $NH_3$ emissions has lagged behind that of other pollutants. It was not until the implementation of the 2018 "Three-Year Action Plan for Winning the Blue-Sky Defense Battle" that agricultural $NH_3$ emissions were formally addressed. This plan emphasized enhanced recycling of livestock waste and measures to reduce $NH_3$ volatilization. Subsequently, the "14th Five-Year Plan for Energy Conservation and Emission Reduction" further targeted improvements in fertilizer and pesticide use efficiency, setting a goal to reduce $NH_3$ emissions from large-scale livestock operations in the Beijing-Tianjin-Hebei region by 5%. Although these recent policies have initiated efforts to mitigate $NH_3$ emissions, the rate of reduction remains substantially lower than that achieved for $SO_2$.

966

Furthermore, in the context of future warming, we analyzed projected emissions of $SO_2$, $NO_x$, and $NH_3$ under five SSP scenarios based on the Dynamic Projection Emission Coefficient (DPEC) inventory developed by Tsinghua University (Figure S19, *SI*). All scenarios indicate declining trends for these pollutants; however, $NH_3$ exhibits the smallest reduction, amounting to roughly two-thirds of the decreases projected for $SO_2$ and $NO_x$. This discrepancy, combined with rising temperatures and decreasing acid gas emissions, is expected to further enhance atmospheric $NH_3$ concentrations. Consequently, despite ongoing mitigation efforts targeting $NH_3$ emissions, the atmospheric $NH_3$ concentration may continue to increase. To counteract the synergistic effects of warming and reductions in acid-neutralizing pollutants, more stringent $NH_3$ emission control policies will be required in China over the coming decades to effectively stabilize or reduce atmospheric $NH_3$ concentrations.


**4 Comparison with previous studies and implications**

To evaluate and contextualize atmospheric $NH_3$ concentrations and dry deposition in China relative to other global regions and different land cover types, we conducted a comprehensive

literature review summarized in Table 4. This table integrates the findings of the present study
with previous assessments of atmospheric $NH_3$ levels and dry deposition fluxes worldwide. The
comparative analysis highlights considerable spatial variability, with $NH_3$ concentrations
ranging from approximately 2 to 10 ppb and area-specific dry deposition fluxes spanning 0.06
to 1.00 g m$^{-2}$ yr$^{-1}$. The values reported in this study are generally consistent with those
documented in comparable geographic and climatic regions.

This study estimates the national average $NH_3$ concentration in China at 4.98 ppb and the
corresponding dry deposition flux at 0.51 g m$^{-2}$ yr$^{-1}$, and the results for each province of China
were also displayed in Figure S20 (*SI*). The national average results closely align with those of
Liu et al. (2020a), who employed IASI satellite retrievals and reported $NH_3$ concentrations of
4.15 ppb and dry deposition fluxes of 0.58 g m$^{-2}$ yr$^{-1}$. The Tianjin megacity, Shandong province,
Henan province, Hebei province and Beijing megacity ranked as the largest top 5 regions for
$NH_3$ concentration and dry deposition flux, where Tianjin and Beijing are located within North
China Plain hotspots, and were largely influenced by atmospheric transport process from nearby
agricultural fields. Compared to Liu et al. (2020a), our analysis extends the observation period
and incorporates adjustments against ground-based monitoring data, thereby achieving higher
accuracy. Jia et al. (2016) estimated the global $NH_3$ dry deposition flux using empirical models
based on ground station measurements, reporting a value of 0.68 g m$^{-2}$ yr$^{-1}$ for China.

In contrast, Xu et al. (2015), utilizing averages from 43 ground stations (including 10 urban
stations, 22 rural stations and 11 background stations) from the National Nitrogen Deposition
Monitoring Network (NNDMN), reported substantially higher values for China (10.65 ppb and
1.00 g m$^{-2}$ yr$^{-1}$) than our study of spatial coverage of whole China. It can be explained by the
representation bias due to the predominance of monitoring sites in urban and rural (mostly
agriculture dominated) regions characterized by elevated $NH_3$ emissions and
underrepresentation of background locations, resulting in overestimation of national averages
when averaging these observation sites. Further evidence of spatial variability is provided by
Hu et al. (2020, 2021), who documented significant differences in $NH_3$ concentrations and
deposition rates between cropland and forested background sites, underscoring the critical
influence of land cover and emission sources on atmospheric $NH_3$ dynamics.

Overall, the synthesis of data summarized in Table 4 indicates that $NH_3$ concentrations in China
generally range from 4 to 10 ppb, with corresponding dry deposition fluxes between 0.5 and
1.0 g $m^{-2}$ $yr^{-1}$. The observed variability is primarily attributed to differences in observation
periods, measurement methodologies, and spatial coverage. By comparison, the United States
exhibits average $NH_3$ concentrations of approximately 2.65 ppb and dry deposition fluxes
ranging from 0.07 to 0.3 g $m^{-2}$ $yr^{-1}$, while Europe reports concentrations near 3.13 ppb and
deposition fluxes between 0.1 and 0.3 g $m^{-2}$ $yr^{-1}$. These findings highlight that both $NH_3$
concentrations and deposition fluxes in China are substantially higher than those reported for
the United States, Europe, and global averages. Notably, Europe has integrated $NH_3$ control
into its air pollution regulatory framework, resulting in measurable emission reductions in
recent years. This experience underscores the importance of implementing more stringent $NH_3$
mitigation policies in China to effectively address the ongoing increases in atmospheric $NH_3$
concentrations and dry deposition fluxes.

Previous studies have typically examined either atmospheric $NH_3$ concentrations or dry
deposition independently, with relatively few providing a comprehensive assessment
integrating both components. This study addresses this gap by combining satellite-based lower
ABL $NH_3$ concentrations with ground-based observations and utilizing the GEOS-Chem
atmospheric chemistry transport model in conjunction with a machine learning-based Random
Forest algorithm to simulate deposition velocities and fluxes. This integrated approach
facilitates the generation of high-resolution, multi-year estimates of $NH_3$ dry deposition across
China. The resulting dataset provides a robust scientific basis for improving national nitrogen
management policies and offers valuable insights into regional and global nitrogen cycling
processes.

**5 Conclusions**

This study presents a comprehensive analysis of the spatial distribution and temporal trends of atmospheric ammonia ($NH_3$) concentrations and dry deposition across China during 2013-2023. The key findings are as follows:

(1) The North China Plain exhibited persistently high $NH_3$ concentrations (>10 ppb), with significant annual increases in central and eastern regions (>0.4 ppb $yr^{-1}$). The largest seasonal increases occurred in summer (0.065 ppb $yr^{-1}$). $NH_3$ concentrations in 2023 were 13.8%-30.6% higher than in 2013 across all seasons. CrIS satellite retrievals were strongly correlated with in-situ measurements (R = 0.79), but are larger than the later by a factor of about two.

(2) The spatial pattern of $NH_3$ dry deposition revealed a pronounced east-west gradient, with the highest flux in the North China Plain and Sichuan Basin, and a significant upward trend along the eastern coast (>0.1 g $m^{-2}$ $yr^{-1}$). Over the 11-year period, $NH_3$ concentrations, deposition flux, and total deposition increased significantly in the land cover types of urban, cropland, and forest ecosystems. Urban areas showed the highest concentration and deposition flux as well as the fastest growth rates, while grasslands exhibited the largest total deposition.

(3) The national mean $NH_3$ concentration and dry deposition flux were estimated to be 4.98 ppb and 0.51 g $m^{-2}$ $yr^{-1}$, respectively. In addition, our analysis indicated that anthropogenic emissions were the dominant driver, accounting for approximately 77.4% of the variance in $NH_3$ concentrations, and meteorological conditions explained the remaining 22.6%; 72.6%-81.2% of trend for $NH_3$ dry deposition was governed by changes in $NH_3$ concentrations. These findings underscore the increasing $NH_3$ pollution across China and provide a critical scientific basis for informed nitrogen management within one of global largest $NH_3$ emission hotspots regions.

**Data Availability:** CrIS satellite retrievals of $NH_3$ were obtained from Environment and Climate Change Canada (ECCC) at https://hpfx.collab.science.gc.ca/~mas001/satellite_ext/cris/ (Shephard et al., 2015; 2020). Ground-based $NH_3$ measurements were sourced from Xu et al. (2019b), available at https://www.nature.com/articles/s41597-019-0061-2. $NH_3$ emission inventories were obtained from the Multi-resolution Emission Inventory for China (MEIC;

http://meicmodel.org.cn/?page_id=560), the Emissions Database for Global Atmospheric Research (EDGAR v8.1; https://edgar.jrc.ec.europa.eu/dataset_ap81#p3), and the Dynamic Projection model for Emissions in China (DPEC; http://meicmodel.org.cn/?page_id=1917). Emission data for $SO_2$ and $NO_x$ were derived from the Inversed Emission Inventory for Chinese Air Quality (CAQIEI; https://www.scidb.cn/en/detail?dataSetId=81cc0de9c68b4a4981e2f295ac612fbf), the Air Benefit and Cost and Attainment Assessment System (ABaCAS; https://abacas-dss.com/abacasChinese/Default.aspx), and the Community Emissions Data System (CEDS; https://github.com/JGCRI/CEDS/). The MEIC and EDGAR inventories were used for both $NH_3$ and $SO_2/NO_x$ emissions. Meteorological data were obtained from the ERA5 reanalysis dataset provided by the European Centre for Medium-Range Weather Forecasts (ECMWF) at https://cds.climate.copernicus.eu/datasets/reanalysis-era5-single-levels. The data of agricultural fertilizer application and livestock population are derived from the National Bureau of Statistics of China (https://www.stats.gov.cn/sj/ndsj/2024/indexch.htm). Agricultural zoning data were obtained from the Resource and Environmental Science Data Center (https://www.resdc.cn/Default.aspx), and land cover data were retrieved from the National Cryosphere Desert Data Center (https://www.ncdc.ac.cn/portal/metadata/9de270f3-b5ad-4e19-afc0-2531f3977f2f).

*Supplement.* The supplement related to this article is available online

**Declaration of Competing Interest**

The authors declare that they have no known competing financial interests or personal relationships that could have appeared to influence the work reported in this paper.

**Author contributions:** FS and CH conducted the data analysis and wrote the draft under supervision of CH, CH designed the study and revised this paper, JS and XL conducted GEOS-Chem modeling, all other co-authors collected supporting data, read and approved the final manuscript.

**Acknowledgments**

Cheng Hu is supported by the National Science founding of China (grant nos. 42475125, 42105117, 42021004 and 41975143), this work was also supported by the National Key R&D

Program of China (nos. 2019YFA0607202 and 2020YFA0607501); Jiangsu Science
Foundation for Distinguished Young Scholar (No. BK20220055); The 333 Project of Jiangsu
Province (No. BRA2017402); R&D Foundation of Jiangsu Province, China (No. BK20220020).
Cheng Hu also thanks the founding support from Key Laboratory of Ecosystem Carbon Source
and Sink, China Meteorological Administration (ECSS-CMA202403). We also Sincerely thank
the support from Environment and Climate Change Canada (ECCC) CrIS group.

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

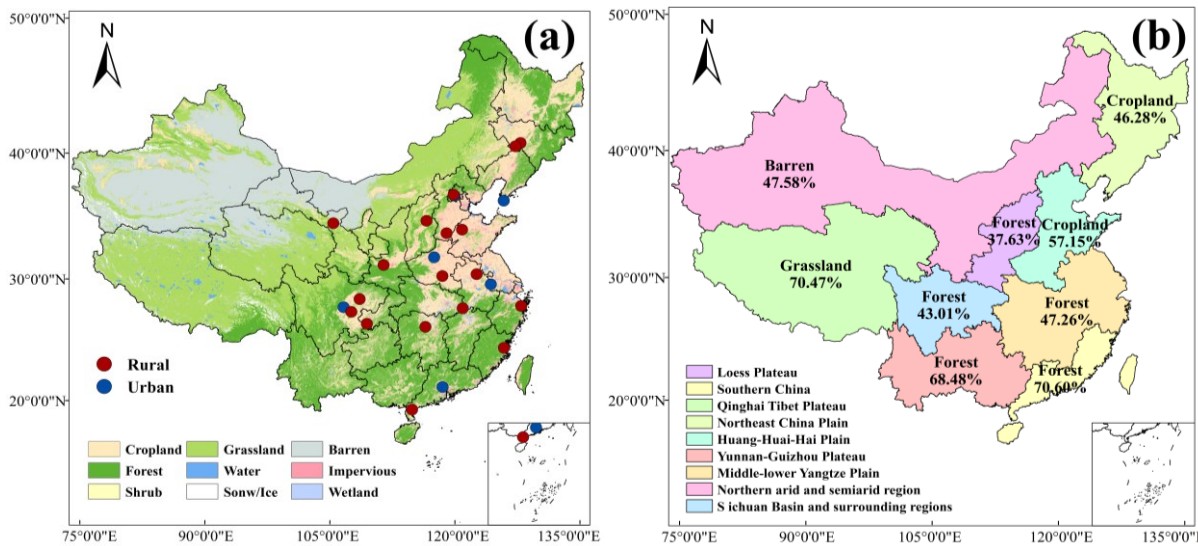

**Figure 1.** (a) Spatial distribution of land cover types and NH₃ monitoring sites in China in 2022, (b) classification of China into nine major agroecological zones based on agricultural practices and climatic conditions, note the percentage values represent area proportion of main land cover type (as list above) to total area in corresponding region.

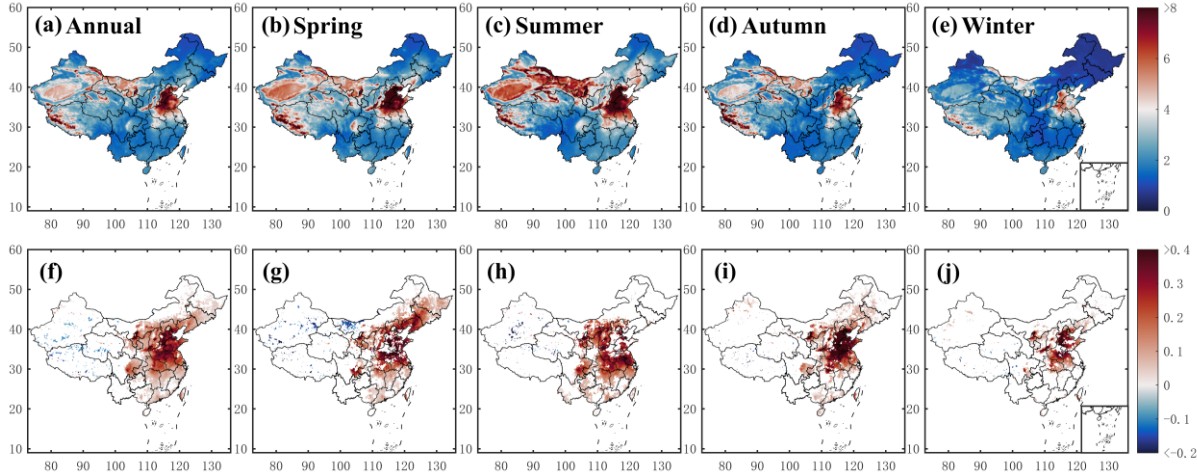

**Figure 2.** Spatial distribution of annual and seasonal averages of column-averaged NH$_3$ concentration from 2013 to 2023, (a) annual averages, (b) average in spring, (c) average in summer, (d) average in autumn, (e) average in winter; and trend of corresponding column-averaged NH$_3$ concentration from 2013 to 2023 for (f) annual averages, (g) average in spring, (h) average in summer, (i) average in autumn, (j) average in winter (Units: ppb for concentration; ppb yr$^{-1}$ for trend), note the white areas in the figure indicate trends that were not statistically significant at the 0.05 level.

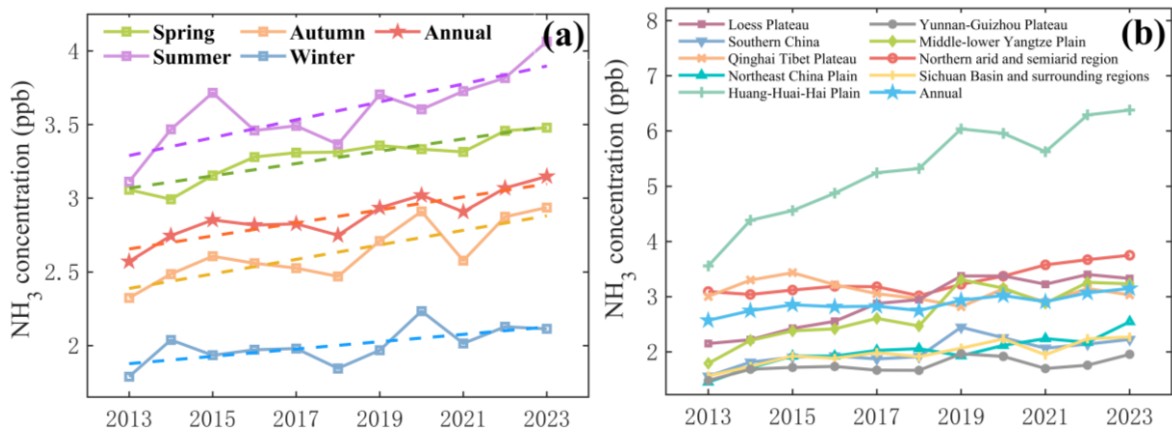

**Figure 3.** (a) Seasonal and (b) regional variations in CrIS satellite-based column-averaged (from ground to 1 km) NH$_3$ concentrations across China from 2013 to 2023 (Unit: ppb).

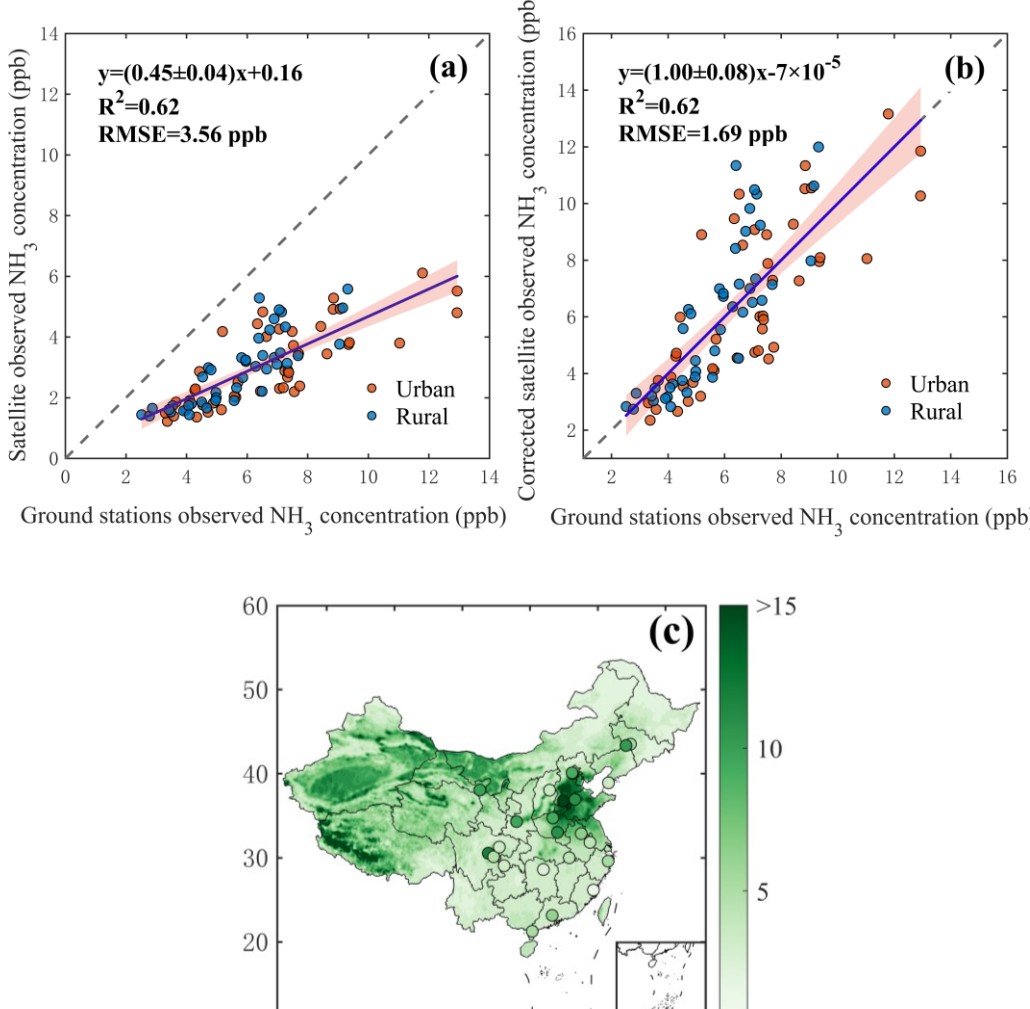

1464

1465

1466

**Figure 4.** (a) Comparison between CrIS satellite-based column average (from ground to ~1 km) NH₃ concentration and ground site based (~1.5 m) NH₃ observations before adjustment; (b) comparison between CrIS satellite-based column average NH₃ concentration and ground site based NH₃ observations after adjustment to ground-level; (c) Spatial distribution of adjusted satellite-based NH₃ concentration and comparisons with ground site based NH₃ concentrations in 2015 (Unit: ppb), note the adjustment from CrIS satellite-based column average (ground to ~1 km) to ground-level (~1.5 m) is conducted by using the linear regression equation derived from panel a, each scatter plot represents monthly averages of all available observations for either urban or rural site.


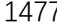



**Figure 5.** NH$_3$ dry deposition velocity in China in 2015: (a) GEOS-Chem simulation; (b) Random forest simulation (includes both validation set and training set); (c) Model difference (Unit: cm·s$^{-1}$)








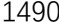

**Figure 6.** Spatial and regional trends in annual mean NH$_3$ dry deposition in China from 2013
to 2023: (a) spatial distribution of annual mean NH$_3$ dry deposition (Unit: g·m$^{-2}$); (b) temporal
trend of NH$_3$ dry deposition (Unit: g·m$^{-2}$·yr$^{-1}$); (c) interannual variation of NH$_3$ dry deposition
across different regions (Unit: g·m$^{-2}$).

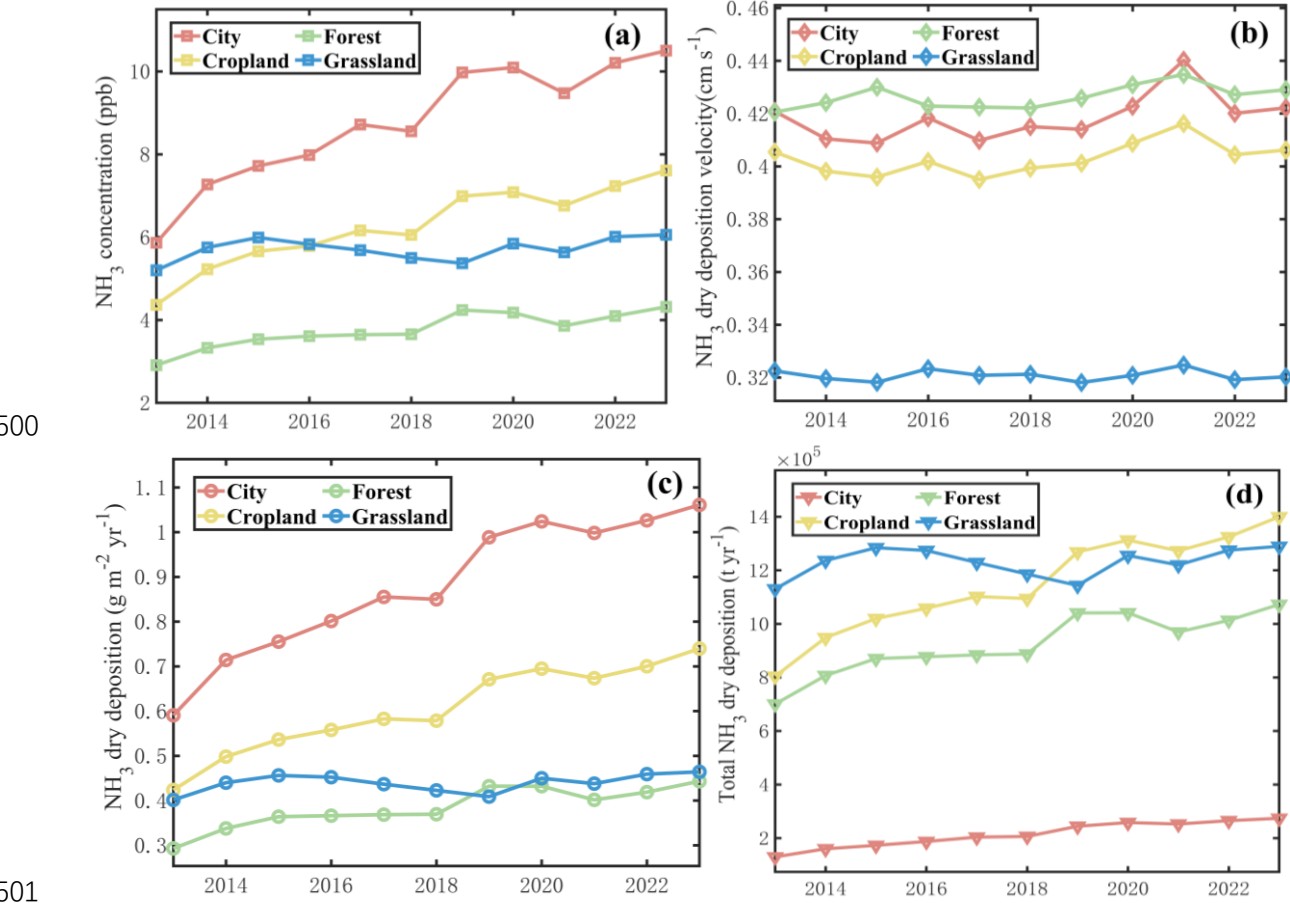


**Figure 7.** Trends in NH$_3$ concentration, dry deposition velocity, and dry deposition amount in China from 2013 to 2023: (a) trends in corrected NH$_3$ concentrations across different land surface types (Unit: ppb); (b) NH$_3$ dry deposition velocities over different land surface types (Unit: cm·s$^{-1}$); (c) trends in NH$_3$ dry deposition flux per unit area over different land surface types (Unit: g·m$^{-2}$); (d) interannual variation in annual NH$_3$ dry deposition over different land cover types (Unit: t yr$^{-1}$).

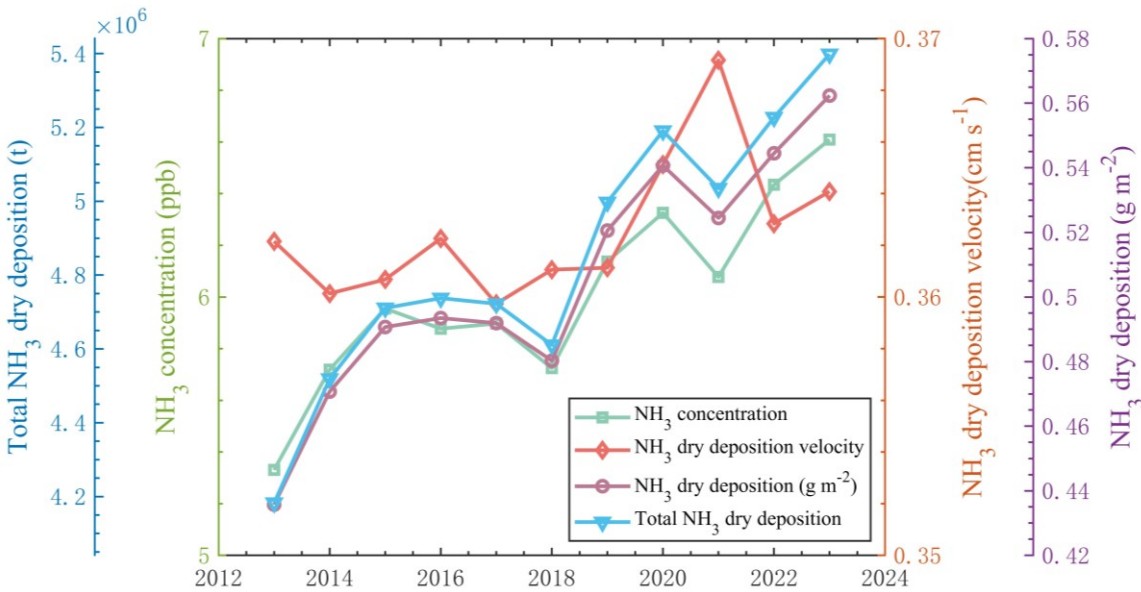


**Figure 8.** Annual changes in NH$_3$ concentration, dry deposition velocity, dry deposition flux

and total dry deposition for China from 2013 to 2023.













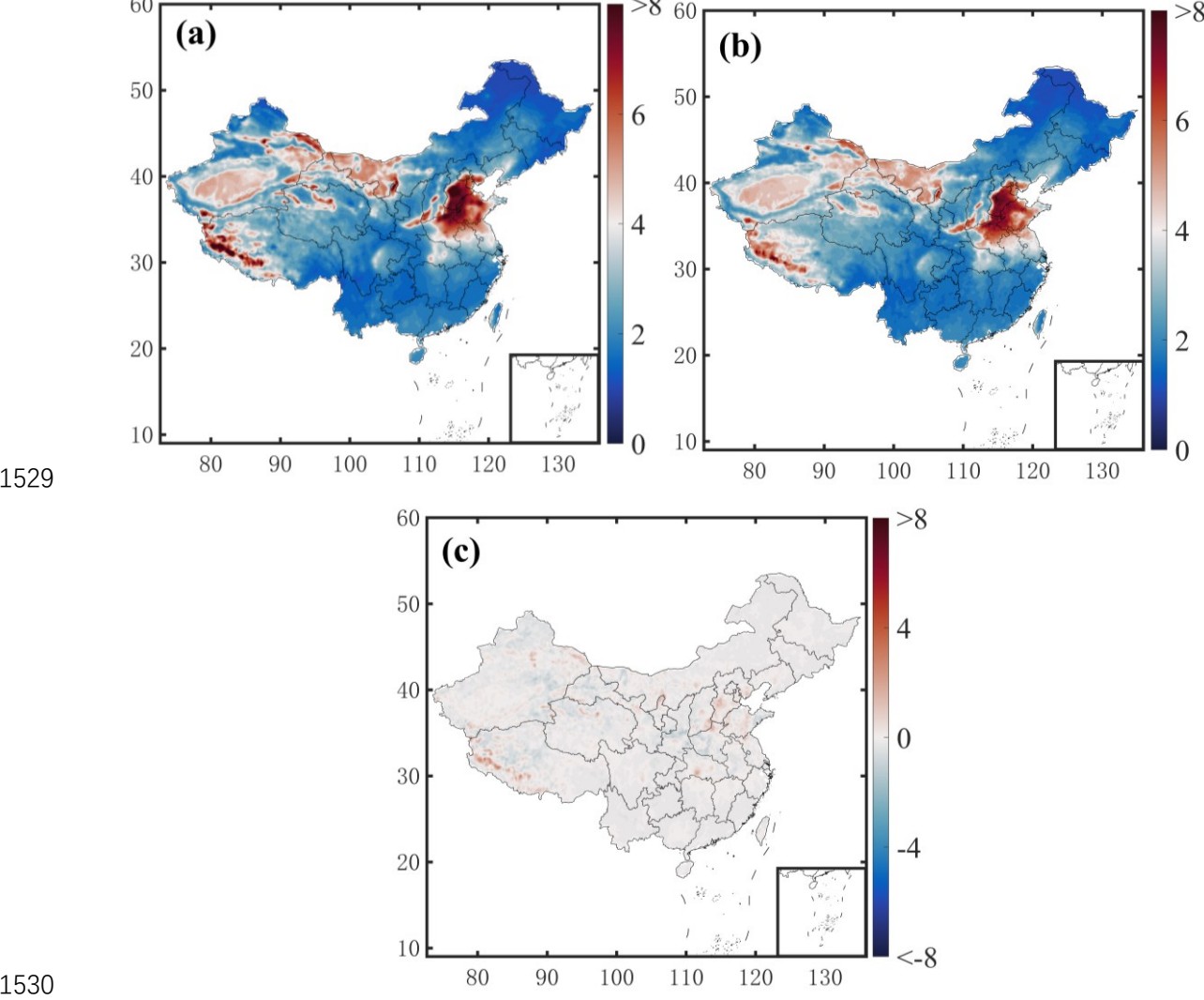



**Figure 9.** (a) Spatial distribution of adjusted ground-level $NH_3$ concentration for averages

between 2013 and 2023, (b) simulation of adjusted ground-level $NH_3$ concentration by RF

model for averages between 2013 and 2023, (c) difference between panel a and b, Units: ppb.

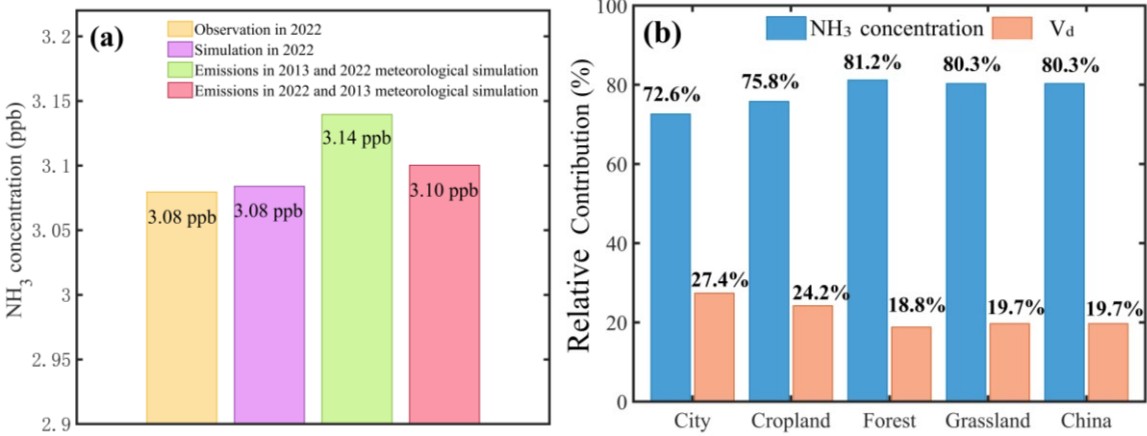

**Figure 10.** (a) Adjusted ground-level NH₃ concentrations and simulations by random forest models under different meteorological and emission scenarios in 2022; (b) Relative contribution of NH₃ concentration and dry deposition velocity to the dry deposition flux changes. Note: in panel a, the yellow bar represents the adjusted ground-level NH₃ concentration in 2022, the purple bar represents the random forest model simulated NH₃ concentration, the green bar represents the simulated NH₃ concentration using 2013 emissions and 2022 meteorological data, and the red bar represents the simulated NH₃ concentration using 2013 meteorological data and 2022 emissions data. And in panel b, the relative contributions of meteorological factors and emissions can be obtained by comparison with the difference in NH₃ concentration in the purple bar graph.)

**Table 1.** Annual and seasonal average $NH_3$ concentrations and their annual mean increment and
relative growth rate during entire study period.

| Season | $NH_3$ concentration (ppb) | Annual growth in $NH_3$ concentration (ppb yr$^{-1}$) | Relative annual growth rates (%) |
|---|---|---|---|
| Annual | 2.88 | 0.045 | 22.5 |
| Spring | 3.28 | 0.039 | 13.8 |
| Summer | 3.59 | 0.065 | 30.6 |
| Autumn | 2.63 | 0.050 | 26.4 |
| Winter | 2.00 | 0.023 | 18.1 |























**Table 2.** Average NH$_3$ concentration per unit area and annual mean increment and corrected

NH$_3$ concentration in the nine major agricultural regions of China from 2013 to 2023.

| Agricultural zoning | NH$_3$ concentration (ppb) | Annual growth in NH$_3$ concentration (ppb yr$^{-1}$) | Relative annual growth rates (%) | Corrected NH$_3$ concentration (ppb) |
|---|---|---|---|---|
| Huang-Huai-Hai Plain | 5.29 | 0.24 | 79.4 | 11.36 |
| Northern arid and semiarid region | 3.29 | 0.08 | 21.3 | 6.93 |
| Qinghai Tibet Plateau | 3.09 | -0.03 | 0.9 | 6.48 |
| Loess Plateau | 2.90 | 0.14 | 54.8 | 6.05 |
| Middle-lower Yangtze Plain | 2.70 | 0.13 | 80.5 | 5.62 |
| Southern China | 2.01 | 0.06 | 42.7 | 4.09 |
| Northeast China Plain | 2.01 | 0.08 | 75.1 | 4.09 |
| Sichuan Basin and surrounding regions | 1.98 | 0.06 | 45.1 | 4.02 |
| Yunnan-Guizhou Plateau | 1.75 | 0.03 | 31.9 | 3.52 |



**Table 3.** Average NH$_3$ dry deposition per unit area and annual mean increment in the nine major
agricultural regions of China from 2013 to 2023.

| Agricultural zoning | Dry deposition of NH$_3$ (g m$^{-2}$) | Annual growth of NH$_3$ dry deposition (g m$^{-2}$ yr$^{-1}$) |
|---|---|---|
| Huang-Huai-Hai Plain | 1.06 | 0.054 |
| Northern arid and semiarid region | 0.61 | 0.012 |
| Qinghai Tibet Plateau | 0.61 | -0.004 |
| Loess Plateau | 0.55 | 0.030 |
| Middle-lower Yangtze Plain | 0.52 | 0.034 |
| Southern China | 0.49 | 0.020 |
| Northeast China Plain | 0.39 | 0.018 |
| Sichuan Basin and surrounding regions | 0.38 | 0.014 |
| Yunnan-Guizhou Plateau | 0.38 | 0.008 |













**Table 4.** Comparison of global and regional NH$_3$ concentrations and dry deposition rates across
different studies. note: All results have been standardized to uniform units.

| Reference | Study period | Study region | NH$_3$ dry deposition (g m$^{-2}$ yr$^{-1}$) | | NH$_3$ concentration (ppb) | |
|---|---|---|---|---|---|---|
| This study | 2013-2023 | China | City | 0.88 | City | 8.76 |
| | | | Forest | 0.38 | Forest | 3.76 |
| | | | Cropland | 0.61 | Cropland | 6.27 |
| | | | Grassland | 0.44 | Grassland | 5.72 |
| | | | China | 0.51 | China | 4.98 |
| | | Global | 0.17 | | -- | |
| | | China | 0.58 | | Crop | 8.04 |
| | | | | | Urban | 6.86 |
| | | | | | Forest | 4.66 |
| | | | | | Grass | 3.10 |
| | | | | | Grass | 3.37 |
| | | | | | Mean | 4.15 |
| | | | | | Crop | 4.00 |
| Liu et al., 2020a | 2008-2016 | | | | Urban | 4.52 |
| | | Europe | 0.36 | | Forest | 3.32 |
| | | | | | Grass | 2.34 |
| | | | | | Grass | 1.87 |
| | | | | | Mean | 3.13 |
| | | | | | Crop | 4.38 |
| | | | | | Urban | 3.10 |
| | | US | 0.26 | | Forest | 2.51 |
| | | | | | Grass | 2.91 |
| | | | | | Grass | 1.87 |
| | | | | | Mean | 2.65 |
| Jia et al., 2016 | 2005-2014 | Asia (China) | 0.29 (0.68) | | -- | |
| | | North America | 0.042 (0.078) | | -- | |

| Reference | Period | Region | | | |
|---|---|---|---|---|---|
| | | (US) | | | |
| | | Europe | 0.11 | -- | |
| | | Africa | 0.32 | -- | |
| | | South America | 0.12 | -- | |
| | | Oceania | 0.037 | -- | |
| | | Global land | 0.18 | -- | |
| Kharol et al., 2018 | 2013 warm season (April-September) | North America | 0.06-1.22 | -- | |
| | | USA | 0.27 | -- | |
| | | Canada | 0.18 | -- | |
| Zhang et al. 2012 | 2006-2008 | US | 0.11 | -- | |
| Liu et al., 2019 | 2008-2016 | China | -- | 4.15 (0.39-22.90) | |
| | | Europe | -- | 3.14 (0.07-16.58) | |
| | | US | -- | 2.66 (0.24-18.52 ) | |
| Xu et al., 2015 | 2010-2014 | China | 1.00 (0.06-1.95) | 10.65 (0.52-22.89) | |
| Phillips et al., 2006 | 1999 Summer | North Carolina | 0.36 | -- | |
| Hu et al., 2020 | November 2017 | Tall-tower (100 m) observations in Minnesota | Forested lands 0.10-0.16 | 56 m | 6.76 |
| | | | Agricultural lands 0.41-0.62 | 100 m | 6.64 |
| Shao et al., 2019 | October - November 2018 | Nanjing | -- | 21.96±9.61 | |
| Hu et al., 2021 | 2017-2019 warm season | US Corn Belt | Forested lands 0.054±0.0054, 0.059±0.011, 0.059±0.011 | Forested lands 0.58±0.12, 0.71±0.14, 0.60±0.12 | |
| | | | Agricultural lands 0.77±0.16, 0.76±0.16, 0.77±0.16 | Agricultural lands 6.87±1.4, 6.76±1.4, 6.48±1.3 | |

| | | | LOTOS- | | |
|---|---|---|---|---|---|
| Van Der Graaf et al., 2018 | 2014 warm season | Europe | EUROS model | 0.21 | -- |
| | | | IASI | 0.27 | |






