# Peer review of "Decadal changes in atmospheric ammonia and dry deposition across China inferred from space-ground measurements and model simulations Fan Sun1, Yu Cui1,2, Jiayin Su3, Mark W. Shephard4, Shailesh K. Kharol4,5, Yifan Zhang1,2, Xuejing Shi1,2, Junging Zhang1,2, Huili Liu1,2, Qitao Xiao6, Xiao Lu3, Zhao-Cheng Zeng7, Timothy J. Griffis8, Cheng Hu1\* 1 College of Ecology and Environment, Joint Center for"

_EGUsphere, 2025_

## Author Comment (AC2)

**Reviewer #1:**

General comments: This manuscript addresses an important topic, spatial-temporal variability of atmospheric NH3 and its dry deposition across China, which has not been jointly studied before, especially for using CrIS in China, as far as I know. However, the main problem is that the logical connection between derived surface-level/near-surface NH3 concentrations and the derived NH3 dry-deposition fluxes is not sufficiently explained. In its current form, it is difficult to follow the storyline between concentrations and dry depositions and contains several conceptual and presentation problems in figures and tables that must be resolved before publication. The title ("one decade of satellite and ground-based observations") is misleading because the text does not make clear which data sources (RF-derived GEOS-Chem simulations, satellite, or ground obs) dominate the results and how they are linked. Suggest alternatives, something like, "Decadal changes in atmospheric ammonia and dry deposition across China inferred from space-ground measurements, and model simulations". The manuscript frequently mixes satellite, ground, reanalysis, and inventory products without a clear, reproducible workflow.

We sincerely thank the reviewer for the valuable suggestions and critical comments. We have considerably revised this MS based all comments, especially on strengthening the description of logical connection derived surface-level/near-surface NH3 concentrations and the derived NH3 dry-deposition fluxes. Besides, all points have been addressed below (review query in black; author response in blue). Changes to the text in the manuscript have been marked in blue.

For the above-mentioned main question of (1) "the logical connection between derived surface-level/near-surface NH3 concentrations and the derived NH3 dry-deposition fluxes is not sufficiently explained. In its current form, it is difficult to follow the storyline between concentrations and dry depositions and contains several conceptual and presentation problems in figures and tables that must be resolved before publication.". We have added more clarification and materials to support the reason and method to NH3 dry deposition calculation, which can be explained that the NH3 dry deposition should be calculated by multiplying dry deposition velocity and ground-level NH3 concentration (~1.5 m, the same height of site-based observations). Given that the lowest retrieval layer of satellite-based NH3 concentrations represents a column average from the ground to approximately 1 km (CrIS), and considering the large vertical gradient within the planetary boundary layer (PBLH), the column-averaged CrIS observations were adjusted to the ground-level.

We have revised the abstract on lines 35-58 as: "This study integrated 2013-2023 satellite-derived NH3 column concentrations from the Cross-track Infrared Sounder (CrIS) with adjustments from approximately five years ground in-situ ground observations to derive spatial-temporal variation in ground-level NH3 concentrations across China. We also used the GEOS-Chem transport model and a random forest algorithm by using emission inventories and reanalysis meteorological fields to simulate NH3 dry deposition velocity and fluxes, and explore the mechanisms driving observed trends. The CrIS observations results show that column-averaged (averages from ground to ~1 km) NH3 concentrations were the highest in the North China Plain (>10 ppb), with notable annual

and seasonal increasing trends. NH3 concentrations in 2023 were 13.8%-30.6% higher than in 2013. CrIS retrievals aligned well with in-situ data, though were generally about twice as high. After applying the regression equation between ground in-situ observations and CrIS column-averaged NH3 concentrations, we derive the spatial-temporal ground-level (1~1.5 m) NH3 concentrations and dry deposition fluxes from 2013 to 2023. The NH3 dry deposition fluxes exhibited a clear east-west gradient, with maxima in the North China Plain, and another hotpot region is also observed in the Sichuan Basin, southwestern China. Increases in ground-level NH3 concentrations and deposition were most pronounced in urban, cropland, and forest regions, with urban areas experiencing the fastest growth and grasslands the highest total deposition. The national mean ground-level NH3 concentration and dry deposition flux were 4.98 ppb and 0.51 g NH3 m-2 yr-1, respectively. Anthropogenic emissions explained 77.4% of the variability in ground-level NH3 concentration trend, and meteorological factors accounted for the remainder. Besides, 72.6%-81.2% of the NH3 dry deposition trend was governed by NH3 concentration changes. This study identifies the underlying cause of increasing ammonia pollution, which can be used to better inform nitrogen management strategies in China."

Besides, we added "Satellite observations provide wide spatial coverage and continuous temporal resolution, helping to fill spatial-temporal observation gaps by ground networks. Satellite-derived NH3 retrievals contain approximately 1 independent piece of information driven by peak sensitivity (averaging kernel) in the ABL (~1-3 km) (Shephard et al., 2011; Shephard et al., 2020) that can be represented as profiles with limited vertical resolution or integrated column-averaged values. Therefore, column-averaged satellite retrievals cannot directly replace ground-level (1~1.5 m) concentrations but provide complementary information that helps fill in monitoring gaps." on lines 140-147.

Added "In addition to these near surface ammonia concentration observations (from either in-situ surface or satellite observations), the dry deposition estimations also depend on deposition velocities (Lei et al., 2021; Liu S et al., 2024). Therefore, an alternative and reliable approach is to combine model simulated dry deposition, ground-level NH3 concentration from sites and satellite-based column-averaged observations, which can make full use of corresponding advantages and eliminate the large uncertainty from emission inventories of different pollution species." on lines 158-164.

Added and revised "In this study, the near surface level of CrIS-derived atmospheric NH3 retrieved profile concentrations was utilized, which are strongly correlated with ABL values around 900 hPa ( $\sim$ 1 km) and can represent column average NH3 concentration from ground to  $\sim$ 1 km. To avoid misunderstanding, we define near surface level in this study as the lowest level of CrIS-derived NH3 retrieved profile (average from ground to  $\sim$ 1km), and the ground-level as height of 1 $\sim$ 1.5 m, which is the typical height of site-based observations." on lines 210-215.

Added "As noted above, the calculation of NH3 dry deposition flux depends on ground-level NH3 concentrations, although tens of site-based NH3 concentration observations are available, they cannot provide long term

spatial-temporal resolved NH3 distributions especially in regions with high spatial heterogeneity within China. Therefore, we combined the advantage of ground-based NH3 observations of which can represent heights of 1~1.5 m, and satellite based spatial-temporal NH3 distributions. A linear relationship was constructed by comparing both datasets at the same location and period, where the regression equation was used to adjust the lower boundary layer satellite mixing ratio observations to ground-level of 1~1.5 m." on lines 261-268. And more detailed specific explanations are also added throughout this revised version as replied below.

For the question of "and contains several conceptual and presentation problems in figures and tables that must be resolved before publication", all comments regarding corresponding figures and tables have been revised or resolved as replied below in details.

For the comment of "The title ("one decade of satellite and ground-based observations") is misleading because the text does not make clear which data sources (RF-derived GEOS-Chem simulations, satellite, or ground obs) dominate the results and how they are linked. Suggest alternatives, something like, "Decadal changes in atmospheric ammonia and dry deposition across China inferred from space-ground measurements, and model simulations", we have changed the title of revised version as "Decadal changes in atmospheric ammonia and dry deposition across China inferred from space-ground measurements and model simulations" based on this suggestion.

For the comments of "The manuscript frequently mixes satellite, ground, reanalysis, and inventory products without a clear, reproducible workflow.", as replied above, the satellite based NH3 column averages and site-based ground observations are combined to analyze the spatial distribution of NH3 concentration and also used to calculate the dry deposition. Besides, the reanalysis, and inventory products will be used to simulate NH3 concentration and deposition and quantify contributions from different factors as emissions and meteorological fields. To make clarification and avoid misreading of satellite, ground, reanalysis, and inventory products on the workflow, we first have added corresponding explanation between satellite, ground on lines 35-47, 211-216 and 259-266. Such as "This study integrated 2013-2023 satellite-derived NH3 column concentrations from the Cross-track Infrared Sounder (CrIS) with adjustments from approximately five years ground in-situ ground observations to derive spatial-temporal variation in ground-level NH3 concentrations across China. We also used the GEOS-Chem transport model and a random forest algorithm by using emission inventories and reanalysis meteorological fields to simulate NH3 dry deposition velocity and fluxes, and explore the mechanisms driving observed trends. The CrIS observations results show that column-averaged (averages from ground to ~1 km) NH3 concentrations were the highest in the North China Plain (>10 ppb), with notable annual and seasonal increasing trends." More detailed explanations have also been added and revised throughout this revised version.

**1. Clarify the satellite and ground linkage and what is actually shown in Figs. 3-4**

Done as suggested, "the satellite and ground linkage" in this study is that the CrIS satellite-based NH3 observation

represent column average from ground to ~1 km within atmospheric boundary layer (ABL) and ground site-based observation represents NH3 concentration at around 1~1.5 m. They should display high consistency caused by regional emissions but with different magnitude, caused by the obvious vertical profiles of NH3 within ABL. Besides, because the NH3 dry deposition is calculated by multiplying dry deposition velocity and ground level NH3 concentration (~1.5 m, the same height of site-based observations), the column averaged CrIS observations should be calibrated to ground level. Most of the revisions regarding "Clarify the satellite and ground linkage" have been replied above, we have added "These studies demonstrate the utility of satellite retrievals in characterizing NH3 pollution and its spatiotemporal evolution, especially in regions lacking surface monitoring. In addition to these near surface ammonia concentration observations (from either in-situ surface or satellite observations), the dry deposition estimations also depend on deposition velocities (Lei et al., 2021; Liu S et al., 2024). Therefore, an alternative and reliable approach is to combine model simulated dry deposition, ground-level NH3 concentration from sites and satellite-based column-averaged observations, which can make full use of corresponding advantages and eliminate the large uncertainty from emission inventories of different pollution species." on lines 156-164. And added "In this study, the near surface level of CrIS-derived atmospheric NH3 retrieved profile concentrations was utilized, which are strongly correlated with ABL values around 900 hPa (~1 km) and can represent column average NH3 concentration from ground to ~1 km. To avoid misunderstanding, we define near surface level in this study as the lowest level of CrIS-derived NH3 retrieved profile (average from ground to ~1km), and the ground-level as height of 1~1.5 m, which is the typical height of site-based observations." on lines 210-215.

For the time series and spatial distribution of NH3 concentration in Figures 3 and 4, the Figure 3 displays CrIS satellite-based column-averaged (from ground to 1 km) NH3 concentrations across China; and Figure 4 displays comparisons between CrIS satellite-based column average NH3 concentration and ground site based NH3 observations. To make clarification, we revised the caption of Figure 3 as "Figure 3. (a) Seasonal and (b) regional variations in CrIS satellite-based column-averaged (from ground to 1 km) NH3 concentrations across China from 2013 to 2023 (Unit: ppb)." And revised the caption of Figure 4 as "Figure 4. (a) Comparison between CrIS satellite-based column average (from ground to ~1 km) NH3 concentration and ground site based (~1.5 m) NH3 observations before calibration; (b) comparison between CrIS satellite-based column average NH3 concentration and ground site based NH3 observations after calibration to ground level; (c) Spatial distribution of calibrated satellite-based NH3 concentration and comparisons with ground site based NH3 concentrations in 2015 (Unit: ppb), note the calibration from CrIS satellite-based column average (ground to ~1 km) to ground level (~1.5 m) is conducted by using the linear regression equation derived from panel a, each scatter plot represents monthly averages of all available observations for either urban or rural site."

The satellite product is described as a "near-surface column average at ~900 m" while ground sites measure at ~1 m. The rationale for using a regression to "correct" or calibrate the satellite is not justified, and Fig. 4b shows that the R2 does not improve after correction. If regression does not raise R2, explain why the regression is still preferred (e.g., reduces bias, corrects seasonal bias, etc.). If the vertical gradient between ~900 m and 1 m

is relatively constant, justify why a simple multiplicative (or additive) conversion factor was not used instead of a regression.

Thanks so much for this comment, as replied above, the reason to use linear regression equation is to calibrate the column averaged NH3 concentration to ground level of 1.5 m, and we can be further use them to derive NH3 dry deposition. Here the reason why  $R^2$  did not change is because that the same equation y=0.35+0.16 was used to all scatter plots which theoretically only change the RMSE and have no influence on  $R^2$ . The decrease of RMSE also indicate that this equation can obviously calibrate the column averaged NH3 concentration to ground level of 1.5 m. Here the conversion is x=(y-0.16)/0.45=2.22x-0.36, where y and x represent CrIS satellite-based column-averaged (from ground to 1 km) and the NH3 value after calibration to 1.5 m, respectively. This approach is also similar with using a simple multiplicative (or additive) conversion factor as mentioned in this comment.

To make clarification, we added "After correction, a new regression (Figure 4b) shows a nearly 1:1 agreement between satellite and ground-based measurements, with the RMSE reduced from 3.56 ppb to 1.69 ppb. The purpose of the linear regression equation is to adjust the column-averaged NH3 concentration to the ground-level at 1.5 m, as described in Section 2.2. This adjustment enables the derivation of NH3 dry deposition, which can then be compared with global observations. The reason that the R2 value remained unchanged is that the same equation, y=0.35+0.16, was applied to all scatter plots. This theoretically affects only the RMSE and does not influence the R2 value. The reduction in RMSE further indicates that this approach effectively adjusts the column-averaged NH3 concentration to the ground-level at 1.5 m. The conversion is given by x=(y-0.16)/0.45=2.22y-0.36, where y represents the CrIS satellite-based column-averaged NH3 concentration (from ground to 1 km), and x denotes the NH3 concentration after adjustment to 1.5 m. This approach is conceptually similar to using a simple multiplicative (or additive) conversion factor." on lines 608-620.

Explicitly state what the satellite product represents (column, layer height, vertical averaging kernel). If you intend to present surface-level NH3, then produce maps and time series of the surface concentration (satellite-derived and corrected by sites) in Sect. 3.1-3.2. If you still keep the near-surface average, explain plainly at the beginning of the results to describe the retrieval layer.

Done as suggested, the CrIS-derived NH3 vertical profile is divided into 15 levels through inversion, and the observation layer of CrIS-derived NH3 retrieved used in this study was the lowest layer and represent the column average from ground to around 900 hPa (~1km), which is defined as near surface layer and can better reflect the impact of human activities and natural source emissions on the near-Earth atmospheric environment. Satellite-derived NH3 retrievals contain approximately 1 independent piece of information driven by peak sensitivity (averaging kernel) in the boundary layer (~1-3 km) (Shephard et al., 2011; Shephard et al., 2020) that can be represented as profiles with limited vertical resolution or integrated column-averaged values. Therefore, column-averaged satellite retrievals cannot directly replace ground level (1.5 m) concentrations but provide complementary information that helps fill in monitoring gaps. By using the calibration from ground site

observations, the calibrated NH3 concentration can more represent concentration at ~1.5 m, which has also been defined as ground level to make it different with above near surface level. And the ground level NH3 concentration will be used to calculate dry deposition in China. Here in Section 3.1-3.2, this study first display the spatial-temporal patterns of the near surface NH3 concentration in China, which can be directly compared with previous studies using the same lowest layer. And in Section 3.3, we also want to display the comparisons between site-based concentration and near surface NH3 concentration, which will be further used to calibrate the near surface NH3 concentration to ground level.

To make clarification, we added "Satellite-derived NH3 retrievals contain approximately 1 independent piece of information driven by peak sensitivity (averaging kernel) in the ABL (~1-3 km) (Shephard et al., 2011; Shephard et al., 2020) that can be represented as profiles with limited vertical resolution or integrated column-averaged values. Therefore, column-averaged satellite retrievals cannot directly replace ground-level (1~1.5 m) concentrations but provide complementary information that helps fill in monitoring gaps." on lines 141-147. "This discrepancy can be attributed to the vertical gradient of NH3 in the atmosphere: ground-based sensors typically local point source observations operate at heights of 1-1.5 m, while satellite observations are regional (14 km) with low vertical resolution (~1km or more), which is shown from the averaging kernels (Shephard et al., 2011, Shephard et al., 2020)." on lines 592-595.

And more explanation at the beginning of the results section 3.1 as: "Using CrIS satellite-derived near surface NH3 concentrations (representing average between ground to  $\sim$ 1 km) from 2013 to 2023, a high-resolution ( $0.1^{\circ} \times 0.1^{\circ}$ ) monthly averaged NH3 concentration dataset across China over an 11-year period was generated. The observation from the near surface layer can reflect the impact of human activities and natural source emissions on the near-Earth atmospheric environment.". Added "In this section, we continue to present the spatiotemporal near-surface NH3 concentrations derived from CrIS lower ABL mixing ratio values." on lines 505-506. And revised the title of Section 3.1 as "3.1 Spatial patterns of near surface satellite NH3 concentration and its trend analysis", Section 3.2 as "3.2 Temporal variation of near surface satellite NH3 concentrations for different regions " and Section 3.3 as "3.3 Comparison between satellite and ground-based NH3 observations and adjustment from surface level to ground-level NH3 concentration".

2. Emission inventories: document, justify choices, and correct low-level mistakes

Done as suggested, the detailed comments have been addressed and replied below.

The manuscript references "six inventories", but it is unclear why different inventories were used for  $SO_2$ ,  $NO_x$ , and  $NH_3$ , and Text S3/Table S2 contains errors (institution names, versions, resolutions).

Done as suggested, the reason to use different inventories of SO2, NOx, and NH3 is based on the reason that many previous studies have concluded large potential bias in using single inventory caused by uncertainties from emission factors and activity data. Therefore, we make full use of all available inventories from different sources

to provide robust trends of emission changes. To make clarification, we added "The reason of using multiple emission inventories instead of only EDGAR is based on the fact that many previous studies have concluded large potential bias in using a single inventory caused by highly uncertain emission factors and activity data discrepancies. Therefore, we make full use of all available inventories from different data sources to provide robust evaluation of their emission changes." on lines 348-352.

Thanks so much for pointing out the typos, and regarding the errors in Text S3/Table S2 (institution name, version, resolution), we also double checked and modified the text and tables.

**Table S2.** Detailed information of 6 different emission inventories.

| Data   | Domain | Major institut          | Version          | Time period                                                      | Resolution | References               |
|--------|--------|-------------------------|------------------|------------------------------------------------------------------|------------|--------------------------|
| CAQIEI | China  | IAP                     | v1.0             | 2013-2020                                                        | 15 km      | Kong et al., 202         |
| MEIC   | China  | Tsinghua Uni
versity | v1.4             | 1990-2020                                                        | Provincial | Zheng et al., 20         |
| ABaCAS | China  | Tsinghua Uni
versity | v2.0             | 2005-2021                                                        | Provincial | Li et al., 2023          |
| CEDS   | Global | JGCRI                   | v_2021_
02_05 | 1970-2019                                                        | 0.5°       | McDuffie et al.,
2020 |
| EDGAR  | Global | JRC                     | v8.1             | 1970-2022                                                        | 0.1°       | Crippa et al., 20        |
| DPEC   | China  | Tsinghua Uni
versity | v1.2             | 2020; 2025; 2
030; 2035; 20
40; 2045; 205
0; 2055; 2060 | Provincial | Cheng et al., 20
23   |

The descriptions of these inventories have been added and revised in Text S3 (SI).

The Inversed Emission Inventory for Chinese Air Quality (CAQIEI), jointly developed by the Institute of Atmospheric Physics, Chinese Academy of Sciences (IAP, CAS), and the China National Environmental Monitoring Center (CNEMC), is a top-down long-term emission inventory for China. It provides emissions data for multiple air pollutants—including SO2 and NOx—from 2013 to 2020, with a horizontal resolution of 15 km. CAQIEI has been shown to effectively reduce biases in prior emission inventories (Kong et al., 2023).

The Multi-resolution Emission Inventory for China (MEIC), developed by Tsinghua University, is a bottom-up

emission inventory model that covers the period from 1990 to 2020. It offers spatially resolved emission data at provincial scale. In this study, provincial-level data from MEIC were utilized to calculate and analyze long-term emission trends of SO2, NOx, and NH3 across China (Zheng et al., 2018).

The Air Benefit and Cost and Attainment Assessment System - Emission Inventory version 2.0 (ABaCAS), co-developed by Tsinghua University, South China University of Technology, and other institutions, is a decision-support system for cost-effectiveness evaluation of air pollution control and attainment planning. The dataset spans from 2005 and has been updated through 2021, with spatial resolutions including provincial scale (Li et al., 2023).

The Community Emissions Data System (CEDS) is a global emission inventory that provides gridded emissions of various gases and aerosol precursors—including  $CO_2$ ,  $CH_4$ ,  $NO_x$ , and  $SO_2$ —with spatial resolutions of  $0.5^{\circ}$ . The data set spans from 1970 to 2019, and most of the data after 1950 are the result of extensive coordination and processing (McDuffie et al., 2020).

The Emissions Database for Global Atmospheric Research (EDGAR), developed by the Joint Research Centre (JRC) of the European Union, provides global gridded emissions at a 0.1° resolution. The temporal resolution is from 1970 to 2022, and this study uses annual grid data to analyze the emission change trend of SO2, NOx, and NH3 (Crippa et al., 2024).

The Dynamic Projection model for Emissions in China (DPEC), developed by Tsinghua University, is a forward-looking model that projects China's future emissions under multiple scenarios. The current version of the DPEC dataset (v1.2) includes five policy scenarios: early peak-net zero-clean air, on-time peak-net zero-clean air, on-time peak-clean air, clean air, and baseline. The spatial resolution is consistent with that of the MEIC inventory (Cheng et al., 2023).

Add a table listing all inventories used with: name, publisher/institution, version/year, spatial resolution, temporal resolution, main purpose, and how each inventory was used in your study.

Done as suggested, we have added the following table in *SI* materials, which shows the emission inventory used name, institution, version, spatial resolution, temporal resolution, main purpose, and how each inventory was used in our study.

Table S5. Details of 6 different emission inventories and their usage purposes.

| Data   | Major institu | Version | Time period | Resolution | Main purpose                    |
|--------|---------------|---------|-------------|------------|---------------------------------|
| CAQIEI | IAP, CAS      | v1.0    | 2013-2020   | 15 km      | The multi-year emission changes |

| ABaCAS       | Tsinghua Un iversity | v2.0                                         | 2005-2021     | Provincial | of $SO_2$ and $NO_x$ are plotted, w hich is used to analyze the chan |
|--------------|----------------------|----------------------------------------------|---------------|------------|----------------------------------------------------------------------|
|              |                      |                                              |               |            | ge trend of SO 2 and NO x emissi               |
| CEDS         | JGCRI                | v_2021_0                                     | 1970-2019     | 0.5°       | ons, and provide data support fo                                     |
| 0220         | 2_05                 | r the analysis of NH 3 concentrat |               |            |                                                                      |
|              |                      |                                              |               |            | ion change trend                                                     |
| EDGAR        | JRC                  | v8.1                                         | 1970-2022     | 0.1°       | The multi-year emission changes                                      |
|              |                      |                                              |               |            | of SO 2 , NO x and NH 3 are plott   |
|              |                      |                                              |               |            | ed, which is used to analyze the                                     |
| MELG         | Tsinghua Un          |                                              | 4000 0000     |            | emission trend of SO 2 , NO x an               |
| MEIC         | iversity             | v1.4                                         | 1990-2020     | Provincial | d NH 3 , and provide data support                         |
|              |                      |                                              |               |            | for the analysis of NH 3 concen                           |
|              |                      |                                              |               |            | tration trend                                                        |
|              |                      |                                              |               |            | Draw the temporal trends of SO 2 ,                        |
|              |                      |                                              | 2020; 2025; 2 |            | NO x and NH 3 from 2020 to 2026,               |
| DDE G | Tsinghua Un          | 1.0                                          | 030; 2035; 20 | 5   | and analyze the future emission                                      |
| DPEC         | iversity             | v1.2                                         | 40; 2045; 205 | Provincial | trends to provide theoretical                                        |
|              |                      |                                              | 0; 2055; 2060 |            | support for the possible future                                      |
|              |                      |                                              |               |            | trend of NH 3 concentration                               |

Explain why different inventories were selected for different species. If possible, use a consistent set of inventories for cross-species comparison, or present a justification for why species-specific choices are necessary.

Done as suggested, as replied above, in order to analyze the influencing factors of NH3 concentration change trend, we adopted a variety of NH3, NOx and SO2 emission inventories. And the reason to use different inventories of SO2, NOx, and NH3 is based on the reason that many previous studies have concluded large potential bias in using single inventories caused by emission factors and activity data. Therefore, we make full use of all available inventories from different sources to provide robust trends of emission changes. To make clarification, we added "The reason of using multiple emission inventories instead of only EDGAR is based on the fact that many previous studies have concluded large potential bias in using a single inventory caused by highly uncertain emission factors and activity data discrepancies. Therefore, we make full use of all available inventories from different data sources to provide robust evaluation of their emission changes." on lines 348-352. Among them, five inventories, CAQIEI, MEIC, ABaCAS, CEDS and EDGAR, were used for NOx and SO2 to analyze their emission changes over the years. However, due to the relatively late start of NH3 research, some inventories do not include NH3 emission data. Therefore, we finally selected representative EDGAR and MEIC inventories globally and in China for special analysis of NH3 emissions.

State whether biomass burning emissions were included in the simulations and, if so, which dataset was used. If biomass burning is excluded, provide justification.

The EDGAR inventory was used in this study to simulate spatial-temporal patterns of NH3 concentration, where there is not biomass burning in EDGAR. However, we extracted emissions from biomass burning from the MEIC inventory for 2013-2020. The following table shows the total emissions of SO2, NOx, and NH3 during this period, as well as the average annual emissions and their proportions from biomass burning. The proportion of these biomass burning among the three gases is less than 3%.

To make clarification, we also added "Note the emissions from EDGAR will be used in this study to simulate spatial-temporal patterns of NH3 concentration. Note the EDGAR does not include biomass burning. However, we also extracted emissions from biomass burning from the MEIC inventory for 2013-2020, the total emissions of SO2, NOx, and NH3 during this period in China, as well as the average annual emissions and their proportions from biomass burning were displayed in Table S6 (*SI*). And the contribution of biomass burning to these three gases was less than 3%, indicating relatively small influence of biomass burning in simulating NH3 concentrations." on lines 366-373.

**Table S6.** Total emissions of SO2, NOx, and NH3 during this period in China, as well as the average annual emissions and their proportions from biomass burning, note data is from the MEIC emission inventory during 2013-2020.

| Gas             | biomass combustion emissions (t yr -1 ) | Total emissions (t yr -1 ) | Proportion |
|-----------------|----------------------------------------------------|---------------------------------------|------------|
| SO 2 | 27.5                                               | 14499.5                               | 0.19%      |
| $NO_x$          | 327.3                                              | 22845.1                               | 1.4%       |
| NH 3 | 254.3                                              | 9917.9                                | 2.6%       |

**3. Random Forest (RF) applications and predictor consistency**

The RF is used for two distinct purposes: (A) to extend/estimate dry deposition velocity (Vd) across 2013-2023 from 2015 simulations, and (B) to identify key drivers of atmospheric NH3. The methods (Sect. 2.4.2) only describe the prior usage incompletely, and there are inconsistent predictor sources (ERA5 used for RF, MERRA-2 used for GEOS-Chem). Fig. 10 and its description are confusing (panel a vs b; emissions vs deposition drivers).

As was mentioned in this comment, the RF will be used for two purpose, with (1) the first purpose of simulating dry deposition velocity ( $V_d$ ) across 2013-2023 from 2015 simulations, which was displayed in Section 2.4.2; and (2) the second purpose is to simulate NH3 concentration and identify key drivers of atmospheric NH3 changes as illustrated in Section 2.6.1. To make the purposes of using RF model clearer, we have added "Overall, the RF

model will be used for two purpose, with (1) the first purpose of simulating dry deposition velocity (Vd) across 2013-2023, which is displayed in this Section; and (2) the second purpose is to simulate NH3 concentration and identify key drivers of atmospheric NH3 changes as illustrated in Section 2.6.1" on lines 311-314. And also revised the sentence on lines 285-288 as "a random forest machine learning algorithm was also applied to simulate dry deposition velocities from 2013 to 2023 based on output from GEOS-Chem model (see more details in Section 2.4), where the spatial resolution can improve to 0.25°, see more details in Section 2.4.".

Regarding the comment of inconsistent predictor sources of using ERA5 in RF model and MERRA-2 in the GEOS-Chem model, it's because MERRA-2 is a widely used reanalysis data for GEOS-Chem model and with spatial resolution of only  $0.5^{\circ} \times 0.625^{\circ}$ . However, in our study, considering the higher spatial resolution of emissions and CH4 concentration of 0.1°, we prefer to make full use this finer spatial resolution and we chose ERA5 data with has much higher spatial resolution of 0.25°. To make clarification, we have revised and added "The most widely used approach to derive  $V_d$  is by model simulation. Here we first used the GEOS-Chem chemical transport model to simulate spatial-temporal varied Vd across China in 2015, with spatial resolution of  $0.5^{\circ} \times 0.625^{\circ}$  at hourly scale. However, considering (1) the spatial resolution of  $0.5^{\circ} \times 0.625^{\circ}$  will lead to aggregation errors when quantifying NH3 concentration and dry deposition from different land cover types within the same grid cell, and (2) the GEOS-Chem model requires substantial computational resources for one decade, and to further improve spatial resolution and computational efficiency (Figure S2, SI), a random forest machine learning algorithm was also applied to simulate dry deposition velocities from 2013 to 2023 based on output from GEOS-Chem model (see more details in Section 2.4), where the spatial resolution can improve to 0.25°, see more details in Section 2.4."; and "This approach allowed us to extend the simulation to the full 2013-2023 period, while improving both computational efficiency and spatial resolution from 0.5° × 0.625° to 0.25° × 0.25°." on lines 641-643.

The modifications to Figure 10 and its description are detailed in the response to the next comment.

Revise Fig. 10 and its caption. Make it explicit what each panel displays. If panels show different metrics (contribution to concentration vs contribution to deposition), label and discuss them separately.

Done as suggested, we have modified the caption of Figure 10 and added more descriptions of Figure 10.

We also discussed Figure 10a and 10b separately, that for Figure 10a, we added and revised the paragraph as "To quantify the contribution of emissions and meteorological factors to changes in NH3 concentrations, we used a random forest model to simulate NH3 concentration with different sensitivity test by replacing single factor, and the difference between them can be treated as contributions from corresponding factor. Figure 10a shows the adjusted ground-level NH3 concentration in 2022 and the simulation results under three different meteorological and emission scenarios. The simulated concentrations are 3.08 ppb, 3.14 ppb, 3.10 ppb. Both meteorological and emission contributions are calculated from the simulation results. Simulation results from the random forest model showed that anthropogenic emissions were the main driver, accounting for approximately 77.4% of the NH3

concentration changes, while meteorological conditions accounted for the remaining 22.6% (Figure 10a)." on lines 780-789.

For Figure 10b, we added and revised the paragraph on lines 863-874 as "To further elucidate the drivers of NH3 dry deposition trends, we employed the method (illustrated in Section 2.6.2) to decompose the relative contributions of changes in NH3 concentrations and deposition velocities across different land cover types (Figure 10b; Table S4, *SI*). All variables were normalized to facilitate comparison of relative contributions. The results show that the change of NH3 dry deposition is mainly driven by the change of atmospheric NH3 concentration, which accounts for 72.6%-81.2% of the total contribution in China and four land cover types. Among them, the concentration changes in urban area contributed the least (72.6%), and the dry deposition rate change contributed the most (27.4%), likely reflecting the more complex aerodynamic and surface resistance conditions in urban environments. In contrast, forested areas showed the highest concentration-driven contribution (81.2%), consistent with their relatively stable surface characteristics and low anthropogenic disturbance.

Added "To quantify the individual contribution from SO2 and NOx, we also applied the constructed RF model with the method introduced in Section 2.6.1. Taking 2013 as the benchmark, the SO2 and NOx emissions in 2022 are simulated back to the level of 2013, and the results are normalized to calculate the relative contribution. The results show that the contribution of SO2 is 27.1% and that of NOx is 72.9%. The contribution of NOx is significantly higher than that of SO2, which is closely related to the earlier start of SO2 emission reduction. Long-term SO2 emission reduction has changed the composition of acid gases in the atmosphere, causing the relative concentration of NOx to rise, gradually becoming the main acid gas reacting with NH3 (Liu S et al., 2024).

Considering the neutralization effect of SO2 and NOx acid gases on NH3, we analyzed the changes of the three emissions (Table S9, *SI*). The data in Table S9 shows that the relative annual reduction rates and total reduction rates of the three are similar, with values around 2.5% and 20.5%. However, in terms of the average annual reduction, the reduction scale of SO2 is about 3 times that of NH3, and that of NOx is about 2.4 times that of NH3. Since the reduction of SO2 and NOx is larger, more NH3 is distributed in the free state in the atmosphere. In addition, SO2 and NOx, as acid gases, can react with NH3 in the atmosphere, and they have a synergistic effect in consuming NH3. Therefore, although the relative annual reduction rates of the three are similar, the contribution of acid gas as a whole to emission reduction is more significant." on lines 875-892.

Figure 10. (a) NH3 concentrations observed by satellite and simulated by random forest models under different meteorological and emission scenarios in 2022; (b) Relative contribution of NH3 concentration and dry deposition velocity to the dry deposition flux changes. Note: in panel a, the yellow bar represents the satellite observed NH3 concentration in 2022, the purple bar represents the random forest model simulated NH3 concentration, the green bar represents the simulated NH3 concentration using 2013 emissions and 2022 meteorological data, and the red bar represents the simulated NH3 concentration using 2013 meteorological data and 2022 emissions data. And in panel b, the relative contributions of meteorological factors and emissions can be obtained by comparison with the difference in NH3 concentration in the purple bar graph.)

If  $SO_2$  and  $NO_x$  are included as predictors, present their individual contributions (don't lump them into "anthropogenic emissions" only).

Done as suggested, we calculated the relative contributions of SO2 and NOx separately.

We applied the constructed RF model with the method as in Section 2.6.1 is adopted. Taking 2013 as the benchmark, the SO2 and NOx emissions in 2022 are simulated back to the level of 2013, and the results are normalized to calculate the relative contribution of the them. The results show that the contribution of SO2 is 27.1% and that of NOx is 72.9%. The contribution of NOx is significantly higher than that of SO2, which is closely related to the earlier start of SO2 emission reduction. Long-term SO2 emission reduction has changed the composition of acid gases in the atmosphere, causing the relative concentration of NOx to rise, gradually becoming the main acid gas reacting with NH3 (Liu S et al., 2024).

To make clarification, we have added "To quantify the individual contribution from SO2 and NOx, we also applied the constructed RF model with the method introduced in Section 2.6.1. Taking 2013 as the benchmark, the SO2 and NOx emissions in 2022 are simulated back to the level of 2013, and the results are normalized to calculate the relative contribution. The results show that the contribution of SO2 is 27.1% and that of NOx is 72.9%. The contribution of NOx is significantly higher than that of SO2, which is closely related to the earlier start of SO2 emission reduction. Long-term SO2 emission reduction has changed the composition of acid gases in the

atmosphere, causing the relative concentration of  $NO_x$  to rise, gradually becoming the main acid gas reacting with  $NH_3$  (Liu S et al., 2024).

Considering the neutralization effect of SO2 and NOx acid gases on NH3, we analyzed the changes of the three emissions (Table S9, SI). The data in Table S9 shows that the relative annual reduction rates and total reduction rates of the three are similar, with values around 2.5% and 20.5%. However, in terms of the average annual reduction, the reduction scale of SO2 is about 3 times that of NH3, and that of NOx is about 2.4 times that of NH3. Since the reduction of SO2 and NOx is larger, more NH3 is distributed in the free state in the atmosphere. In addition, SO2 and NOx, as acid gases, can react with NH3 in the atmosphere, and they have a synergistic effect in consuming NH3. Therefore, although the relative annual reduction rates of the three are similar, the contribution of acid gas as a whole to emission reduction is more significant.

From the perspective of chemical reaction measurement relationship, the equation for the reaction between  $SO_2$  and  $NH_3$  to generate ammonium sulfate is:  $2SO_2 + 4NH_3 + 2H_2O + O_2 \rightarrow 2$  ( $NH_4$ )  $_2SO_4$ . In this reaction, 1 molecule of  $SO_2$  can consume 2 molecules of  $NH_3$ ; The equation for the reaction between  $NO_x$  and  $NH_3$  to generate ammonium nitrate is:  $NH_3 + HNO_3 \rightleftharpoons NH_4NO_3$ . This reaction is a 1: 1 measurement relationship and is a reversible reaction. It will re-decompose and release  $NH_3$  under higher temperature or lower concentration conditions. With the intensification of global warming,  $NH_4NO_3$  in the atmosphere will also decompose and release  $NH_3$ . Therefore, although the emissions of  $SO_2$ ,  $NO_x$  and  $NH_3$  have all decreased by about 20.5% from 2013 to 2025, the massive emission reduction of  $SO_2$  and  $NO_x$  has weakened the consumption capacity of  $NH_3$ , resulting in a relative surplus of  $NH_3$  that should have been neutralized, causing  $NH_3$  in the atmosphere. The concentration continues to rise, and the increase of  $NH_3$  concentration also promotes the increase of  $NH_3$  dry deposition." on lines 875-904.

For RF validation: show diagnostics (train/test split) and present performance metrics separately for validation. For spatial maps (e.g., Fig. 5b for RF-predicted Vd in 2015) indicate whether values shown include both training and validation pixels; better: show a validation map or a scatter of observed vs predicted Vd for the validation set.

Done as suggested, we added the comparisons by using train and test dataset, which has been described as "The dataset was randomly split into a training set (60%) and a validation set (40%), the comparisons between using two approaches will be evaluated in Section 3.4.1.". The comparisons are displayed below, the scatter plots are generally around 1:1 line, and the R2 values are 0.93 and 0.83, respectively, indicating that the simulation results of the two models have good agreement. We revised the caption of Figure 5 as "Figure 5. NH3 dry deposition velocity in China in 2015: (a) GEOS-Chem simulation; (b) Random forest simulation (includes validation set and training set); (c) Model difference (Unit: cm·s-1)", and also added Figure S12 in SI material.

**Figure S12.** Scatter density maps of dry deposition rates simulated by GEOS-Chem model and random forest model (Unit: cm·s-1); (a) test set; (b) training set

**4. Trend analysis and how representative the 24 sites + satellite decade are**

The trend analysis relies on 24 ground sites and 11 years of satellite data. Few ground sites have >10 years of continuous records (as mentioned in the Introduction); this could bias trend estimates.

Here the ground site NH3 observation data are collected from different monitoring departments, influenced by data availability of ground sites based NH3 concentrations, it's hard to get observation data for more than 10 years for these sites. Therefore, it's the reason why we combined both ground based NH3 observations with satellite based spatial-temporal NH3 distributions. Overall, the trend analysis is mainly influenced by temporal variations of CrIS-derived NH3 concentration, with calibration from ground site based observations to ground level. Most of the reasons have been replied above together with detailed revisions.

We have also added more clarification as "As noted above, the calculation of NH3 dry deposition flux depends on ground-level NH3 concentrations, although tens of site-based NH3 concentration observations are available, they cannot provide long term spatial-temporal resolved NH3 distributions especially in regions with high spatial heterogeneity within China. Therefore, we combined the advantage of ground-based NH3 observations of which can represent heights of 1~1.5 m, and satellite based spatial-temporal NH3 distributions. A linear relationship was constructed by comparing both datasets at the same location and period, where the regression equation was used to adjust the lower boundary layer satellite mixing ratio observations to ground-level of 1~1.5 m." on lines 261-268.

Provide a clear description of the temporal coverage at each of the 24 sites, or justify the site selection procedure

Done as suggested, as recommended, we have added detailed information for 24 sites to the *SI* file, as shown in the table below. The following table shows the information of site name, locations, LUT (land use types), and observation period for each site. The observation time range of most sites is from 2010 to 2015. Since the selected

satellite data is from 2013 to 2023, the time period coinciding with the satellite research scope is selected for analysis in this study. Since the observation time of some sites is not within the satellite research period and the number of other types of sites is small and unrepresentative, we selected the following 24 sites.

To make clarification, we have added "The observation periods for most sites range from 2010 to 2015, with detailed site information, including site names, locations, land cover types, and observation periods, provided in Table S1 (*SI*). Given that the satellite data selected for this study spans from 2013 to 2023, the analysis is limited to the period corresponding to the satellite data coverage. For sites where the observation period does not overlap with the satellite research period, and considering the typically low NH3 concentrations at background sites, this study selected 24 representative urban and rural stations for adjustment to improve the reliability of subsequent NH3 dry deposition estimates. The locations of monitoring sites and land cover types across China are also shown in Figure. 1a." on lines 251-259.

**Table S1.** NH3 concentration observation site information, including site names, locations, land cover types, and observation periods.

| Site name                     | Lon    | Lat   | LUT   | Monitoring period   |
|-------------------------------|--------|-------|-------|---------------------|
| China Agricultural University | 116.28 | 40.02 | Urban | Apr. 2010-Dec. 2015 |
| Zhengzhou                     | 113.37 | 34.75 | Urban | Oct. 2010-Dec.2015  |
| Dalian                        | 121.58 | 38.92 | Urban | Sep. 2010-Dec. 2015 |
| Nanjing                       | 118.85 | 31.84 | Urban | Jan. 2015-Dec.2015  |
| Baiyun                        | 113.27 | 23.16 | Urban | May. 2010-Dec.2015  |
| Wenjiang                      | 103.84 | 30.55 | Urban | Oct. 2010-Dec.2015  |
| Shangzhuang                   | 116.20 | 40.11 | Rural | Apr. 2010-Dec. 2015 |
| Quzhou                        | 114.94 | 36.78 | Rural | Apr. 2010-Dec. 2015 |
| Yangqu                        | 112.89 | 38.05 | Rural | Apr. 2010-Dec. 2015 |
| Zhumadian                     | 114.05 | 33.02 | Rural | Apr. 2010-Dec. 2015 |
| Yangling                      | 108.01 | 34.31 | Rural | Apr. 2010-Dec. 2015 |
| Yucheng                       | 116.63 | 36.94 | Rural | Sep. 2012-Dec. 2015 |
| Gongzhuling                   | 124.83 | 43.53 | Rural | Jul. 2010-Dec. 2015 |
| Lishu                         | 124.17 | 43.36 | Rural | Jul. 2010-Dec. 2015 |
| Wuwei                         | 102.60 | 38.07 | Rural | Oct. 2010-Dec.2015  |
| Wuxue                         | 115.79 | 30.01 | Rural | Aug. 2011-Dec.2015  |
| Taojiang                      | 111.97 | 28.61 | Rural | Oct. 2010-Dec.2015  |
| Fengyang                      | 117.56 | 32.88 | Rural | Feb. 2013-Dec.2015  |
| Zhanjiang                     | 110.33 | 21.26 | Rural | Aug. 2010-Dec.2015  |
| Fuzhou                        | 119.36 | 26.17 | Rural | Apr. 2010-Dec.2015  |
| Fenghua                       | 121.53 | 29.61 | Rural | Aug. 2010-Dec.2015  |
| Ziyang                        | 104.63 | 30.13 | Rural | Jul. 2010-Dec.2015  |

| Yanting  | 105.47 | 31.28 | Rural | May. 2011-Dec.2015 |
|----------|--------|-------|-------|--------------------|
| Jiangjin | 106.18 | 29.06 | Rural | Jan. 2013-Dec.2015 |

Where inventories disagree with inferred trends (e.g., fertilizer usage trends vs EDGAR/MEIC vs policy implementation), explicitly discuss the discrepancy. Possible reasons: (1) differences between bottom-up inventories and top-down estimates, (2) regional heterogeneity in fertilizer use, (3) post-2015 changes not captured in inventory updates, (4) changes in emission factors or agricultural practices.

Thanks so much for these suggestions and done as suggested, we added the following two figures in the supplementary file and analyzed them in the text:

We also added more clarification and analysis on lines 805-833 as "We also investigated the temporal changes of agricultural fertilizer application and livestock farming in China from 2013 to 2023, which are treated as the dominating source of NH3 emissions in China (Figures S16-S17, *SI*). During the study period, the application rate of agricultural fertilizers in China showed a trend of first increasing and then decreasing, reaching a peak in 2015, and then continuing to decline until 2023. In order to reveal the changing characteristics of different regions more clearly, we examined the change of agricultural fertilizer amount in each region, and the results indicated that all regions showed a downward trend. At the same time, the total amount of livestock breeding in China first decreased and then rose during the same period.

Furthermore, it is important to note that, although satellite based observations from 2013 to 2023 reveal a clear upward trend in NH3 concentrations at both column-averaged near surface level and ground level, emission inventories from EDGAR, MEIC, and previous bottom-up estimates suggest that NH3 emissions in China have stabilized or declined gradually in recent years (Liao et al., 2022; Zheng et al., 2018). This discrepancy is not only evident in the current study but has also been observed in other research, where some satellite-based NH3 inversion studies show varying degrees of increasing trends (Zhang et al., 2017; Evangeliou et al., 2021; Luo et al., 2022). The difference may stem from the inherent contrasts between "bottom-up" and "top-down" estimation methods. Several top-down studies indicate that the observed rise in NH3 emissions could be partially explained by the neglect of SO2 and NOx column concentration changes. For instance, Luo (2022) estimated global NH3 emissions from 2008 to 2018 using a top-down approach and found that NH3 emissions in eastern China increased by 61% per decade (6.6 Tg a-1 per decade), particularly after 2013, driven primarily by the rise in IASI NH3 column concentrations. However, when the model incorporated the decreasing SO2 and NOx column concentrations, NH3 emissions in eastern China were found to decrease by 19% per decade, with the decline becoming more pronounced after 2013 (28% per decade), aligning more closely with inventory results. This suggests that SO2 and NOx concentrations play a significant role in mitigating atmospheric NH3 levels. Additionally, both SO2 and NOx emissions are negatively correlated with NH3 concentrations to some extent (Deng et al., 2022)."

Figure S16. Changes in agricultural fertilizer quantities in different regions from 2013 to 2023

Figure S17. Changes in total livestock farming, 2013-2023.

Liao W, Liu M, Huang X, et al. Estimation for ammonia emissions at county level in China from 2013 to 2018[J]. Science China Earth Sciences, 2022, 65(6): 1116-1127.

Zheng B, Tong D, Li M, et al. Trends in China's anthropogenic emissions since 2010 as the consequence of clean air actions[J]. Atmospheric Chemistry and Physics, 2018, 18(19): 14095-14111.

Zhang X, Wu Y, Liu X, et al. Ammonia emissions may be substantially underestimated in China[J]. Environmental Science & Technology, 2017, 51(21): 12089-12096.

Evangeliou N, Balkanski Y, Eckhardt S, et al. 10-year satellite-constrained fluxes of ammonia improve performance of chemistry transport models[J]. Atmospheric Chemistry and Physics, 2021, 21(6): 4431-4451.

Luo Z, Zhang Y, Chen W, et al. Estimating global ammonia (NH3) emissions based on IASI observations from 2008 to 2018[J]. Atmospheric Chemistry and Physics Discussions, 2022, 2022: 1-22.

Deng Z. Satellite ammonia (NH3) remote sensing retrieval technology and its application in China, 2022.

The authors should cross-check with additional inventories or top-down emission estimates. Reword the manuscript to avoid implying firm causal attribution unless supported by consistent evidence (inventory trends, policy timing, and observational trends).

Done as recommended, we modified Figure. S13c to include a comparison of different types of emission inventories and top-down estimates, and also added more comparisons as

Figure S15. Time series of annual emissions of (a) SO2, (b) NOx and (c) NH3 over China from 2013 to 2022 obtained from EDGAR emission inventories and different approaches including both bottom-up and top-down results, note: MEIC, EDGAR v8.1, Liao 2020, Li 2025 are bottom-up emissions datasets, Liu 2022, Chen 2023, Wen 2024 are top-down emissions datasets.

There is a significant difference in the findings of the top-down versus bottom-up approach. Studies of bottom-up methods usually show a downward trend in emissions to varying degrees, while top-down methods generally reflect an upward trend. The bottom-up method mainly estimates emissions through the product of the level of socioeconomic activity and the corresponding emission coefficient. Its data mostly come from field surveys or statistical data, but it often fails to fully cover small-scale, scattered anthropogenic emission sources (Zeng et al., 2018). In addition, several studies have shown that emissions from sectors such as industry and transportation are also severely underestimated in traditional bottom-up studies (Van Damme et al, 2018; Chang et al., 2019). In

contrast, satellite observation methods have better continuity and comparability, and can more comprehensively reflect the changing trend of emissions. The study by Evangeliou et al. also shows that the results based on satellite observation inversion can represent global NH3 emissions more accurately than traditional emission inventories. However, in Figure. S13c, the research results of different bottom-up methods are relatively close, while there is a big difference between the research results of top-down methods, with the highest value being about 68% higher than the lowest value. This difference may be related to differences in model settings in different studies and different settings for the effective lifetime of NH3 in the atmosphere. In addition, China's emissions of acid gases such as SO2 and NOx have dropped significantly in the past ten years, reducing the consumption of NH3 in the atmosphere, which may lead to high NH3 emission estimation results based on top-down methods that do not fully consider the impact of acid gases. To sum up, there are large differences in the estimation of NH3 emissions by different methods, so it is necessary to further strengthen the comprehensive analysis and mutual verification of various methods (such as emission factor method, satellite observation inversion method and field observation method) to improve the accuracy and reliability of estimation results (Chen P et al., 2023).

We have added more descriptions on lines 827-850 as: "Furthermore, it is important to note that, although satellite based observations from 2013 to 2023 reveal a clear upward trend in NH3 concentrations at both column-averaged near surface level and ground-level, emission inventories from EDGAR, MEIC, and previous bottom-up estimates suggest that NH3 emissions in China have stabilized or declined gradually in recent years (Liao et al., 2022; Zheng et al., 2018). This discrepancy is not only evident in the current study but has also been observed in other research, where some satellite-based NH3 inversion studies show varying degrees of increasing trends (Zhang et al., 2017; Evangeliou et al., 2021; Luo et al., 2022). The difference may stem from the inherent contrasts between "bottom-up" and "top-down" estimation methods as displayed in Figure 13c. Several top-down studies indicate that the observed rise in NH3 emissions could be partially explained by the neglect of SO2 and NOx column concentration changes. For instance, Luo (2022) estimated global NH3 emissions from 2008 to 2018 using a top-down approach and found that NH3 emissions in eastern China increased by 61% per decade (6.6 Tg a-1 per decade), particularly after 2013, driven primarily by the rise in IASI NH3 column concentrations. However, when the model incorporated the decreasing SO2 and NOx column concentrations, NH3 emissions in eastern China were found to decrease by 19% per decade, with the decline becoming more pronounced after 2013 (28% per decade), aligning more closely with inventory results. This suggests that SO2 and NOx concentrations play a significant role in mitigating atmospheric NH3 levels. Additionally, both SO2 and NOx emissions are negatively correlated with NH3 concentrations to some extent (Deng et al., 2022). In summary, there are large differences in the estimation of NH3 emissions by different methods, so it is necessary to further strengthen the comprehensive analysis and mutual verification of various methods (such as emission factor method, satellite observation inversion method and field observation method) to improve the accuracy and reliability of estimation results (Chen P et al., 2023)."

Li D, Liu H, Duan G. High-Resolution Anthropogenic Emission Inventory for China (2015-2024): Spatiotemporal Changes and Environmental Application[J]. Atmospheric Environment, 2025: 121495.

Liu P, Ding J, Liu L, et al. Estimation of surface ammonia concentrations and emissions in China from the polar-orbiting Infrared Atmospheric Sounding Interferometer and the FY-4A Geostationary Interferometric Infrared Sounder[J]. Atmospheric Chemistry and Physics, 2022, 22(13): 9099-9110.

Wen P, Zhang C, Yang Q, et al. Characterization of spatial and temporal distribution of NH3 concentrations and emissions in China based on IASI observations[J]. China Environmental Science, 2024, 44(06): 3040-3051.

Zeng Y, Tian S, Pan Y. Revealing the sources of atmospheric ammonia: a review[J]. Current Pollution Reports, 2018, 4(3): 189-197.

Van Damme M, Clarisse L, Whitburn S, et al. Industrial and agricultural ammonia point sources exposed[J]. Nature, 2018, 564(7734): 99-103.

Chang Y, Zou Z, Zhang Y, et al. Assessing contributions of agricultural and nonagricultural emissions to atmospheric ammonia in a Chinese megacity[J]. Environmental science & technology, 2019, 53(4): 1822-1833.

5. Quantitative comparison of changes in  $NH_3$ ,  $SO_2$  and  $NO_x$  and their role in deposition

•The manuscript claims the increase in dry deposition flux is driven by NH3 concentration increases arising from declining SO2/NOx. However, Fig. S13 indicates NH3 emissions also decline ~20% over the decade, which contradicts the claim. There is no quantitative comparison of the rates of change of NH3 vs SO2/NOx emissions or concentrations.

Done as recommended, we have added the following table to show the average annual emission reduction, average annual emission reduction rate and relative average annual emission reduction rate of SO2, NOx and NH3 emissions from 2013 to 2023 (taking the EDGAR emission inventory as an example), and added the following analysis content to the main text:

Table S9. Average annual reduction, rates, and relative rates of SO2, NOx, and NH3 during 2013-2023.

|                 | Average annual reduction | Average annual reduction rates | Relative annual reduction rates |
|-----------------|--------------------------|--------------------------------|---------------------------------|
|                 | (Tg yr -1 )   | (%)                            | (%)                             |
| $SO_2$          | 0.76                     | 2.5                            | 20.0                            |
| $NO_x$          | 0.61                     | 2.5                            | 20.1                            |
| NH 3 | 0.25                     | 2.6                            | 21.5                            |

According to the calculation and analysis, we found that the increase of NH3 dry deposition was mainly driven by the increase of NH3 concentration, but during the study period, NH3 emissions showed a downward trend. Considering the neutralization effect of SO2 and NOx acid gases on NH3, we analyzed the changes of the three emissions (Table S8). The data in Table S8 shows that the relative annual reduction rates of the three are similar to the average annual reduction rates, both around 20.5% and 2.5%. However, in terms of the average annual reduction, the reduction scale of SO2 is about 3 times that of NH3, and that of NOx is about 2.4 times that of NH3. Since the reduction of SO2 and NOx is larger, more NH3 is distributed in the free state in the atmosphere. In addition, SO2 and NOx, as acid gases, can react with NH3 in the atmosphere, and they have a synergistic effect in consuming NH3. Therefore, although the relative annual reduction rates of the three are similar, the contribution of acid gas as a whole to emission reduction is more significant. From the perspective of chemical reaction measurement relationship, the equation for the reaction between SO2 and NH3 to generate ammonium sulfate is:  $2SO_2 + 4NH_3 + 2H_2O + O_2 \rightarrow 2 \text{ (NH4) } 2SO_4$ . In this reaction, 1 molecule of  $SO_2$  can consume 2 molecules of  $NH_3$ ; The equation for the reaction between NOx and NH3 to generate ammonium nitrate is: NH3 + HNO3 ⇒NH4NO3. This reaction is a 1:1 measurement relationship and is a reversible reaction. It will re-decompose and release NH3 under higher temperature or lower concentration conditions. With the intensification of global warming, NH4NO3 in the atmosphere will also decompose and release NH3. Therefore, although the emissions of SO2, NOx and NH3 have all decreased by about 20.5%, the massive emission reduction of SO2 and NOx has weakened the consumption capacity of NH3, resulting in a relative surplus of NH3 that should have been neutralized, causing NH3 in the atmosphere. The concentration continues to rise, and the increase of NH3 concentration also promotes the increase of NH3 dry deposition.

To make clarification, we have added corresponding reasons and analysis on lines 884-904 as: "Considering the neutralization effect of SO2 and NOx acid gases on NH3, we analyzed the changes of the three emissions (Table S8). The data in Table S8 shows that the relative annual reduction rates of the three are similar to the average annual reduction rates, both around 20.5% and 2.5%. However, in terms of the average annual reduction, the reduction scale of SO2 is about 3 times that of NH3, and that of NOx is about 2.4 times that of NH3. Since the reduction of SO2 and NOx is larger, more NH3 is distributed in the free state in the atmosphere. In addition, SO2 and NOx, as acid gases, can react with NH3 in the atmosphere, and they have a synergistic effect in consuming NH3. Therefore, although the relative annual reduction rates of the three are similar, the contribution of acid gas as a whole to emission reduction is more significant. From the perspective of chemical reaction measurement relationship, the equation for the reaction between SO2 and NH3 to generate ammonium sulfate is:  $2SO_2 + 4NH_3 + 2H_2O + O_2 \rightarrow 2$ (NH4) 2SO4. In this reaction, 1 molecule of SO2 can consume 2 molecules of NH3; The equation for the reaction between NOx and NH3 to generate ammonium nitrate is: NH3 + HNO3 ⇒NH4NO3. This reaction is a 1: 1 measurement relationship and is a reversible reaction. It will re-decompose and release NH3 under higher temperature or lower concentration conditions. With the intensification of global warming, NH4NO3 in the atmosphere will also decompose and release NH3. Therefore, although the emissions of SO2, NOx and NH3 have all decreased by about 20.5%, the massive emission reduction of SO2 and NOx has weakened the consumption capacity of NH3, resulting in a relative surplus of NH3 that should have been neutralized, causing NH3 in the

atmosphere. The concentration continues to rise, and the increase of NH3 concentration also promotes the increase of NH3 dry deposition."

•Provide table or plots that show trends for emissions and concentrations of NH3, SO2, and NOx over the study period. Quantitatively compare declining speeds for NH3, SO2, and NOx emissions/concentrations. If NH3 emissions themselves decreased, explain how a concurrent increase in observed NH3 concentrations could arise. Show analyses that reconcile emissions and observed concentrations.

Done as suggested, and revisions are made as replied above. NH3 emissions are inconsistent with the observed trend of NH3 concentration. The main reasons are reflected in the following three aspects:

1) From 2013 to 2023, the average annual reduction of  $SO_2$  (0.8 Tg) is about 3 times that of NH3 (0.3 Tg), and that of NOx (0.6 Tg) is about 2.4 times that of NH3. Due to the larger reduction of  $SO_2$  and NOx, more NH3 is distributed in the atmosphere in the free state.

2) As acid gases,  $SO_2$  and  $NO_x$  can react with  $NH_3$  in the atmosphere. They have a synergistic effect in consuming  $NH_3$ , and acid gases as a whole contribute more significantly to emission reduction.

3) From the perspective of chemical reaction measurement relationship, during the reaction between  $SO_2$  and  $NH_3$  to generate ammonium sulfate, 1 molecule of  $SO_2$  can consume 2 molecules of  $NH_3$ ; The reaction between  $NO_x$  and  $NH_3$  to generate ammonium nitrate, although the measurement relationship is 1: 1, is a reversible reaction. It will re-decompose and release  $NH_3$  under higher temperature or lower concentration conditions. With the intensification of global warming, it will also cause more  $NH_4NO_3$  in the atmosphere to decompose and release  $NH_3$ .

Therefore, although the emission of  $NH_3$  is in a downward trend, the massive emission reduction of  $SO_2$  and  $NO_x$  weakens the consumption capacity of  $NH_3$ , resulting in a relative surplus of  $NH_3$  that should have been neutralized, causing the concentration of  $NH_3$  in the atmosphere to continue to rise.

Specific comments

Line 67, Paulot et al. (2014) provides emissions for 2005-2008 but does not compare emissions from India. Please update these numbers using more recent emission estimates:

Cropland emissions → Zhan 2020 (already cited in line 443), Xu, P., Li, G., Zheng, Y. et al. Fertilizer management for global ammonia emission reduction. Nature 626, 792-798 (2024). https://doi.org/10.1038/s41586-024-07020-z

top-down estimates -> Luo, Z., Zhang, Y., Chen, W., Van Damme, M., Coheur, P.-F., and Clarisse, L.: Estimating global ammonia (NH3) emissions based on IASI observations from 2008 to 2018, Atmos. Chem. Phys., 22, 10375-10388, https://doi.org/10.5194/acp-22-10375-2022

**bottom up -> from global inventories such as EDGAR, CEDS**

Done as suggested for all above comments, we have revised this paragraph and updated corresponding numbers as "As the world's largest agricultural country in terms of total crop yield, China is also among the top NH3 emitters globally. In 2018, the global NH3 emissions from rice, wheat and corn fields were  $4.3 \pm 1.0$  Tg N yr-1, of which China's emissions per unit area were as high as 19.7 kg N ha-1 yr-1, which was much higher than that of the United States (9.1 kg N ha-1 yr-1) and India (10.8 kg N ha-1 yr-1) (Zhan et al., 2020; Luo et al., 2022). From global inventories such as EDGAR and CEDS, China's NH3 emissions accounted for 19.8% of the global total in 2013. In 2022, this proportion had declined to about 14.5% (Crippa et al., 2024)." on lines 78-85.

Luo Z, Zhang Y, Chen W, et al. Estimating global ammonia (NH3) emissions based on IASI observations from 2008 to 2018[J]. Atmospheric Chemistry and Physics Discussions, 2022, 2022: 1-22.

Crippa M, Guizzardi D, Pagani F, et al. Insights into the spatial distribution of global, national, and subnational greenhouse gas emissions in the Emissions Database for Global Atmospheric Research (EDGAR v8.0). Earth System Science Data, 2024, 16(6): 2811-2830.

**Line 76-78; 106-107; 581-583, Add appropriate references.**

Done as suggested, we have double checked the lines 76-78; Lines 106-107; Statements on lines 581-583, and added corresponding references.

"Non-agricultural sources—such as wildfire of biomass burning, wastewater treatment, human excreta, and transportation—remain relatively minor (Behera et al., 2013; Zhu et al., 2015; Van Damme et al., 2018; Lutsch et al., 2019). Although the growth rate of both agricultural and non-agricultural emissions in China has slowed in recent years, the absolute emissions continue to rise (Chen J et al., 2023)."

"NH3 emission estimates remain highly uncertain due to outdated activity data, poorly constrained emission factors, and underrepresented sources such as cities (Chang et al., 2021). Compared to most other air pollutants, NH3 exhibits greater variability and uncertainty in different inventories and models, particularly because of its diverse agricultural sources (Beusen et al., 2008; Behera et al., 2013)."

"relative to surrounding rural areas (Santamouris et al., 2013; Chang et al., 2021)"

Santamouris M. Heat-island effect[M]//Energy and climate in the urban built environment. Routledge, 2013: 48-68.

Chang, Y., Gao, Y., Lu, Y., Qiao, L., Kuang, Y., Cheng, K., Wu, Y., Lou, S., Jing, S., Wang, H., and Huang, C.: Discovery of a Potent Source of Gaseous Amines in Urban China, Environmental Science & Technology Letters, 10.1021/acs.estlett.1c00229, 2021.

**Line 119-121; 137-139, unclear —> please rewrite for clarity.**

Done as suggested, we have rewritten this part on line 140-147 as "Satellite observations provide wide spatial coverage and continuous temporal resolution, helping to fill spatial-temporal observation gaps by ground networks. Satellite-derived NH3 retrievals contain approximately 1 independent piece of information driven by peak sensitivity (averaging kernel) in the ABL (~1-3 km) (Shephard et al., 2011; Shephard et al., 2020) that can be represented as profiles with limited vertical resolution or integrated column-averaged values. Therefore, column-averaged satellite retrievals cannot directly replace ground-level (1~1.5 m) concentrations but provide complementary information that helps fill in monitoring gaps."

And rewrite the description on lines 175-179 as "Therefore, accurate estimation of NH3 dry deposition and its driving factors are becoming increasingly critical."

Line 143-149, it is mentioned that "long-term studies remain scarce, and the drivers of spatiotemporal variation in  $NH_3$  concentrations and dry deposition ..." but this does not clearly connect with the previous sentence on high  $NH_3$  concentrations in China. Also, a decade-long study may not fully address the gap in long-term observations.

Done as suggested, we have rewritten this paragraph as: "In China, satellite observations indicate that elevated NH3 concentrations are predominantly observed in the North China Plain, Northeast China, and the Sichuan Basin, whereas lower concentrations are found on the Tibetan Plateau (Liu et al., 2017b). Despite the prominent NH3 pollution identified in several regions of China, there remains a lack of comprehensive long-term studies that examine the spatiotemporal variations of NH3 concentrations and dry deposition. The key drivers behind these variations—impacted by rapid urbanization, land-use changes, climate change, and shifts in fertilizer application practices—have not been sufficiently quantified. While observational studies conducted over a ten-year period cannot fully address the data gap, they offer valuable insights into the medium- and long-term trends in NH3 concentrations and deposition patterns."

Satellite-based atmospheric NH3 concentration section: you mentioned two overpass times and two satellite missions. Were both times and missions used for the entire study period? Were all NH3 data over 73°-136°E and 3°-54°N included, or only those over China?

The Suomi National Polar-orbiting Partnership (SNPP) satellite provides observations from May 2012 to May 2021, with data missing from April to July 2019. The NOAA-20 satellite offers observations from March 2019 to the present. The study period covers 2013 to 2023, and therefore, both SNPP and NOAA-20 data, along with the transit times of the two satellites, were incorporated. NH3 data were extracted from regions within China, defined by the coordinates 73°-136°E and 3°-54°N, thus limiting the dataset to Chinese territories. More details of satellite data have been added as illustrated below.

We have revised this paragraph in section 2.1 as "The CrIS (version 1.6.4) satellite-based atmospheric NH3 concentration used in this study. The CrIS is a hyperspectral infrared sounder onboard the Suomi National Polar-orbiting Partnership (Suomi NPP), NOAA-20, and NOAA-21 satellites (Shephard et al., 2020). Operating in a sun-synchronous orbit at an altitude of approximately 824 km, CrIS provides global coverage twice daily, with local overpass time around 13:30 (daytime) and 01:30 (nighttime). The instrument has a swath width of up to 2200 km, with a nadir spatial resolution of approximately 14 km, and excellent signal-to-noise ratio (Zavyalov et al., 2013). The CrIS fast physical retrieval (CFPR) algorithm (Shephard and Cady-Pereira, 2015) produces NH3 retrievals using CrIS onboard Suomi NPP from May 2012 to May 2021, and CrIS onboard NOAA-20 since March 8, 2019.

In this study, the near surface level of CrIS-derived atmospheric NH3 retrieved profile concentrations was utilized, which are strongly correlated with ABL values around 900 hPa (~1 km) and can represent column average NH3 concentration from ground to ~1 km. To avoid misunderstanding, we define near surface level in this study as the lowest level of CrIS-derived NH3 retrieved profile (average from ground to ~1km), and the ground-level as height of 1~1.5 m, which is the typical height of site-based observations. As this study focuses on China, we used NH3 data over regions of 73°-136°E and 3°-54°N and extracted NH3 concentration within China. To ensure data reliability, only high-quality retrievals were included, filtered using a Quality Flag (QF)  $\geq$  3 and Cloud\_Flag = 0. Non-detects (Cloud\_Flag = 3) that account for values below the detection limit of the sensor were not included in this study (White et al., 2023; Shephard et al., 2025), but are not expected to have a significant impact in source regions found in China. The analysis period spans from 2013 to 2023, covering both the SNPP and NOAA-20 satellite missions, and provides an 11-year, near-continuous time series of atmospheric NH3 observations over China. To assess the consistency between the two satellite missions, a regression analysis was performed using monthly averaged NH3 concentrations from the overlapping period (2019-2021), revealing strong agreement and consistency across China (Figure S1, *SI*). For subsequent analyses, the original satellite retrievals were resampled to a uniform spatial resolution of 0.1° × 0.1°."

**Line 197/572, Is "land use types" the same as "land cover types" mentioned in lines 208/575?**

Thanks so much for pointing out this description, the terms "land use types" (lines 197/572) and 'land cover types' (lines 208/575) in the manuscript were the same. For clarity and consistency, both terms have now been standardized to 'land cover types' in the revised manuscript.

GEOS-Chem simulation: You simulate Vd at  $0.5^{\circ} \times 0.625^{\circ}$  over China. Why not: (1) directly use F from simulations, and (2) analyze at this resolution instead of downscaling to  $0.25^{\circ}$ ?

The reason why using RF model to downscaling  $V_d$  to  $0.25^\circ$  are mainly based on the following two considerations: (1) Within spatial resolution of  $0.5^\circ \times 0.625^\circ$ , it will lead to aggregation error when quantify NH3 concentration and dry deposition from different land cover types; (2) the GEOS-Chem model requires substantial computational resources for one decade, and to further improve spatial resolution and computational efficiency.

We have added more clarification to explain the reasons: "However, considering (1) the spatial resolution of  $0.5^{\circ} \times 0.625^{\circ}$  will lead to aggregation errors when quantifying NH3 concentration and dry deposition from different land cover types within the same grid cell, and (2) the GEOS-Chem model requires substantial computational resources for one decade, and to further improve spatial resolution and computational efficiency (Figure S2, SI), a random forest machine learning algorithm was also applied to simulate dry deposition velocities from 2013 to 2023 based on output from GEOS-Chem model (see more details in Section 2.4), where the spatial resolution can improve to  $0.25^{\circ}$ , see more details in Section 2.4."

**Line 257, Clarify what "two approaches" refers to.**

The two methods mentioned in line 257 are the GEOS-Chem model and the random forest (RF) algorithm. We have modified the description to avoid ambiguity as: "the comparisons of  $V_d$  simulation by using GEOS-Chem and RF model will be evaluated in Section 3.4.1."

Line 305, "soil moisture" is not a meteorological variable -> land-surface/hydrological variable.

Done as suggested, we have made changes to line 305 as "input parameters included five ERA5-derived meteorological and hydrological variables".

**Eq.3 and Eq.4, derived from Eq.2, why is it dlnC/dlnF instead of dlnF/dlnC?**

Here the  $\Delta lnF$  represents changes of NH3 dry deposition, dlnC/dlnF represents changes of NH3 concentration to dry deposition, we have revised for avoiding misunderstanding as "where  $\Delta ln$  denotes the change in the natural logarithm, C and Vd represent contributions from NH3 concentration and dry deposition velocity to dry deposition of F, respectively."

Line 354-357, Higher accuracy is typically associated with higher thermal contrast; conversely, lower thermal contrast would lead to higher uncertainties in  $NH_3$  retrievals.

Done as recommended, we made changes to lines 354-357 as: "Higher accuracy is typically associated with higher thermal contrast; conversely, lower thermal contrast would lead to higher uncertainties in NH3 retrievals,"

Line 369-372, it is mentioned "lowering NH3 emissions from pastoral sources". Please specify the exact sources.

Done as suggested, we have revised this sentence as: "These measures have significantly alleviated the ecological pressure on grasslands and fostered the transformation and upgrading of grassland animal husbandry, as well as environmental optimization. Therefore, with policy support, they contribute to reducing environmental pollution from animal husbandry in grassland areas, thereby lowering NH3 emissions."

Line 465, Explain how the "7 % per year growth rate" was calculated and provide the national average growth rate.

In the study, so we determined the growth rate by calculating the compound annual growth rate. The specific calculation formula is as follows:

$$CAGR = \left(\frac{EV}{BV}\right)^{\frac{l}{n}} - 1$$

where EV is the Ending Value, BV is the Beginning Value, and n is the Number of periods.

The average annual growth rate of Huang-Huai-Hai Plain is 6.0%, and the average annual growth rate of the whole country is 2.0%. We revised this sentence in the original text as: "We used CAGR method to calculate the annual growth rate of NH3 concentration across the country and in the Huang-Huai-Hai Plain region. The Huang-Huai-Hai Plain showed the steepest increase, with an average annual rise of 0.24 ppb, corresponding to a 6.0% per year growth rate—3 times the national average of 2.0% (Manisha et al., 2025)."

Manisha K, Singh I, Chettry V. Investigating and analyzing the causality amid tourism, environment, economy, energy consumption, and carbon emissions using Toda-Yamamoto approach for Himachal Pradesh, India[J]. Environment, Development and Sustainability, 2025, 27(4): 8731-8766.

**Line 557-558, Does this mean Vd is decreasing in these regions? Clarify.**

The  $V_d$  is decreasing in this region, we have rewritten this sentence as: "Unlike the NH3 concentration trends, there is no region in western China displayed a statistically significant increase in dry deposition flux, which was caused by trend of  $V_d$  in this region, emphasizing the spatial decoupling between emission intensity and deposition patterns in less industrialized regions."

Line 645, it is mentioned "the continuous expansion of urban areas from 2013 to 2023", Consider adding a supplementary figure showing this expansion.

Done as suggestion, we added the temporal variations of urban area from 2013 to 2022.

Figure S14. Area change in urban areas from 2013 to 2022 (Units: km²)

Line 700, I don't see any introduction about the "logarithmic differential method" in the main text or SI

The method used here is illustrated from section 2.6.2, we have revised it as: "To further elucidate the drivers of NH3 dry deposition trends, we employed the method (illustrated in Section 2.6.2) to decompose the relative contributions of changes in NH3 concentrations and deposition velocities across different land cover types (Figure 10b; Table S4, SI).".

Line 718-727, Acid rain does not appear directly related to your NH3 concentration results; consider removing or tightening this section.

As recommended, we have deleted lines 720-727.

Line 742-750, Move to Methods section or remove if not part of the main results.

As suggested, we have deleted the part between 742-747.

Line 782-783, if you also used ground obs from this NNDMN, why are there differences?

Here we mainly mean the observation from Xu et al. (2015, NNDMN) were higher than our results. The results from Xu et al. (2015) are based on the observation average of 43 ground stations (including 10 urban stations, 22 rural stations and 11 background stations), while our results can more represent China with spatial coverage of whole China.

To avoid misunderstand, we have revised this sentence as "In contrast, Xu et al. (2015), utilizing averages from 43 ground stations (including 10 urban stations, 22 rural stations and 11 background stations) from the National

Nitrogen Deposition Monitoring Network (NNDMN), reported substantially higher values for China (10.65 ppb and 1.00 g m-2 yr-1) than our study of spatial coverage of whole China. It can be explained by the representation bias due to the predominance of monitoring sites in urban and rural (mostly agriculture dominated) regions characterized by elevated NH3 emissions and underrepresentation of background locations, resulting in overestimation of national averages when averaging these observation sites."

Line 784-785, urban and rural regions?

They are urban and rural (mostly agriculture dominated) regions, we have revised this sentence as replied above.

Line 835, specify what "atmospheric dynamics" based on your conclusions

Done as suggested, we have revised this sentence as "72.6%-81.2% of deposition changes were governed by changes in NH3 concentrations."

Table 1, Suggest removing from the main text -> information is already in Fig. 1.

Done as suggested, we have removed Table 1 from the main text to SI.

Table 2/3 and line 424/454, Add relative annual growth rates (percentage) to check if the increase in summer/Huang-Huai-Hai Plain is still the largest.

Done as suggested, we added relative annual growth rates to Table 2/3. Through the relative annual growth rate, it is found that the growth in summer is still the largest, but the growth in the middle and lower reaches of the Yangtze River plain exceeds that in the Huang-Huai-Hai plain by 80.5%. Therefore, we added a new description to line 418/line 459 of the original text:

The seasonal rates of increase, in descending order, were: summer (0.065 ppb yr-1), autumn (0.050 ppb yr-1), annual (0.045 ppb yr-1), spring (0.039 ppb yr-1), and winter (0.023 ppb yr-1), as well as the relative annual growth rate in summer is the highest.

In terms of relative growth rate (Table 3), the middle and lower reaches of the Yangtze River plain is the highest (80.5%), followed by the Huang-Huai-Hai plain (79.4%). Similar to the Huang-Huai-Hai plain, the middle and lower reaches of the Yangtze River plain is an important granary in China and a key emission reduction area for acid gases such as  $SO_2$  and  $NO_x$  in China.

**Table 2.** Annual and seasonal average NH3 concentrations and their annual mean increment and relative annual growth rates.

| Caasan | NH 3 concentration | Annual growth in NH 3      | Relative growth rates |
|--------|-------------------------------|---------------------------------------|-----------------------|
| Season | (ppb)                         | concentration (ppb yr -1 ) | (%)                   |
| Annual | 2.88                          | 0.045                                 | 22.5                  |
| Spring | 3.28                          | 0.039                                 | 13.8                  |
| Summer | 3.59                          | 0.065                                 | 30.6                  |
| Autumn | 2.63                          | 0.050                                 | 26.4                  |
| Winter | 2.00                          | 0.023                                 | 18.1                  |

**Table 3.** Average NH3 concentration per unit area and annual mean increment and corrected NH3 concentration in the nine major agricultural regions of China from 2013 to 2023.

|                       | NH 3          | Annual growth in NH 3 | Relative      | Corrected NH 3 |  |
|-----------------------|--------------------------|----------------------------------|---------------|---------------------------|--|
| Agricultural zoning   | concentration            | concentration (ppb               | annual growth | concentration (ppb)       |  |
|                       | (ppb) yr -1 ) |                                  | rates (%)     | concentration (ppo)       |  |
| Huang-Huai-Hai Plain  | 5.29                     | 0.24                             | 79.4          | 11.36                     |  |
| Northern arid and     | 2.20                     | 0.00                             | 21.2          | 6.00                      |  |
| semiarid region       | 3.29                     | 0.08                             | 21.3          | 6.93                      |  |
| Qinghai Tibet Plateau | 3.09                     | -0.03                            | 0.9           | 6.48                      |  |
| Loess Plateau         | 2.90                     | 0.14                             | 54.8          | 6.05                      |  |
| Middle-lower Yangtze  | 2.70                     | 0.12                             | 00.5          | 5.62                      |  |
| Plain                 | 2.70                     | 0.13                             | 80.5          | 5.62                      |  |
| Southern China        | 2.01                     | 0.06                             | 42.7          | 4.09                      |  |
| Northeast China Plain | 2.01                     | 0.08                             | 75.1          | 4.09                      |  |
| Sichuan Basin and     | 1.00                     | 0.00                             | 45.1          | 4.02                      |  |
| surrounding regions   | 1.98                     | 0.06                             | 45.1          | 4.02                      |  |
| Yunnan-Guizhou        | 1.75                     | 0.02                             | 21.0          | 2.52                      |  |
| Plateau               | 1.75                     | 0.03                             | 31.9          | 3.52                      |  |

Table 4, Consider combining the left and right parts into a single table—current format is confusing.

Done as suggested, we modified Table 4 as shown in the following table:

**Table 4.** Average NH3 dry deposition per unit area and annual mean increment in the nine major agricultural regions of China from 2013 to 2023.

| A subseditional matrices | Dry deposition of NH 3 | Annual growth of NH 3 |
|--------------------------|-----------------------------------|----------------------------------|
| Agricultural zoning      | (g m -2 )              | dry deposition                   |

|                                       |      | $(g m^{-2} yr^{-1})$ |
|---------------------------------------|------|----------------------|
| Huang-Huai-Hai Plain                  | 1.06 | 0.054                |
| Northern arid and semiarid region     | 0.61 | 0.012                |
| Qinghai Tibet Plateau                 | 0.61 | -0.004               |
| Loess Plateau                         | 0.55 | 0.030                |
| Middle-lower Yangtze Plain            | 0.52 | 0.034                |
| Southern China                        | 0.49 | 0.020                |
| Northeast China Plain                 | 0.39 | 0.018                |
| Sichuan Basin and surrounding regions | 0.38 | 0.014                |
| Yunnan-Guizhou Plateau                | 0.38 | 0.008                |

Table 5, The comparison with global results and by land cover may not be necessary; consider simplifying.

Thanks for your suggestion, this study takes China as the study area, and we want to evaluate whether the observed NH3 concentration and dry deposition are higher than other regions, to make clarification, we have added "To evaluate and contextualize atmospheric NH3 concentrations and dry deposition in China relative to other global regions and different land cover types, we conducted a comprehensive literature review summarized in Table 5."

Fig. 1b, Provide explanation for percentage values

The percentage values represent area proportion of main land cover type(as list above) to total area in corresponding region, we have added in the caption of Figure 1b as ", note the percentage values represent area proportion of main land cover type(as list above) to total area in corresponding region".

Fig. 2a-j, specify which subplots show trends and which show concentrations.

Done as suggested, we have revised the caption of Figure 2 as "Spatial distribution of annual and seasonal averages of column-averaged NH3 concentration from 2013 to 2023, (a) annual averages, (b) average in spring, (c) average in summer, (d) average in autumn, (e) average in winter; and trend of corresponding column-averaged NH3 concentration from 2013 to 2023 for (f) annual averages, (g) average in spring, (h) average in summer, (i) average in autumn, (j) average in winter (Units: ppb for concentration; ppb yr-1 for trend)."

Fig. S8 and S9, subplot order is inconsistent -> please standardize.

Thanks so much for pointing out this typo, done as suggested.

Fig. 7d, Clarify whether this shows interannual variability; define the term in the text and specify it in the subplot y-axis label.

Figure. 7d shows the annual change of total dry deposition of NH3 for different land cover types. According to the suggestion, we have defined the term in the text and specify it in the subplot y-axis label.

---

## Author Response (AR1)

**Reviewer #1:**

*General comments: This manuscript addresses an important topic, spatial-temporal variability of atmospheric NH₃ and its dry deposition across China, which has not been jointly studied before, especially for using CrIS in China, as far as I know. However, the main problem is that the logical connection between derived surface-level/near-surface NH₃ concentrations and the derived NH₃ dry-deposition fluxes is not sufficiently explained. In its current form, it is difficult to follow the storyline between concentrations and dry depositions and contains several conceptual and presentation problems in figures and tables that must be resolved before publication. The title ("one decade of satellite and ground-based observations") is misleading because the text does not make clear which data sources (RF-derived GEOS-Chem simulations, satellite, or ground obs) dominate the results and how they are linked. Suggest alternatives, something like, "Decadal changes in atmospheric ammonia and dry deposition across China inferred from space-ground measurements, and model simulations". The manuscript frequently mixes satellite, ground, reanalysis, and inventory products without a clear, reproducible workflow.*

We sincerely thank the reviewer for the valuable suggestions and critical comments. We have considerably revised this MS based all comments, especially on strengthening the description of logical connection derived surface-level/near-surface NH₃ concentrations and the derived NH₃ dry-deposition fluxes. Besides, all points have been addressed below (review query in black; author response in blue). Changes to the text in the manuscript have been marked in blue.

For the above-mentioned main question of (1) "the logical connection between derived surface-level/near-surface NH₃ concentrations and the derived NH₃ dry-deposition fluxes is not sufficiently explained. In its current form, it is difficult to follow the storyline between concentrations and dry depositions and contains several conceptual and presentation problems in figures and tables that must be resolved before publication.". We have added more clarification and materials to support the reason and method to NH₃ dry deposition calculation, which can be explained that the NH₃ dry deposition should be calculated by multiplying dry deposition velocity and ground-level NH₃ concentration (~1.5 m, the same height of site-based observations). Given that the lowest retrieval layer of satellite-based NH₃ concentrations represents a column average from the ground to approximately 1 km (CrIS), and considering the large vertical gradient within the planetary boundary layer (PBLH), the column-averaged CrIS observations were adjusted to the ground-level.

We have revised the abstract on lines 35-58 as: "This study integrated 2013-2023 satellite-derived NH₃ column concentrations from the Cross-track Infrared Sounder (CrIS) with adjustments from approximately five years ground in-situ ground observations to derive spatial-temporal variation in ground-level NH₃ concentrations across China. We also used the GEOS-Chem transport model and a random forest algorithm by using emission inventories and reanalysis meteorological fields to simulate NH₃ dry deposition velocity and fluxes, and explore the mechanisms driving observed trends. The CrIS observations results show that column-averaged (averages from ground to ~1 km) NH₃ concentrations were the highest in the North China Plain (>10 ppb), with notable annual

and seasonal increasing trends. NH$_3$ concentrations in 2023 were 13.8%-30.6% higher than in 2013. CrIS retrievals aligned well with in-situ data, though were generally about twice as high. After applying the regression equation between ground in-situ observations and CrIS column-averaged NH$_3$ concentrations, we derive the spatial-temporal ground-level (1~1.5 m) NH$_3$ concentrations and dry deposition fluxes from 2013 to 2023. The NH$_3$ dry deposition fluxes exhibited a clear east-west gradient, with maxima in the North China Plain, and another hotpot region is also observed in the Sichuan Basin, southwestern China. Increases in ground-level NH$_3$ concentrations and deposition were most pronounced in urban, cropland, and forest regions, with urban areas experiencing the fastest growth and grasslands the highest total deposition. The national mean ground-level NH$_3$ concentration and dry deposition flux were 4.98 ppb and 0.51 g NH$_3$ m$^{-2}$ yr$^{-1}$, respectively. Anthropogenic emissions explained 77.4% of the variability in ground-level NH$_3$ concentration trend, and meteorological factors accounted for the remainder. Besides, 72.6%-81.2% of the NH$_3$ dry deposition trend was governed by NH$_3$ concentration changes. This study identifies the underlying cause of increasing ammonia pollution, which can be used to better inform nitrogen management strategies in China."

Besides, we added "Satellite observations provide wide spatial coverage and continuous temporal resolution, helping to fill spatial-temporal observation gaps by ground networks. Satellite-derived NH$_3$ retrievals contain approximately 1 independent piece of information driven by peak sensitivity (averaging kernel) in the ABL (~1-3 km) (Shephard et al., 2011; Shephard et al., 2020) that can be represented as profiles with limited vertical resolution or integrated column-averaged values. Therefore, column-averaged satellite retrievals cannot directly replace ground-level (1~1.5 m) concentrations but provide complementary information that helps fill in monitoring gaps. " on lines 140-147.

Added "In addition to these near surface ammonia concentration observations (from either in-situ surface or satellite observations), the dry deposition estimations also depend on deposition velocities (Lei et al., 2021; Liu S et al., 2024). Therefore, an alternative and reliable approach is to combine model simulated dry deposition, ground-level NH$_3$ concentration from sites and satellite-based column-averaged observations, which can make full use of corresponding advantages and eliminate the large uncertainty from emission inventories of different pollution species." on lines 158-164.

Added and revised "In this study, the near surface level of CrIS-derived atmospheric NH$_3$ retrieved profile concentrations was utilized, which are strongly correlated with ABL values around 900 hPa (~1 km) and can represent column average NH$_3$ concentration from ground to ~1 km. To avoid misunderstanding, we define near surface level in this study as the lowest level of CrIS-derived NH$_3$ retrieved profile (average from ground to ~1km), and the ground-level as height of 1~1.5 m, which is the typical height of site-based observations." on lines 210-215.

Added "As noted above, the calculation of NH$_3$ dry deposition flux depends on ground-level NH$_3$ concentrations, although tens of site-based NH$_3$ concentration observations are available, they cannot provide long term

spatial-temporal resolved NH$_3$ distributions especially in regions with high spatial heterogeneity within China. Therefore, we combined the advantage of ground-based NH$_3$ observations of which can represent heights of 1~1.5 m, and satellite based spatial-temporal NH$_3$ distributions. A linear relationship was constructed by comparing both datasets at the same location and period, where the regression equation was used to adjust the lower boundary layer satellite mixing ratio observations to ground-level of 1~1.5 m." on lines 261-268. And more detailed specific explanations are also added throughout this revised version as replied below.

For the question of "and contains several conceptual and presentation problems in figures and tables that must be resolved before publication", all comments regarding corresponding figures and tables have been revised or resolved as replied below in details.

For the comment of "The title ("one decade of satellite and ground-based observations") is misleading because the text does not make clear which data sources (RF-derived GEOS-Chem simulations, satellite, or ground obs) dominate the results and how they are linked. Suggest alternatives, something like, "Decadal changes in atmospheric ammonia and dry deposition across China inferred from space-ground measurements, and model simulations", we have changed the title of revised version as "Decadal changes in atmospheric ammonia and dry deposition across China inferred from space-ground measurements and model simulations" based on this suggestion.

For the comments of "The manuscript frequently mixes satellite, ground, reanalysis, and inventory products without a clear, reproducible workflow.", as replied above, the satellite based NH$_3$ column averages and site-based ground observations are combined to analyze the spatial distribution of NH$_3$ concentration and also used to calculate the dry deposition. Besides, the reanalysis, and inventory products will be used to simulate NH$_3$ concentration and deposition and quantify contributions from different factors as emissions and meteorological fields. To make clarification and avoid misreading of satellite, ground, reanalysis, and inventory products on the workflow, we first have added corresponding explanation between satellite, ground on lines 35-47, 211-216 and 259-266. Such as "This study integrated 2013-2023 satellite-derived NH$_3$ column concentrations from the Cross-track Infrared Sounder (CrIS) with adjustments from approximately five years ground in-situ ground observations to derive spatial-temporal variation in ground-level NH$_3$ concentrations across China. We also used the GEOS-Chem transport model and a random forest algorithm by using emission inventories and reanalysis meteorological fields to simulate NH$_3$ dry deposition velocity and fluxes, and explore the mechanisms driving observed trends. The CrIS observations results show that column-averaged (averages from ground to ~1 km) NH$_3$ concentrations were the highest in the North China Plain (>10 ppb), with notable annual and seasonal increasing trends.". More detailed explanations have also been added and revised throughout this revised version.

*1.Clarify the satellite and ground linkage and what is actually shown in Figs. 3-4*

Done as suggested, "the satellite and ground linkage" in this study is that the CrIS satellite-based NH$_3$ observation

represent column average from ground to ~1 km within atmospheric boundary layer (ABL) and ground site-based observation represents NH$_3$ concentration at around 1~1.5 m. They should display high consistency caused by regional emissions but with different magnitude, caused by the obvious vertical profiles of NH$_3$ within ABL. Besides, because the NH$_3$ dry deposition is calculated by multiplying dry deposition velocity and ground level NH$_3$ concentration (~1.5 m, the same height of site-based observations), the column averaged CrIS observations should be calibrated to ground level. Most of the revisions regarding "Clarify the satellite and ground linkage" have been replied above, we have added "These studies demonstrate the utility of satellite retrievals in characterizing NH$_3$ pollution and its spatiotemporal evolution, especially in regions lacking surface monitoring. In addition to these near surface ammonia concentration observations (from either in-situ surface or satellite observations), the dry deposition estimations also depend on deposition velocities (Lei et al., 2021; Liu S et al., 2024). Therefore, an alternative and reliable approach is to combine model simulated dry deposition, ground-level NH$_3$ concentration from sites and satellite-based column-averaged observations, which can make full use of corresponding advantages and eliminate the large uncertainty from emission inventories of different pollution species." on lines 156-164. And added "In this study, the near surface level of CrIS-derived atmospheric NH$_3$ retrieved profile concentrations was utilized, which are strongly correlated with ABL values around 900 hPa (~1 km) and can represent column average NH$_3$ concentration from ground to ~1 km. To avoid misunderstanding, we define near surface level in this study as the lowest level of CrIS-derived NH$_3$ retrieved profile (average from ground to ~1km), and the ground-level as height of 1~1.5 m, which is the typical height of site-based observations." on lines 210-215.

For the time series and spatial distribution of NH$_3$ concentration in Figures 3 and 4, the Figure 3 displays CrIS satellite-based column-averaged (from ground to 1 km) NH$_3$ concentrations across China; and Figure 4 displays comparisons between CrIS satellite-based column average NH$_3$ concentration and ground site based NH$_3$ observations. To make clarification, we revised the caption of Figure 3 as "Figure 3. (a) Seasonal and (b) regional variations in CrIS satellite-based column-averaged (from ground to 1 km) NH$_3$ concentrations across China from 2013 to 2023 (Unit: ppb)." And revised the caption of Figure 4 as "Figure 4. (a) Comparison between CrIS satellite-based column average (from ground to ~1 km) NH$_3$ concentration and ground site based (~1.5 m) NH$_3$ observations before calibration; (b) comparison between CrIS satellite-based column average NH$_3$ concentration and ground site based NH$_3$ observations after calibration to ground level; (c) Spatial distribution of calibrated satellite-based NH$_3$ concentration and comparisons with ground site based NH$_3$ concentrations in 2015 (Unit: ppb), note the calibration from CrIS satellite-based column average (ground to ~1 km) to ground level (~1.5 m) is conducted by using the linear regression equation derived from panel a, each scatter plot represents monthly averages of all available observations for either urban or rural site."

*The satellite product is described as a "near-surface column average at ~900 m" while ground sites measure at ~1 m. The rationale for using a regression to "correct" or calibrate the satellite is not justified, and Fig. 4b shows that the R2 does not improve after correction. If regression does not raise R2, explain why the regression is still preferred (e.g., reduces bias, corrects seasonal bias, etc.). If the vertical gradient between ~900 m and 1 m*

*is relatively constant, justify why a simple multiplicative (or additive) conversion factor was not used instead of a regression.*

Thanks so much for this comment, as replied above, the reason to use linear regression equation is to calibrate the column averaged $NH_3$ concentration to ground level of 1.5 m, and we can be further use them to derive $NH_3$ dry deposition. Here the reason why $R^2$ did not change is because that the same equation y=0.35+0.16 was used to all scatter plots which theoretically only change the RMSE and have no influence on $R^2$. The decrease of RMSE also indicate that this equation can obviously calibrate the column averaged $NH_3$ concentration to ground level of 1.5 m. Here the conversion is x=(y-0.16)/0.45=2.22x-0.36, where y and x represent CrIS satellite-based column-averaged (from ground to 1 km) and the $NH_3$ value after calibration to 1.5 m, respectively. This approach is also similar with using a simple multiplicative (or additive) conversion factor as mentioned in this comment.

To make clarification, we added "After correction, a new regression (Figure 4b) shows a nearly 1:1 agreement between satellite and ground-based measurements, with the RMSE reduced from 3.56 ppb to 1.69 ppb. The purpose of the linear regression equation is to adjust the column-averaged $NH_3$ concentration to the ground-level at 1.5 m, as described in Section 2.2. This adjustment enables the derivation of $NH_3$ dry deposition, which can then be compared with global observations. The reason that the $R^2$ value remained unchanged is that the same equation, y=0.35+0.16, was applied to all scatter plots. This theoretically affects only the RMSE and does not influence the $R^2$ value. The reduction in RMSE further indicates that this approach effectively adjusts the column-averaged $NH_3$ concentration to the ground-level at 1.5 m. The conversion is given by x=(y-0.16)/0.45=2.22y-0.36 , where y represents the CrIS satellite-based column-averaged $NH_3$ concentration (from ground to 1 km), and x denotes the $NH_3$ concentration after adjustment to 1.5 m. This approach is conceptually similar to using a simple multiplicative (or additive) conversion factor." on lines 608-620.

*Explicitly state what the satellite product represents (column, layer height, vertical averaging kernel). If you intend to present surface-level $NH_3$ , then produce maps and time series of the surface concentration (satellite-derived and corrected by sites) in Sect. 3.1-3.2. If you still keep the near-surface average, explain plainly at the beginning of the results to describe the retrieval layer.*

Done as suggested, the CrIS-derived $NH_3$ vertical profile is divided into 15 levels through inversion, and the observation layer of CrIS-derived $NH_3$ retrieved used in this study was the lowest layer and represent the column average from ground to around 900 hPa (~1km), which is defined as near surface layer and can better reflect the impact of human activities and natural source emissions on the near-Earth atmospheric environment. Satellite-derived $NH_3$ retrievals contain approximately 1 independent piece of information driven by peak sensitivity (averaging kernel) in the boundary layer (~1-3 km) (Shephard et al., 2011; Shephard et al., 2020) that can be represented as profiles with limited vertical resolution or integrated column-averaged values. Therefore, column-averaged satellite retrievals cannot directly replace ground level (1.5 m) concentrations but provide complementary information that helps fill in monitoring gaps. By using the calibration from ground site

observations, the calibrated NH₃ concentration can more represent concentration at ~1.5 m, which has also been defined as ground level to make it different with above near surface level. And the ground level NH₃ concentration will be used to calculate dry deposition in China. Here in Section 3.1-3.2, this study first display the spatial-temporal patterns of the near surface NH₃ concentration in China, which can be directly compared with previous studies using the same lowest layer. And in Section 3.3, we also want to display the comparisons between site-based concentration and near surface NH₃ concentration, which will be further used to calibrate the near surface NH₃ concentration to ground level.

To make clarification, we added "Satellite-derived NH₃ retrievals contain approximately 1 independent piece of information driven by peak sensitivity (averaging kernel) in the ABL (~1-3 km) (Shephard et al., 2011; Shephard et al., 2020) that can be represented as profiles with limited vertical resolution or integrated column-averaged values. Therefore, column-averaged satellite retrievals cannot directly replace ground-level (1~1.5 m) concentrations but provide complementary information that helps fill in monitoring gaps." on lines 141-147. "This discrepancy can be attributed to the vertical gradient of NH₃ in the atmosphere: ground-based sensors typically local point source observations operate at heights of 1-1.5 m, while satellite observations are regional (14 km) with low vertical resolution (~1km or more), which is shown from the averaging kernels (Shephard et al.,2011, Shephard et al., 2020)." on lines 592-595.

And more explanation at the beginning of the results section 3.1 as: "Using CrIS satellite-derived near surface NH₃ concentrations (representing average between ground to ~1 km) from 2013 to 2023, a high-resolution (0.1° × 0.1°) monthly averaged NH₃ concentration dataset across China over an 11-year period was generated. The observation from the near surface layer can reflect the impact of human activities and natural source emissions on the near-Earth atmospheric environment.". Added "In this section, we continue to present the spatiotemporal near-surface NH₃ concentrations derived from CrIS lower ABL mixing ratio values." on lines 505-506. And revised the title of Section 3.1 as "3.1 Spatial patterns of near surface satellite NH₃ concentration and its trend analysis", Section 3.2 as "3.2 Temporal variation of near surface satellite NH₃ concentrations for different regions" and Section 3.3 as "3.3 Comparison between satellite and ground-based NH₃ observations and adjustment from surface level to ground-level NH₃ concentration".

***2.Emission inventories: document, justify choices, and correct low-level mistakes***
Done as suggested, the detailed comments have been addressed and replied below.

***The manuscript references "six inventories", but it is unclear why different inventories were used for SO₂, NOₓ, and NH₃ , and Text S3/Table S2 contains errors (institution names, versions, resolutions).***

Done as suggested, the reason to use different inventories of SO₂, NOx, and NH₃ is based on the reason that many previous studies have concluded large potential bias in using single inventory caused by uncertainties from emission factors and activity data. Therefore, we make full use of all available inventories from different sources

to provide robust trends of emission changes. To make clarification, we added "The reason of using multiple emission inventories instead of only EDGAR is based on the fact that many previous studies have concluded large potential bias in using a single inventory caused by highly uncertain emission factors and activity data discrepancies. Therefore, we make full use of all available inventories from different data sources to provide robust evaluation of their emission changes." on lines 348-352.

Thanks so much for pointing out the typos, and regarding the errors in Text S3/Table S2 (institution name, version, resolution), we also double checked and modified the text and tables.

**Table S2.** Detailed information of 6 different emission inventories.

| Data | Domain | Major institutions | Version | Time period | Resolution | References |
|---|---|---|---|---|---|---|
| CAQIEI | China | IAP | v1.0 | 2013-2020 | 15 km | Kong et al., 2023 |
| MEIC | China | Tsinghua University | v1.4 | 1990-2020 | Provincial | Zheng et al., 2018 |
| ABaCAS | China | Tsinghua University | v2.0 | 2005-2021 | Provincial | Li et al., 2023 |
| CEDS | Global | JGCRI | v_2021_02_05 | 1970-2019 | 0.5° | McDuffie et al., 2020 |
| EDGAR | Global | JRC | v8.1 | 1970-2022 | 0.1° | Crippa et al., 2024 |
| DPEC | China | Tsinghua University | v1.2 | 2020; 2025; 2030; 2035; 2040; 2045; 2050; 2055; 2060 | Provincial | Cheng et al., 2023 |

The descriptions of these inventories have been added and revised in Text S3 (*SI*).

The Inversed Emission Inventory for Chinese Air Quality (CAQIEI), jointly developed by the Institute of Atmospheric Physics, Chinese Academy of Sciences (IAP, CAS), and the China National Environmental Monitoring Center (CNEMC), is a top-down long-term emission inventory for China. It provides emissions data for multiple air pollutants—including $SO_2$ and $NO_x$—from 2013 to 2020, with a horizontal resolution of 15 km. CAQIEI has been shown to effectively reduce biases in prior emission inventories (Kong et al., 2023).

The Multi-resolution Emission Inventory for China (MEIC), developed by Tsinghua University, is a bottom-up

emission inventory model that covers the period from 1990 to 2020. It offers spatially resolved emission data at provincial scale. In this study, provincial-level data from MEIC were utilized to calculate and analyze long-term emission trends of $SO_2$, $NO_x$, and $NH_3$ across China (Zheng et al., 2018).

The Air Benefit and Cost and Attainment Assessment System - Emission Inventory version 2.0 (ABaCAS), co-developed by Tsinghua University, South China University of Technology, and other institutions, is a decision-support system for cost-effectiveness evaluation of air pollution control and attainment planning. The dataset spans from 2005 and has been updated through 2021, with spatial resolutions including provincial scale (Li et al., 2023).

The Community Emissions Data System (CEDS) is a global emission inventory that provides gridded emissions of various gases and aerosol precursors—including $CO_2$, $CH_4$, $NO_x$, and $SO_2$—with spatial resolutions of 0.5°. The data set spans from 1970 to 2019, and most of the data after 1950 are the result of extensive coordination and processing (McDuffie et al., 2020).

The Emissions Database for Global Atmospheric Research (EDGAR), developed by the Joint Research Centre (JRC) of the European Union, provides global gridded emissions at a 0.1° resolution. The temporal resolution is from 1970 to 2022, and this study uses annual grid data to analyze the emission change trend of $SO_2$, $NO_x$, and $NH_3$ (Crippa et al., 2024).

The Dynamic Projection model for Emissions in China (DPEC), developed by Tsinghua University, is a forward-looking model that projects China's future emissions under multiple scenarios. The current version of the DPEC dataset (v1.2) includes five policy scenarios: early peak-net zero-clean air, on-time peak-net zero-clean air, on-time peak-clean air, clean air, and baseline. The spatial resolution is consistent with that of the MEIC inventory (Cheng et al., 2023).

*Add a table listing all inventories used with: name, publisher/institution, version/year, spatial resolution, temporal resolution, main purpose, and how each inventory was used in your study.*

Done as suggested, we have added the following table in *SI* materials, which shows the emission inventory used name, institution, version, spatial resolution, temporal resolution, main purpose, and how each inventory was used in our study.

**Table S5.** Details of 6 different emission inventories and their usage purposes.

| Data | Major institutions | Version | Time period | Resolution | Main purpose |
|---|---|---|---|---|---|
| CAQIEI | IAP, CAS | v1.0 | 2013-2020 | 15 km | The multi-year emission changes |

| | | | | | |
|---|---|---|---|---|---|
| ABaCAS | Tsinghua University | v2.0 | 2005-2021 | Provincial | of $SO_2$ and $NO_x$ are plotted, which is used to analyze the change trend of $SO_2$ and $NO_x$ emissions, and provide data support for the analysis of $NH_3$ concentration change trend |
| CEDS | JGCRI | v_2021_02_05 | 1970-2019 | 0.5° | |
| EDGAR | JRC | v8.1 | 1970-2022 | 0.1° | The multi-year emission changes of $SO_2$, $NO_x$ and $NH_3$ are plotted, which is used to analyze the emission trend of $SO_2$, $NO_x$ and $NH_3$, and provide data support for the analysis of $NH_3$ concentration trend |
| MEIC | Tsinghua University | v1.4 | 1990-2020 | Provincial | |
| DPEC | Tsinghua University | v1.2 | 2020; 2025; 2030; 2035; 2040; 2045; 2050; 2055; 2060 | Provincial | Draw the temporal trends of $SO_2$, $NO_x$ and $NH_3$ from 2020 to 2026, and analyze the future emission trends to provide theoretical support for the possible future trend of $NH_3$ concentration |

***Explain why different inventories were selected for different species. If possible, use a consistent set of inventories for cross-species comparison, or present a justification for why species-specific choices are necessary.***

Done as suggested, as replied above, in order to analyze the influencing factors of $NH_3$ concentration change trend, we adopted a variety of $NH_3$, $NO_x$ and $SO_2$ emission inventories. And the reason to use different inventories of $SO_2$, $NO_x$, and $NH_3$ is based on the reason that many previous studies have concluded large potential bias in using single inventories caused by emission factors and activity data. Therefore, we make full use of all available inventories from different sources to provide robust trends of emission changes. To make clarification, we added "The reason of using multiple emission inventories instead of only EDGAR is based on the fact that many previous studies have concluded large potential bias in using a single inventory caused by highly uncertain emission factors and activity data discrepancies. Therefore, we make full use of all available inventories from different data sources to provide robust evaluation of their emission changes." on lines 348-352. Among them, five inventories, CAQIEI, MEIC, ABaCAS, CEDS and EDGAR, were used for $NO_x$ and $SO_2$ to analyze their emission changes over the years. However, due to the relatively late start of $NH_3$ research, some inventories do not include $NH_3$ emission data. Therefore, we finally selected representative EDGAR and MEIC inventories globally and in China for special analysis of $NH_3$ emissions.

***State whether biomass burning emissions were included in the simulations and, if so, which dataset was used. If biomass burning is excluded, provide justification.***

The EDGAR inventory was used in this study to simulate spatial-temporal patterns of $NH_3$ concentration, where there is not biomass burning in EDGAR. However, we extracted emissions from biomass burning from the MEIC inventory for 2013-2020. The following table shows the total emissions of $SO_2$, $NO_x$, and $NH_3$ during this period, as well as the average annual emissions and their proportions from biomass burning. The proportion of these biomass burning among the three gases is less than 3%.

To make clarification, we also added " Note the emissions from EDGAR will be used in this study to simulate spatial-temporal patterns of $NH_3$ concentration. Note the EDGAR does not include biomass burning. However, we also extracted emissions from biomass burning from the MEIC inventory for 2013-2020, the total emissions of $SO_2$, $NO_x$, and $NH_3$ during this period in China, as well as the average annual emissions and their proportions from biomass burning were displayed in Table S6 (*SI*). And the contribution of biomass burning to these three gases was less than 3%, indicating relatively small influence of biomass burning in simulating $NH_3$ concentrations." on lines 366-373.

**Table S6.** Total emissions of $SO_2$, $NO_x$, and $NH_3$ during this period in China, as well as the average annual emissions and their proportions from biomass burning, note data is from the MEIC emission inventory during 2013-2020.

| Gas | biomass combustion emissions (t yr$^{-1}$) | Total emissions (t yr$^{-1}$) | Proportion |
|-----|--------------------------------------------|-------------------------------|------------|
| $SO_2$ | 27.5 | 14499.5 | 0.19% |
| $NO_x$ | 327.3 | 22845.1 | 1.4% |
| $NH_3$ | 254.3 | 9917.9 | 2.6% |

*3.Random Forest (RF) applications and predictor consistency*

***The RF is used for two distinct purposes: (A) to extend/estimate dry deposition velocity (Vd) across 2013-2023 from 2015 simulations, and (B) to identify key drivers of atmospheric $NH_3$ . The methods (Sect. 2.4.2) only describe the prior usage incompletely, and there are inconsistent predictor sources (ERA5 used for RF, MERRA-2 used for GEOS-Chem). Fig. 10 and its description are confusing (panel a vs b; emissions vs deposition drivers).***

As was mentioned in this comment, the RF will be used for two purpose, with (1) the first purpose of simulating dry deposition velocity ($V_d$) across 2013-2023 from 2015 simulations, which was displayed in Section 2.4.2; and (2) the second purpose is to simulate $NH_3$ concentration and identify key drivers of atmospheric $NH_3$ changes as illustrated in Section 2.6.1. To make the purposes of using RF model clearer, we have added "Overall, the RF

model will be used for two purpose, with (1) the first purpose of simulating dry deposition velocity ($V_d$) across 2013-2023, which is displayed in this Section; and (2) the second purpose is to simulate $NH_3$ concentration and identify key drivers of atmospheric $NH_3$ changes as illustrated in Section 2.6.1" on lines 311-314. And also revised the sentence on lines 285-288 as "a random forest machine learning algorithm was also applied to simulate dry deposition velocities from 2013 to 2023 based on output from GEOS-Chem model (see more details in Section 2.4), where the spatial resolution can improve to 0.25°, see more details in Section 2.4.".

Regarding the comment of inconsistent predictor sources of using ERA5 in RF model and MERRA-2 in the GEOS-Chem model, it's because MERRA-2 is a widely used reanalysis data for GEOS-Chem model and with spatial resolution of only 0.5°×0.625°. However, in our study, considering the higher spatial resolution of emissions and $CH_4$ concentration of 0.1°, we prefer to make full use this finer spatial resolution and we chose ERA5 data with has much higher spatial resolution of 0.25°. To make clarification, we have revised and added "The most widely used approach to derive $V_d$ is by model simulation. Here we first used the GEOS-Chem chemical transport model to simulate spatial-temporal varied $V_d$ across China in 2015, with spatial resolution of 0.5° × 0.625° at hourly scale. However, considering (1) the spatial resolution of 0.5° × 0.625° will lead to aggregation errors when quantifying $NH_3$ concentration and dry deposition from different land cover types within the same grid cell, and (2) the GEOS-Chem model requires substantial computational resources for one decade, and to further improve spatial resolution and computational efficiency (Figure S2, *SI*), a random forest machine learning algorithm was also applied to simulate dry deposition velocities from 2013 to 2023 based on output from GEOS-Chem model (see more details in Section 2.4), where the spatial resolution can improve to 0.25°, see more details in Section 2.4."; and "This approach allowed us to extend the simulation to the full 2013-2023 period, while improving both computational efficiency and spatial resolution from 0.5° × 0.625° to 0.25° × 0.25°." on lines 641-643.

The modifications to Figure 10 and its description are detailed in the response to the next comment.

***Revise Fig. 10 and its caption. Make it explicit what each panel displays. If panels show different metrics (contribution to concentration vs contribution to deposition), label and discuss them separately.***

Done as suggested, we have modified the caption of Figure 10 and added more descriptions of Figure 10.

We also discussed Figure 10a and 10b separately, that for Figure 10a, we added and revised the paragraph as "To quantify the contribution of emissions and meteorological factors to changes in $NH_3$ concentrations, we used a random forest model to simulate $NH_3$ concentration with different sensitivity test by replacing single factor, and the difference between them can be treated as contributions from corresponding factor. Figure 10a shows the adjusted ground-level $NH_3$ concentration in 2022 and the simulation results under three different meteorological and emission scenarios. The simulated concentrations are 3.08 ppb, 3.14 ppb, 3.10 ppb. Both meteorological and emission contributions are calculated from the simulation results. Simulation results from the random forest model showed that anthropogenic emissions were the main driver, accounting for approximately 77.4% of the $NH_3$

concentration changes, while meteorological conditions accounted for the remaining 22.6% (Figure 10a)." on lines 780-789.

For Figure 10b, we added and revised the paragraph on lines 863-874 as "To further elucidate the drivers of $NH_3$ dry deposition trends, we employed the method (illustrated in Section 2.6.2) to decompose the relative contributions of changes in $NH_3$ concentrations and deposition velocities across different land cover types (Figure 10b; Table S4, *SI*). All variables were normalized to facilitate comparison of relative contributions. The results show that the change of $NH_3$ dry deposition is mainly driven by the change of atmospheric $NH_3$ concentration, which accounts for 72.6%-81.2% of the total contribution in China and four land cover types. Among them, the concentration changes in urban area contributed the least (72.6%), and the dry deposition rate change contributed the most (27.4%), likely reflecting the more complex aerodynamic and surface resistance conditions in urban environments. In contrast, forested areas showed the highest concentration-driven contribution (81.2%), consistent with their relatively stable surface characteristics and low anthropogenic disturbance.

Added "To quantify the individual contribution from $SO_2$ and $NO_x$, we also applied the constructed RF model with the method introduced in Section 2.6.1. Taking 2013 as the benchmark, the $SO_2$ and $NO_x$ emissions in 2022 are simulated back to the level of 2013, and the results are normalized to calculate the relative contribution. The results show that the contribution of $SO_2$ is 27.1% and that of $NO_x$ is 72.9%. The contribution of $NO_x$ is significantly higher than that of $SO_2$, which is closely related to the earlier start of $SO_2$ emission reduction. Long-term $SO_2$ emission reduction has changed the composition of acid gases in the atmosphere, causing the relative concentration of $NO_x$ to rise, gradually becoming the main acid gas reacting with $NH_3$ (Liu S et al., 2024).

Considering the neutralization effect of $SO_2$ and $NO_x$ acid gases on $NH_3$, we analyzed the changes of the three emissions (Table S9, *SI*). The data in Table S9 shows that the relative annual reduction rates and total reduction rates of the three are similar, with values around 2.5% and 20.5%. However, in terms of the average annual reduction, the reduction scale of $SO_2$ is about 3 times that of $NH_3$, and that of $NO_x$ is about 2.4 times that of $NH_3$. Since the reduction of $SO_2$ and $NO_x$ is larger, more $NH_3$ is distributed in the free state in the atmosphere. In addition, $SO_2$ and $NO_x$, as acid gases, can react with $NH_3$ in the atmosphere, and they have a synergistic effect in consuming $NH_3$. Therefore, although the relative annual reduction rates of the three are similar, the contribution of acid gas as a whole to emission reduction is more significant." on lines 875-892.

[Figure]

**Figure 10.** (a) NH$_3$ concentrations observed by satellite and simulated by random forest models under different meteorological and emission scenarios in 2022; (b) Relative contribution of NH$_3$ concentration and dry deposition velocity to the dry deposition flux changes. Note: in panel a, the yellow bar represents the satellite observed NH$_3$ concentration in 2022, the purple bar represents the random forest model simulated NH$_3$ concentration, the green bar represents the simulated NH$_3$ concentration using 2013 emissions and 2022 meteorological data, and the red bar represents the simulated NH$_3$ concentration using 2013 meteorological data and 2022 emissions data. And in panel b, the relative contributions of meteorological factors and emissions can be obtained by comparison with the difference in NH$_3$ concentration in the purple bar graph.)

*If SO$_2$ and NO$_x$ are included as predictors, present their individual contributions (don't lump them into "anthropogenic emissions" only).*

Done as suggested, we calculated the relative contributions of SO$_2$ and NO$_x$ separately.

We applied the constructed RF model with the method as in Section 2.6.1 is adopted. Taking 2013 as the benchmark, the SO$_2$ and NO$_x$ emissions in 2022 are simulated back to the level of 2013, and the results are normalized to calculate the relative contribution of the them. The results show that the contribution of SO$_2$ is 27.1% and that of NO$_x$ is 72.9%. The contribution of NO$_x$ is significantly higher than that of SO$_2$, which is closely related to the earlier start of SO$_2$ emission reduction. Long-term SO$_2$ emission reduction has changed the composition of acid gases in the atmosphere, causing the relative concentration of NO$_x$ to rise, gradually becoming the main acid gas reacting with NH$_3$ (Liu S et al., 2024).

To make clarification, we have added "To quantify the individual contribution from SO$_2$ and NO$_x$, we also applied the constructed RF model with the method introduced in Section 2.6.1. Taking 2013 as the benchmark, the SO$_2$ and NO$_x$ emissions in 2022 are simulated back to the level of 2013, and the results are normalized to calculate the relative contribution. The results show that the contribution of SO$_2$ is 27.1% and that of NO$_x$ is 72.9%. The contribution of NO$_x$ is significantly higher than that of SO$_2$, which is closely related to the earlier start of SO$_2$ emission reduction. Long-term SO$_2$ emission reduction has changed the composition of acid gases in the

atmosphere, causing the relative concentration of $NO_x$ to rise, gradually becoming the main acid gas reacting with $NH_3$ (Liu S et al., 2024).

Considering the neutralization effect of $SO_2$ and $NO_x$ acid gases on $NH_3$, we analyzed the changes of the three emissions (Table S9, *SI*). The data in Table S9 shows that the relative annual reduction rates and total reduction rates of the three are similar, with values around 2.5% and 20.5%. However, in terms of the average annual reduction, the reduction scale of $SO_2$ is about 3 times that of $NH_3$, and that of $NO_x$ is about 2.4 times that of $NH_3$. Since the reduction of $SO_2$ and $NO_x$ is larger, more $NH_3$ is distributed in the free state in the atmosphere. In addition, $SO_2$ and $NO_x$, as acid gases, can react with $NH_3$ in the atmosphere, and they have a synergistic effect in consuming $NH_3$. Therefore, although the relative annual reduction rates of the three are similar, the contribution of acid gas as a whole to emission reduction is more significant.

From the perspective of chemical reaction measurement relationship, the equation for the reaction between $SO_2$ and $NH_3$ to generate ammonium sulfate is: $2SO_2 + 4NH_3 + 2H_2O + O_2 \rightarrow 2 (NH_4)_2SO_4$. In this reaction, 1 molecule of $SO_2$ can consume 2 molecules of $NH_3$; The equation for the reaction between $NO_x$ and $NH_3$ to generate ammonium nitrate is: $NH_3 + HNO_3 \rightleftharpoons NH_4NO_3$. This reaction is a 1: 1 measurement relationship and is a reversible reaction. It will re-decompose and release $NH_3$ under higher temperature or lower concentration conditions. With the intensification of global warming, $NH_4NO_3$ in the atmosphere will also decompose and release $NH_3$. Therefore, although the emissions of $SO_2$, $NO_x$ and $NH_3$ have all decreased by about 20.5% from 2013 to 2025, the massive emission reduction of $SO_2$ and $NO_x$ has weakened the consumption capacity of $NH_3$, resulting in a relative surplus of $NH_3$ that should have been neutralized, causing $NH_3$ in the atmosphere. The concentration continues to rise, and the increase of $NH_3$ concentration also promotes the increase of $NH_3$ dry deposition." on lines 875-904.

*For RF validation: show diagnostics (train/test split) and present performance metrics separately for validation. For spatial maps (e.g., Fig. 5b for RF-predicted Vd in 2015) indicate whether values shown include both training and validation pixels; better: show a validation map or a scatter of observed vs predicted Vd for the validation set.*

Done as suggested, we added the comparisons by using train and test dataset, which has been described as "The dataset was randomly split into a training set (60%) and a validation set (40%), the comparisons between using two approaches will be evaluated in Section 3.4.1.". The comparisons are displayed below, the scatter plots are generally around 1:1 line, and the $R^2$ values are 0.93 and 0.83, respectively, indicating that the simulation results of the two models have good agreement. We revised the caption of Figure 5 as "Figure 5. $NH_3$ dry deposition velocity in China in 2015: (a) GEOS-Chem simulation; (b) Random forest simulation (includes validation set and training set); (c) Model difference (Unit: $cm \cdot s^{-1}$)", and also added Figure S12 in *SI* material.

[Figure]

**Figure S12.** Scatter density maps of dry deposition rates simulated by GEOS-Chem model and random forest model (Unit: cm·s$^{-1}$); (a) test set; (b) training set

***4. Trend analysis and how representative the 24 sites + satellite decade are***

***The trend analysis relies on 24 ground sites and 11 years of satellite data. Few ground sites have >10 years of continuous records (as mentioned in the Introduction); this could bias trend estimates.***

Here the ground site NH$_3$ observation data are collected from different monitoring departments, influenced by data availability of ground sites based NH$_3$ concentrations, it's hard to get observation data for more than 10 years for these sites. Therefore, it's the reason why we combined both ground based NH$_3$ observations with satellite based spatial-temporal NH$_3$ distributions. Overall, the trend analysis is mainly influenced by temporal variations of CrIS-derived NH$_3$ concentration, with calibration from ground site based observations to ground level. Most of the reasons have been replied above together with detailed revisions.

We have also added more clarification as "As noted above, the calculation of NH$_3$ dry deposition flux depends on ground-level NH$_3$ concentrations, although tens of site-based NH$_3$ concentration observations are available, they cannot provide long term spatial-temporal resolved NH$_3$ distributions especially in regions with high spatial heterogeneity within China. Therefore, we combined the advantage of ground-based NH$_3$ observations of which can represent heights of 1~1.5 m, and satellite based spatial-temporal NH$_3$ distributions. A linear relationship was constructed by comparing both datasets at the same location and period, where the regression equation was used to adjust the lower boundary layer satellite mixing ratio observations to ground-level of 1~1.5 m." on lines 261-268.

***Provide a clear description of the temporal coverage at each of the 24 sites, or justify the site selection procedure***

Done as suggested, as recommended, we have added detailed information for 24 sites to the *SI* file, as shown in the table below. The following table shows the information of site name, locations, LUT (land use types), and observation period for each site. The observation time range of most sites is from 2010 to 2015. Since the selected

satellite data is from 2013 to 2023, the time period coinciding with the satellite research scope is selected for analysis in this study. Since the observation time of some sites is not within the satellite research period and the number of other types of sites is small and unrepresentative, we selected the following 24 sites.

To make clarification, we have added "The observation periods for most sites range from 2010 to 2015, with detailed site information, including site names, locations, land cover types, and observation periods, provided in Table S1 (*SI*). Given that the satellite data selected for this study spans from 2013 to 2023, the analysis is limited to the period corresponding to the satellite data coverage. For sites where the observation period does not overlap with the satellite research period, and considering the typically low $NH_3$ concentrations at background sites, this study selected 24 representative urban and rural stations for adjustment to improve the reliability of subsequent $NH_3$ dry deposition estimates. The locations of monitoring sites and land cover types across China are also shown in Figure. 1a." on lines 251-259.

**Table S1.** $NH_3$ concentration observation site information, including site names, locations, land cover types, and observation periods.

| Site name | Lon | Lat | LUT | Monitoring period |
|---|---|---|---|---|
| China Agricultural University | 116.28 | 40.02 | Urban | Apr. 2010-Dec. 2015 |
| Zhengzhou | 113.37 | 34.75 | Urban | Oct. 2010-Dec.2015 |
| Dalian | 121.58 | 38.92 | Urban | Sep. 2010-Dec.2015 |
| Nanjing | 118.85 | 31.84 | Urban | Jan. 2015-Dec.2015 |
| Baiyun | 113.27 | 23.16 | Urban | May. 2010-Dec.2015 |
| Wenjiang | 103.84 | 30.55 | Urban | Oct. 2010-Dec.2015 |
| Shangzhuang | 116.20 | 40.11 | Rural | Apr. 2010-Dec. 2015 |
| Quzhou | 114.94 | 36.78 | Rural | Apr. 2010-Dec. 2015 |
| Yangqu | 112.89 | 38.05 | Rural | Apr. 2010-Dec. 2015 |
| Zhumadian | 114.05 | 33.02 | Rural | Apr. 2010-Dec. 2015 |
| Yangling | 108.01 | 34.31 | Rural | Apr. 2010-Dec. 2015 |
| Yucheng | 116.63 | 36.94 | Rural | Sep. 2012-Dec. 2015 |
| Gongzhuling | 124.83 | 43.53 | Rural | Jul. 2010-Dec. 2015 |
| Lishu | 124.17 | 43.36 | Rural | Jul. 2010-Dec. 2015 |
| Wuwei | 102.60 | 38.07 | Rural | Oct. 2010-Dec.2015 |
| Wuxue | 115.79 | 30.01 | Rural | Aug. 2011-Dec.2015 |
| Taojiang | 111.97 | 28.61 | Rural | Oct. 2010-Dec.2015 |
| Fengyang | 117.56 | 32.88 | Rural | Feb. 2013-Dec.2015 |
| Zhanjiang | 110.33 | 21.26 | Rural | Aug. 2010-Dec.2015 |
| Fuzhou | 119.36 | 26.17 | Rural | Apr. 2010-Dec.2015 |
| Fenghua | 121.53 | 29.61 | Rural | Aug. 2010-Dec.2015 |
| Ziyang | 104.63 | 30.13 | Rural | Jul. 2010-Dec.2015 |

| | | | | |
|---|---|---|---|---|
| Yanting | 105.47 | 31.28 | Rural | May. 2011-Dec.2015 |
| Jiangjin | 106.18 | 29.06 | Rural | Jan. 2013-Dec.2015 |

***Where inventories disagree with inferred trends (e.g., fertilizer usage trends vs EDGAR/MEIC vs policy implementation), explicitly discuss the discrepancy. Possible reasons: (1) differences between bottom-up inventories and top-down estimates, (2) regional heterogeneity in fertilizer use, (3) post-2015 changes not captured in inventory updates, (4) changes in emission factors or agricultural practices.***

Thanks so much for these suggestions and done as suggested, we added the following two figures in the supplementary file and analyzed them in the text:

We also added more clarification and analysis on lines 805-833 as "We also investigated the temporal changes of agricultural fertilizer application and livestock farming in China from 2013 to 2023, which are treated as the dominating source of $NH_3$ emissions in China (Figures S16-S17, *SI*). During the study period, the application rate of agricultural fertilizers in China showed a trend of first increasing and then decreasing, reaching a peak in 2015, and then continuing to decline until 2023. In order to reveal the changing characteristics of different regions more clearly, we examined the change of agricultural fertilizer amount in each region, and the results indicated that all regions showed a downward trend. At the same time, the total amount of livestock breeding in China first decreased and then rose during the same period.

Furthermore, it is important to note that, although satellite based observations from 2013 to 2023 reveal a clear upward trend in $NH_3$ concentrations at both column-averaged near surface level and ground level, emission inventories from EDGAR, MEIC, and previous bottom-up estimates suggest that $NH_3$ emissions in China have stabilized or declined gradually in recent years (Liao et al., 2022; Zheng et al., 2018). This discrepancy is not only evident in the current study but has also been observed in other research, where some satellite-based $NH_3$ inversion studies show varying degrees of increasing trends (Zhang et al., 2017; Evangeliou et al., 2021; Luo et al., 2022). The difference may stem from the inherent contrasts between "bottom-up" and "top-down" estimation methods. Several top-down studies indicate that the observed rise in $NH_3$ emissions could be partially explained by the neglect of $SO_2$ and NOx column concentration changes. For instance, Luo (2022) estimated global $NH_3$ emissions from 2008 to 2018 using a top-down approach and found that $NH_3$ emissions in eastern China increased by 61% per decade (6.6 Tg $a^{-1}$ per decade), particularly after 2013, driven primarily by the rise in IASI $NH_3$ column concentrations. However, when the model incorporated the decreasing $SO_2$ and NOx column concentrations, $NH_3$ emissions in eastern China were found to decrease by 19% per decade, with the decline becoming more pronounced after 2013 (28% per decade), aligning more closely with inventory results. This suggests that $SO_2$ and NOx concentrations play a significant role in mitigating atmospheric $NH_3$ levels. Additionally, both $SO_2$ and NOx emissions are negatively correlated with $NH_3$ concentrations to some extent (Deng et al., 2022)."

[Figure]

**Figure S16.** Changes in agricultural fertilizer quantities in different regions from 2013 to 2023

[Figure]

**Figure S17.** Changes in total livestock farming, 2013-2023.

Liao W, Liu M, Huang X, et al. Estimation for ammonia emissions at county level in China from 2013 to 2018[J]. Science China Earth Sciences, 2022, 65(6): 1116-1127.

Zheng B, Tong D, Li M, et al. Trends in China's anthropogenic emissions since 2010 as the consequence of clean air actions[J]. Atmospheric Chemistry and Physics, 2018, 18(19): 14095-14111.

Zhang X, Wu Y, Liu X, et al. Ammonia emissions may be substantially underestimated in China[J]. Environmental Science & Technology, 2017, 51(21): 12089-12096.

Evangeliou N, Balkanski Y, Eckhardt S, et al. 10-year satellite-constrained fluxes of ammonia improve performance of chemistry transport models[J]. Atmospheric Chemistry and Physics, 2021, 21(6): 4431-4451.

Luo Z, Zhang Y, Chen W, et al. Estimating global ammonia (NH₃) emissions based on IASI observations from 2008 to 2018[J]. Atmospheric Chemistry and Physics Discussions, 2022, 2022: 1-22.

Deng Z. Satellite ammonia (NH₃) remote sensing retrieval technology and its application in China, 2022.

***The authors should cross-check with additional inventories or top-down emission estimates. Reword the manuscript to avoid implying firm causal attribution unless supported by consistent evidence (inventory trends, policy timing, and observational trends).***

Done as recommended, we modified Figure. S13c to include a comparison of different types of emission inventories and top-down estimates, and also added more comparisons as

[Figure]

**Figure S15.** Time series of annual emissions of (a) SO₂, (b) NOₓ and (c) NH₃ over China from 2013 to 2022 obtained from EDGAR emission inventories and different approaches including both bottom-up and top-down results, note: MEIC, EDGAR v8.1, Liao 2020, Li 2025 are bottom-up emissions datasets, Liu 2022, Chen 2023, Wen 2024 are top-down emissions datasets.

There is a significant difference in the findings of the top-down versus bottom-up approach. Studies of bottom-up methods usually show a downward trend in emissions to varying degrees, while top-down methods generally reflect an upward trend. The bottom-up method mainly estimates emissions through the product of the level of socioeconomic activity and the corresponding emission coefficient. Its data mostly come from field surveys or statistical data, but it often fails to fully cover small-scale, scattered anthropogenic emission sources (Zeng et al., 2018). In addition, several studies have shown that emissions from sectors such as industry and transportation are also severely underestimated in traditional bottom-up studies (Van Damme et al, 2018; Chang et al., 2019). In

contrast, satellite observation methods have better continuity and comparability, and can more comprehensively reflect the changing trend of emissions. The study by Evangeliou et al. also shows that the results based on satellite observation inversion can represent global $NH_3$ emissions more accurately than traditional emission inventories. However, in Figure. S13c, the research results of different bottom-up methods are relatively close, while there is a big difference between the research results of top-down methods, with the highest value being about 68% higher than the lowest value. This difference may be related to differences in model settings in different studies and different settings for the effective lifetime of $NH_3$ in the atmosphere. In addition, China's emissions of acid gases such as $SO_2$ and $NO_x$ have dropped significantly in the past ten years, reducing the consumption of $NH_3$ in the atmosphere, which may lead to high $NH_3$ emission estimation results based on top-down methods that do not fully consider the impact of acid gases. To sum up, there are large differences in the estimation of $NH_3$ emissions by different methods, so it is necessary to further strengthen the comprehensive analysis and mutual verification of various methods (such as emission factor method, satellite observation inversion method and field observation method) to improve the accuracy and reliability of estimation results (Chen P et al., 2023).

We have added more descriptions on lines 827-850 as: "Furthermore, it is important to note that, although satellite based observations from 2013 to 2023 reveal a clear upward trend in $NH_3$ concentrations at both column-averaged near surface level and ground-level, emission inventories from EDGAR, MEIC, and previous bottom-up estimates suggest that $NH_3$ emissions in China have stabilized or declined gradually in recent years (Liao et al., 2022; Zheng et al., 2018). This discrepancy is not only evident in the current study but has also been observed in other research, where some satellite-based $NH_3$ inversion studies show varying degrees of increasing trends (Zhang et al., 2017; Evangeliou et al., 2021; Luo et al., 2022). The difference may stem from the inherent contrasts between "bottom-up" and "top-down" estimation methods as displayed in Figure 13c. Several top-down studies indicate that the observed rise in $NH_3$ emissions could be partially explained by the neglect of $SO_2$ and NOx column concentration changes. For instance, Luo (2022) estimated global $NH_3$ emissions from 2008 to 2018 using a top-down approach and found that $NH_3$ emissions in eastern China increased by 61% per decade (6.6 Tg $a^{-1}$ per decade), particularly after 2013, driven primarily by the rise in IASI $NH_3$ column concentrations. However, when the model incorporated the decreasing $SO_2$ and NOx column concentrations, $NH_3$ emissions in eastern China were found to decrease by 19% per decade, with the decline becoming more pronounced after 2013 (28% per decade), aligning more closely with inventory results. This suggests that $SO_2$ and NOx concentrations play a significant role in mitigating atmospheric $NH_3$ levels. Additionally, both $SO_2$ and NOx emissions are negatively correlated with $NH_3$ concentrations to some extent (Deng et al., 2022). In summary, there are large differences in the estimation of $NH_3$ emissions by different methods, so it is necessary to further strengthen the comprehensive analysis and mutual verification of various methods (such as emission factor method, satellite observation inversion method and field observation method) to improve the accuracy and reliability of estimation results (Chen P et al., 2023)."

Li D, Liu H, Duan G. High-Resolution Anthropogenic Emission Inventory for China (2015-2024): Spatiotemporal Changes and Environmental Application[J]. Atmospheric Environment, 2025: 121495.

Liu P, Ding J, Liu L, et al. Estimation of surface ammonia concentrations and emissions in China from the polar-orbiting Infrared Atmospheric Sounding Interferometer and the FY-4A Geostationary Interferometric Infrared Sounder[J]. Atmospheric Chemistry and Physics, 2022, 22(13): 9099-9110.

Wen P, Zhang C, Yang Q, et al. Characterization of spatial and temporal distribution of NH3 concentrations and emissions in China based on IASl observations[J]. China Environmental Science, 2024, 44(06): 3040-3051.

Zeng Y, Tian S, Pan Y. Revealing the sources of atmospheric ammonia: a review[J]. Current Pollution Reports, 2018, 4(3): 189-197.

Van Damme M, Clarisse L, Whitburn S, et al. Industrial and agricultural ammonia point sources exposed[J]. Nature, 2018, 564(7734): 99-103.

Chang Y, Zou Z, Zhang Y, et al. Assessing contributions of agricultural and nonagricultural emissions to atmospheric ammonia in a Chinese megacity[J]. Environmental science & technology, 2019, 53(4): 1822-1833.

**5.Quantitative comparison of changes in $NH_3$ , $SO_2$ and $NO_x$ and their role in deposition**

●*The manuscript claims the increase in dry deposition flux is driven by $NH_3$ concentration increases arising from declining $SO_2/NO_x$. However, Fig. S13 indicates $NH_3$ emissions also decline ~20% over the decade, which contradicts the claim. There is no quantitative comparison of the rates of change of $NH_3$ vs $SO_2/NO_x$ emissions or concentrations.*

Done as recommended, we have added the following table to show the average annual emission reduction, average annual emission reduction rate and relative average annual emission reduction rate of $SO_2$, $NO_x$ and $NH_3$ emissions from 2013 to 2023 (taking the EDGAR emission inventory as an example), and added the following analysis content to the main text:

**Table S9.** Average annual reduction, rates, and relative rates of $SO_2$, $NO_x$, and $NH_3$ during 2013-2023.

| | Average annual reduction (Tg yr$^{-1}$) | Average annual reduction rates (%) | Relative annual reduction rates (%) |
|---|---|---|---|
| $SO_2$ | 0.76 | 2.5 | 20.0 |
| $NO_x$ | 0.61 | 2.5 | 20.1 |
| $NH_3$ | 0.25 | 2.6 | 21.5 |

According to the calculation and analysis, we found that the increase of $NH_3$ dry deposition was mainly driven by the increase of $NH_3$ concentration, but during the study period, $NH_3$ emissions showed a downward trend. Considering the neutralization effect of $SO_2$ and $NO_x$ acid gases on $NH_3$, we analyzed the changes of the three emissions (Table S8). The data in Table S8 shows that the relative annual reduction rates of the three are similar to the average annual reduction rates, both around 20.5% and 2.5%. However, in terms of the average annual reduction, the reduction scale of $SO_2$ is about 3 times that of $NH_3$, and that of $NO_x$ is about 2.4 times that of $NH_3$. Since the reduction of $SO_2$ and $NO_x$ is larger, more $NH_3$ is distributed in the free state in the atmosphere. In addition, $SO_2$ and $NO_x$, as acid gases, can react with $NH_3$ in the atmosphere, and they have a synergistic effect in consuming $NH_3$. Therefore, although the relative annual reduction rates of the three are similar, the contribution of acid gas as a whole to emission reduction is more significant. From the perspective of chemical reaction measurement relationship, the equation for the reaction between $SO_2$ and $NH_3$ to generate ammonium sulfate is: $2SO_2 + 4NH_3 + 2H_2O + O_2 \rightarrow 2\ (NH_4)_2SO_4$. In this reaction, 1 molecule of $SO_2$ can consume 2 molecules of $NH_3$; The equation for the reaction between $NO_x$ and $NH_3$ to generate ammonium nitrate is: $NH_3 + HNO_3 \rightleftharpoons NH_4NO_3$. This reaction is a 1: 1 measurement relationship and is a reversible reaction. It will re-decompose and release $NH_3$ under higher temperature or lower concentration conditions. With the intensification of global warming, $NH_4NO_3$ in the atmosphere will also decompose and release $NH_3$. Therefore, although the emissions of $SO_2$, $NO_x$ and $NH_3$ have all decreased by about 20.5%, the massive emission reduction of $SO_2$ and $NO_x$ has weakened the consumption capacity of $NH_3$, resulting in a relative surplus of $NH_3$ that should have been neutralized, causing $NH_3$ in the atmosphere. The concentration continues to rise, and the increase of $NH_3$ concentration also promotes the increase of $NH_3$ dry deposition.

To make clarification, we have added corresponding reasons and analysis on lines 884-904 as: " Considering the neutralization effect of $SO_2$ and $NO_x$ acid gases on $NH_3$, we analyzed the changes of the three emissions (Table S8). The data in Table S8 shows that the relative annual reduction rates of the three are similar to the average annual reduction rates, both around 20.5% and 2.5%. However, in terms of the average annual reduction, the reduction scale of $SO_2$ is about 3 times that of $NH_3$, and that of $NO_x$ is about 2.4 times that of $NH_3$. Since the reduction of $SO_2$ and $NO_x$ is larger, more $NH_3$ is distributed in the free state in the atmosphere. In addition, $SO_2$ and $NO_x$, as acid gases, can react with $NH_3$ in the atmosphere, and they have a synergistic effect in consuming $NH_3$. Therefore, although the relative annual reduction rates of the three are similar, the contribution of acid gas as a whole to emission reduction is more significant. From the perspective of chemical reaction measurement relationship, the equation for the reaction between $SO_2$ and $NH_3$ to generate ammonium sulfate is: $2SO_2 + 4NH_3 + 2H_2O + O_2 \rightarrow 2\ (NH_4)_2SO_4$. In this reaction, 1 molecule of $SO_2$ can consume 2 molecules of $NH_3$; The equation for the reaction between $NO_x$ and $NH_3$ to generate ammonium nitrate is: $NH_3 + HNO_3 \rightleftharpoons NH_4NO_3$. This reaction is a 1: 1 measurement relationship and is a reversible reaction. It will re-decompose and release $NH_3$ under higher temperature or lower concentration conditions. With the intensification of global warming, $NH_4NO_3$ in the atmosphere will also decompose and release $NH_3$. Therefore, although the emissions of $SO_2$, $NO_x$ and $NH_3$ have all decreased by about 20.5%, the massive emission reduction of $SO_2$ and $NO_x$ has weakened the consumption capacity of $NH_3$, resulting in a relative surplus of $NH_3$ that should have been neutralized, causing $NH_3$ in the

atmosphere. The concentration continues to rise, and the increase of $NH_3$ concentration also promotes the increase of $NH_3$ dry deposition."

●*Provide table or plots that show trends for emissions and concentrations of $NH_3$ , $SO_2$, and $NO_x$ over the study period. Quantitatively compare declining speeds for $NH_3$ , $SO_2$, and $NO_x$ emissions/concentrations. If $NH_3$ emissions themselves decreased, explain how a concurrent increase in observed $NH_3$ concentrations could arise. Show analyses that reconcile emissions and observed concentrations.*

Done as suggested, and revisions are made as replied above. $NH_3$ emissions are inconsistent with the observed trend of $NH_3$ concentration. The main reasons are reflected in the following three aspects:

1)From 2013 to 2023, the average annual reduction of $SO_2$ (0.8 Tg) is about 3 times that of $NH_3$ (0.3 Tg), and that of $NO_x$ (0.6 Tg) is about 2.4 times that of $NH_3$. Due to the larger reduction of $SO_2$ and $NO_x$, more $NH_3$ is distributed in the atmosphere in the free state.

2)As acid gases, $SO_2$ and $NO_x$ can react with $NH_3$ in the atmosphere. They have a synergistic effect in consuming $NH_3$, and acid gases as a whole contribute more significantly to emission reduction.

3)From the perspective of chemical reaction measurement relationship, during the reaction between $SO_2$ and $NH_3$ to generate ammonium sulfate, 1 molecule of $SO_2$ can consume 2 molecules of $NH_3$; The reaction between $NO_x$ and $NH_3$ to generate ammonium nitrate, although the measurement relationship is 1: 1, is a reversible reaction. It will re-decompose and release $NH_3$ under higher temperature or lower concentration conditions. With the intensification of global warming, it will also cause more $NH_4NO_3$ in the atmosphere to decompose and release $NH_3$.

Therefore, although the emission of $NH_3$ is in a downward trend, the massive emission reduction of $SO_2$ and $NO_x$ weakens the consumption capacity of $NH_3$, resulting in a relative surplus of $NH_3$ that should have been neutralized, causing the concentration of $NH_3$ in the atmosphere to continue to rise.

*Specific comments*

*Line 67, Paulot et al. (2014) provides emissions for 2005-2008 but does not compare emissions from India. Please update these numbers using more recent emission estimates:*

*Cropland emissions → Zhan 2020 (already cited in line 443), Xu, P., Li, G., Zheng, Y. et al. Fertilizer management for global ammonia emission reduction. Nature 626, 792-798 (2024). https://doi.org/10.1038/s41586-024-07020-z*

*top-down estimates -> Luo, Z., Zhang, Y., Chen, W., Van Damme, M., Coheur, P.-F., and Clarisse, L.: Estimating global ammonia (NH₃) emissions based on IASI observations from 2008 to 2018, Atmos. Chem. Phys., 22, 10375-10388, https://doi.org/10.5194/acp-22-10375-2022*

*bottom up -> from global inventories such as EDGAR, CEDS*

Done as suggested for all above comments, we have revised this paragraph and updated corresponding numbers as "As the world's largest agricultural country in terms of total crop yield, China is also among the top NH₃ emitters globally. In 2018, the global NH₃ emissions from rice, wheat and corn fields were $4.3 \pm 1.0$ Tg N yr$^{-1}$, of which China's emissions per unit area were as high as 19.7 kg N ha$^{-1}$ yr$^{-1}$, which was much higher than that of the United States (9.1 kg N ha$^{-1}$ yr$^{-1}$) and India (10.8 kg N ha$^{-1}$ yr$^{-1}$) (Zhan et al., 2020; Luo et al., 2022). From global inventories such as EDGAR and CEDS, China's NH₃ emissions accounted for 19.8% of the global total in 2013. In 2022, this proportion had declined to about 14.5% (Crippa et al., 2024)." on lines 78-85.

Luo Z, Zhang Y, Chen W, et al. Estimating global ammonia (NH₃) emissions based on IASI observations from 2008 to 2018[J]. Atmospheric Chemistry and Physics Discussions, 2022, 2022: 1-22.

Crippa M, Guizzardi D, Pagani F, et al. Insights into the spatial distribution of global, national, and subnational greenhouse gas emissions in the Emissions Database for Global Atmospheric Research (EDGAR v8.0). Earth System Science Data, 2024, 16(6): 2811-2830.

*Line 76-78; 106-107; 581-583, Add appropriate references.*

Done as suggested, we have double checked the lines 76-78; Lines 106-107; Statements on lines 581-583, and added corresponding references.

"Non-agricultural sources—such as wildfire of biomass burning, wastewater treatment, human excreta, and transportation—remain relatively minor (Behera et al., 2013; Zhu et al., 2015; Van Damme et al., 2018; Lutsch et al., 2019). Although the growth rate of both agricultural and non-agricultural emissions in China has slowed in recent years, the absolute emissions continue to rise (Chen J et al., 2023)."

"NH₃ emission estimates remain highly uncertain due to outdated activity data, poorly constrained emission factors, and underrepresented sources such as cities (Chang et al., 2021). Compared to most other air pollutants, NH₃ exhibits greater variability and uncertainty in different inventories and models, particularly because of its diverse agricultural sources (Beusen et al., 2008; Behera et al., 2013)."

"relative to surrounding rural areas (Santamouris et al., 2013; Chang et al., 2021)"

Santamouris M. Heat-island effect[M]//Energy and climate in the urban built environment. Routledge, 2013: 48-68.

Chang, Y., Gao, Y., Lu, Y., Qiao, L., Kuang, Y., Cheng, K., Wu, Y., Lou, S., Jing, S., Wang, H., and Huang, C.: Discovery of a Potent Source of Gaseous Amines in Urban China, Environmental Science & Technology Letters, 10.1021/acs.estlett.1c00229, 2021.

*Line 119-121; 137-139, unclear —> please rewrite for clarity.*

Done as suggested, we have rewritten this part on line 140-147 as "Satellite observations provide wide spatial coverage and continuous temporal resolution, helping to fill spatial-temporal observation gaps by ground networks. Satellite-derived $NH_3$ retrievals contain approximately 1 independent piece of information driven by peak sensitivity (averaging kernel) in the ABL (~1-3 km) (Shephard et al., 2011; Shephard et al., 2020) that can be represented as profiles with limited vertical resolution or integrated column-averaged values. Therefore, column-averaged satellite retrievals cannot directly replace ground-level (1~1.5 m) concentrations but provide complementary information that helps fill in monitoring gaps."

And rewrite the description on lines 175-179 as "Therefore, accurate estimation of $NH_3$ dry deposition and its driving factors are becoming increasingly critical."

*Line 143-149, it is mentioned that "long-term studies remain scarce, and the drivers of spatiotemporal variation in $NH_3$ concentrations and dry deposition ..." but this does not clearly connect with the previous sentence on high $NH_3$ concentrations in China. Also, a decade-long study may not fully address the gap in long-term observations.*

Done as suggested, we have rewritten this paragraph as: "In China, satellite observations indicate that elevated $NH_3$ concentrations are predominantly observed in the North China Plain, Northeast China, and the Sichuan Basin, whereas lower concentrations are found on the Tibetan Plateau (Liu et al., 2017b). Despite the prominent $NH_3$ pollution identified in several regions of China, there remains a lack of comprehensive long-term studies that examine the spatiotemporal variations of $NH_3$ concentrations and dry deposition. The key drivers behind these variations—impacted by rapid urbanization, land-use changes, climate change, and shifts in fertilizer application practices—have not been sufficiently quantified. While observational studies conducted over a ten-year period cannot fully address the data gap, they offer valuable insights into the medium- and long-term trends in $NH_3$ concentrations and deposition patterns."

*Satellite-based atmospheric $NH_3$ concentration section: you mentioned two overpass times and two satellite missions. Were both times and missions used for the entire study period? Were all $NH_3$ data over 73°-136°E and 3°-54°N included, or only those over China?*

The Suomi National Polar-orbiting Partnership (SNPP) satellite provides observations from May 2012 to May 2021, with data missing from April to July 2019. The NOAA-20 satellite offers observations from March 2019 to the present. The study period covers 2013 to 2023, and therefore, both SNPP and NOAA-20 data, along with the transit times of the two satellites, were incorporated. $NH_3$ data were extracted from regions within China, defined by the coordinates 73°-136°E and 3°-54°N, thus limiting the dataset to Chinese territories. More details of satellite data have been added as illustrated below.

We have revised this paragraph in section 2.1 as "The CrIS (version 1.6.4) satellite-based atmospheric $NH_3$ concentration used in this study. The CrIS is a hyperspectral infrared sounder onboard the Suomi National Polar-orbiting Partnership (Suomi NPP), NOAA-20, and NOAA-21 satellites (Shephard et al., 2020). Operating in a sun-synchronous orbit at an altitude of approximately 824 km, CrIS provides global coverage twice daily, with local overpass time around 13:30 (daytime) and 01:30 (nighttime). The instrument has a swath width of up to 2200 km, with a nadir spatial resolution of approximately 14 km, and excellent signal-to-noise ratio (Zavyalov et al., 2013). The CrIS fast physical retrieval (CFPR) algorithm (Shephard and Cady-Pereira, 2015) produces $NH_3$ retrievals using CrIS onboard Suomi NPP from May 2012 to May 2021, and CrIS onboard NOAA-20 since March 8, 2019.

In this study, the near surface level of CrIS-derived atmospheric $NH_3$ retrieved profile concentrations was utilized, which are strongly correlated with ABL values around 900 hPa (~1 km) and can represent column average $NH_3$ concentration from ground to ~1 km. To avoid misunderstanding, we define near surface level in this study as the lowest level of CrIS-derived $NH_3$ retrieved profile (average from ground to ~1km), and the ground-level as height of 1~1.5 m, which is the typical height of site-based observations. As this study focuses on China, we used $NH_3$ data over regions of 73°-136°E and 3°-54°N and extracted $NH_3$ concentration within China. To ensure data reliability, only high-quality retrievals were included, filtered using a Quality Flag (QF) $\geq 3$ and Cloud_Flag = 0. Non-detects (Cloud_Flag = 3) that account for values below the detection limit of the sensor were not included in this study (White et al., 2023; Shephard et al., 2025), but are not expected to have a significant impact in source regions found in China. The analysis period spans from 2013 to 2023, covering both the SNPP and NOAA-20 satellite missions, and provides an 11-year, near-continuous time series of atmospheric $NH_3$ observations over China. To assess the consistency between the two satellite missions, a regression analysis was performed using monthly averaged $NH_3$ concentrations from the overlapping period (2019-2021), revealing strong agreement and consistency across China (Figure S1, *SI*). For subsequent analyses, the original satellite retrievals were resampled to a uniform spatial resolution of 0.1° × 0.1°."

***Line 197/572, Is "land use types" the same as "land cover types" mentioned in lines 208/575?***

Thanks so much for pointing out this description, the terms "land use types" (lines 197/572) and 'land cover types' (lines 208/575) in the manuscript were the same. For clarity and consistency, both terms have now been standardized to 'land cover types' in the revised manuscript.

***GEOS-Chem simulation: You simulate Vd at 0.5° × 0.625° over China. Why not: (1) directly use F from simulations, and (2) analyze at this resolution instead of downscaling to 0.25°?***

The reason why using RF model to downscaling $V_d$ to 0.25° are mainly based on the following two considerations: (1) Within spatial resolution of 0.5° × 0.625°, it will lead to aggregation error when quantify $NH_3$ concentration and dry deposition from different land cover types; (2) the GEOS-Chem model requires substantial computational resources for one decade, and to further improve spatial resolution and computational efficiency.

We have added more clarification to explain the reasons: "However, considering (1) the spatial resolution of 0.5° × 0.625° will lead to aggregation errors when quantifying $NH_3$ concentration and dry deposition from different land cover types within the same grid cell, and (2) the GEOS-Chem model requires substantial computational resources for one decade, and to further improve spatial resolution and computational efficiency (Figure S2, *SI*), a random forest machine learning algorithm was also applied to simulate dry deposition velocities from 2013 to 2023 based on output from GEOS-Chem model (see more details in Section 2.4), where the spatial resolution can improve to 0.25°, see more details in Section 2.4."

***Line 257, Clarify what "two approaches" refers to.***

The two methods mentioned in line 257 are the GEOS-Chem model and the random forest (RF) algorithm. We have modified the description to avoid ambiguity as: "the comparisons of $V_d$ simulation by using GEOS-Chem and RF model will be evaluated in Section 3.4.1. "

***Line 305, "soil moisture" is not a meteorological variable -> land-surface/hydrological variable.***

Done as suggested, we have made changes to line 305 as "input parameters included five ERA5-derived meteorological and hydrological variables".

***Eq.3 and Eq.4, derived from Eq.2, why is it dlnC/dlnF instead of dlnF/dlnC?***

Here the *ΔlnF* represents changes of $NH_3$ dry deposition, *dlnC/dlnF* represents changes of $NH_3$ concentration to dry deposition, we have revised for avoiding misunderstanding as "where *Δln* denotes the change in the natural logarithm, C and $V_d$ represent contributions from $NH_3$ concentration and dry deposition velocity to dry deposition of F, respectively."

***Line 354-357, Higher accuracy is typically associated with higher thermal contrast; conversely, lower thermal contrast would lead to higher uncertainties in $NH_3$ retrievals.***

Done as recommended, we made changes to lines 354-357 as: "Higher accuracy is typically associated with higher thermal contrast; conversely, lower thermal contrast would lead to higher uncertainties in $NH_3$ retrievals,"

***Line 369-372, it is mentioned "lowering NH₃ emissions from pastoral sources". Please specify the exact sources.***

Done as suggested, we have revised this sentence as: "These measures have significantly alleviated the ecological pressure on grasslands and fostered the transformation and upgrading of grassland animal husbandry, as well as environmental optimization. Therefore, with policy support, they contribute to reducing environmental pollution from animal husbandry in grassland areas, thereby lowering NH₃ emissions."

***Line 465, Explain how the "7 % per year growth rate" was calculated and provide the national average growth rate.***

In the study, so we determined the growth rate by calculating the compound annual growth rate. The specific calculation formula is as follows:

$$CAGR = \left( \frac{EV}{BV} \right)^{\frac{1}{n}} - 1$$

where $EV$ is the Ending Value, $BV$ is the Beginning Value, and $n$ is the Number of periods.

The average annual growth rate of Huang-Huai-Hai Plain is 6.0%, and the average annual growth rate of the whole country is 2.0%. We revised this sentence in the original text as: "We used CAGR method to calculate the annual growth rate of NH₃ concentration across the country and in the Huang-Huai-Hai Plain region. The Huang-Huai-Hai Plain showed the steepest increase, with an average annual rise of 0.24 ppb, corresponding to a 6.0% per year growth rate—3 times the national average of 2.0% (Manisha et al., 2025)."

Manisha K, Singh I, Chettry V. Investigating and analyzing the causality amid tourism, environment, economy, energy consumption, and carbon emissions using Toda-Yamamoto approach for Himachal Pradesh, India[J]. Environment, Development and Sustainability, 2025, 27(4): 8731-8766.

***Line 557-558, Does this mean Vd is decreasing in these regions? Clarify.***

The V_d is decreasing in this region, we have rewritten this sentence as: "Unlike the NH₃ concentration trends, there is no region in western China displayed a statistically significant increase in dry deposition flux, which was caused by trend of V_d in this region, emphasizing the spatial decoupling between emission intensity and deposition patterns in less industrialized regions."

***Line 645, it is mentioned "the continuous expansion of urban areas from 2013 to 2023", Consider adding a supplementary figure showing this expansion.***

Done as suggestion, we added the temporal variations of urban area from 2013 to 2022.

[Figure]

**Figure S14.** Area change in urban areas from 2013 to 2022 (Units: km²)

*Line 700, I don't see any introduction about the "logarithmic differential method" in the main text or SI*

The method used here is illustrated from section 2.6.2, we have revised it as: "To further elucidate the drivers of NH₃ dry deposition trends, we employed the method (illustrated in Section 2.6.2) to decompose the relative contributions of changes in NH₃ concentrations and deposition velocities across different land cover types (Figure 10b; Table S4, *SI*).".

*Line 718-727, Acid rain does not appear directly related to your NH₃ concentration results; consider removing or tightening this section.*

As recommended, we have deleted lines 720-727.

*Line 742-750, Move to Methods section or remove if not part of the main results.*

As suggested, we have deleted the part between 742-747.

*Line 782-783, if you also used ground obs from this NNDMN, why are there differences?*

Here we mainly mean the observation from Xu et al. (2015, NNDMN) were higher than our results. The results from Xu et al. (2015) are based on the observation average of 43 ground stations (including 10 urban stations, 22 rural stations and 11 background stations), while our results can more represent China with spatial coverage of whole China.

To avoid misunderstand, we have revised this sentence as "In contrast, Xu et al. (2015), utilizing averages from 43 ground stations (including 10 urban stations, 22 rural stations and 11 background stations) from the National

Nitrogen Deposition Monitoring Network (NNDMN), reported substantially higher values for China (10.65 ppb and 1.00 g m$^{-2}$ yr$^{-1}$) than our study of spatial coverage of whole China. It can be explained by the representation bias due to the predominance of monitoring sites in urban and rural (mostly agriculture dominated) regions characterized by elevated NH$_3$ emissions and underrepresentation of background locations, resulting in overestimation of national averages when averaging these observation sites."

*Line 784-785, urban and rural regions?*

They are urban and rural (mostly agriculture dominated) regions, we have revised this sentence as replied above.

*Line 835, specify what "atmospheric dynamics" based on your conclusions*

Done as suggested, we have revised this sentence as "72.6%-81.2% of deposition changes were governed by changes in NH$_3$ concentrations."

*Table 1, Suggest removing from the main text -> information is already in Fig. 1.*

Done as suggested, we have removed Table 1 from the main text to *SI*.

*Table 2/3 and line 424/454, Add relative annual growth rates (percentage) to check if the increase in summer/Huang-Huai-Hai Plain is still the largest.*

Done as suggested, we added relative annual growth rates to Table 2/3.Through the relative annual growth rate, it is found that the growth in summer is still the largest, but the growth in the middle and lower reaches of the Yangtze River plain exceeds that in the Huang-Huai-Hai plain by 80.5%. Therefore, we added a new description to line 418/line 459 of the original text:

The seasonal rates of increase, in descending order, were: summer (0.065 ppb yr$^{-1}$), autumn (0.050 ppb yr$^{-1}$), annual (0.045 ppb yr$^{-1}$), spring (0.039 ppb yr$^{-1}$), and winter (0.023 ppb yr$^{-1}$), as well as the relative annual growth rate in summer is the highest.

In terms of relative growth rate (Table 3), the middle and lower reaches of the Yangtze River plain is the highest (80.5%), followed by the Huang-Huai-Hai plain (79.4%). Similar to the Huang-Huai-Hai plain, the middle and lower reaches of the Yangtze River plain is an important granary in China and a key emission reduction area for acid gases such as SO$_2$ and NO$_x$ in China.

**Table 2.** Annual and seasonal average NH$_3$ concentrations and their annual mean increment and relative annual growth rates.

| Season | NH$_3$ concentration (ppb) | Annual growth in NH$_3$ concentration (ppb yr$^{-1}$) | Relative growth rates (%) |
|---|---|---|---|
| Annual | 2.88 | 0.045 | 22.5 |
| Spring | 3.28 | 0.039 | 13.8 |
| Summer | 3.59 | 0.065 | 30.6 |
| Autumn | 2.63 | 0.050 | 26.4 |
| Winter | 2.00 | 0.023 | 18.1 |

**Table 3.** Average NH$_3$ concentration per unit area and annual mean increment and corrected NH$_3$ concentration in the nine major agricultural regions of China from 2013 to 2023.

| Agricultural zoning | NH$_3$ concentration (ppb) | Annual growth in NH$_3$ concentration (ppb yr$^{-1}$) | Relative annual growth rates (%) | Corrected NH$_3$ concentration (ppb) |
|---|---|---|---|---|
| Huang-Huai-Hai Plain | 5.29 | 0.24 | 79.4 | 11.36 |
| Northern arid and semiarid region | 3.29 | 0.08 | 21.3 | 6.93 |
| Qinghai Tibet Plateau | 3.09 | -0.03 | 0.9 | 6.48 |
| Loess Plateau | 2.90 | 0.14 | 54.8 | 6.05 |
| Middle-lower Yangtze Plain | 2.70 | 0.13 | 80.5 | 5.62 |
| Southern China | 2.01 | 0.06 | 42.7 | 4.09 |
| Northeast China Plain | 2.01 | 0.08 | 75.1 | 4.09 |
| Sichuan Basin and surrounding regions | 1.98 | 0.06 | 45.1 | 4.02 |
| Yunnan-Guizhou Plateau | 1.75 | 0.03 | 31.9 | 3.52 |

*Table 4, Consider combining the left and right parts into a single table—current format is confusing.*

Done as suggested, we modified Table 4 as shown in the following table:

**Table 4.** Average NH$_3$ dry deposition per unit area and annual mean increment in the nine major agricultural regions of China from 2013 to 2023.

| Agricultural zoning | Dry deposition of NH$_3$ (g m$^{-2}$) | Annual growth of NH$_3$ dry deposition |
|---|---|---|

|  |  | (g m$^{-2}$ yr$^{-1}$) |
| --- | --- | --- |
| Huang-Huai-Hai Plain | 1.06 | 0.054 |
| Northern arid and semiarid region | 0.61 | 0.012 |
| Qinghai Tibet Plateau | 0.61 | -0.004 |
| Loess Plateau | 0.55 | 0.030 |
| Middle-lower Yangtze Plain | 0.52 | 0.034 |
| Southern China | 0.49 | 0.020 |
| Northeast China Plain | 0.39 | 0.018 |
| Sichuan Basin and surrounding regions | 0.38 | 0.014 |
| Yunnan-Guizhou Plateau | 0.38 | 0.008 |

**Table 5, The comparison with global results and by land cover may not be necessary; consider simplifying.**

Thanks for your suggestion. this study takes China as the study area, and we want to evaluate whether the observed NH$_3$ concentration and dry deposition are higher than other regions, to make clarification, we have added "To evaluate and contextualize atmospheric NH$_3$ concentrations and dry deposition in China relative to other global regions and different land cover types, we conducted a comprehensive literature review summarized in Table 5."

**Fig. 1b, Provide explanation for percentage values**

The percentage values represent area proportion of main land cover type(as list above) to total area in corresponding region, we have added in the caption of Figure 1b as ", note the percentage values represent area proportion of main land cover type(as list above) to total area in corresponding region".

**Fig. 2a-j, specify which subplots show trends and which show concentrations.**

Done as suggested, we have revised the caption of Figure 2 as "Spatial distribution of annual and seasonal averages of column-averaged NH$_3$ concentration from 2013 to 2023, (a) annual averages, (b) average in spring, (c) average in summer, (d) average in autumn, (e) average in winter; and trend of corresponding column-averaged NH$_3$ concentration from 2013 to 2023 for (f) annual averages, (g) average in spring, (h) average in summer, (i) average in autumn, (j) average in winter (Units: ppb for concentration; ppb yr$^{-1}$ for trend). "

**Fig. S8 and S9, subplot order is inconsistent -> please standardize.**

Thanks so much for pointing out this typo, done as suggested.

***Fig. 7d, Clarify whether this shows interannual variability; define the term in the text and specify it in the subplot y-axis label.***

Figure. 7d shows the annual change of total dry deposition of NH₃ for different land cover types. According to the suggestion, we have defined the term in the text and specify it in the subplot y-axis label.

[Figure]

***Reviewer #2:***

***The manuscript attempts to investigate spatial-temporal variations of atmospheric NH₃ concentration and its dry deposition across China in 2013-20123 by combining satellite-based, ground-level observational data in publica domain and 3-D modeling results. The analysis sounds scientific, but it needs a substantial revision on potential uncertainty and modeling accuracy. The major comments are listed as below:***

Thanks so much for these detailed suggestions. All points have been addressed below (review query in black; author response in blue). Changes to the text in the manuscript have been marked in blue.

***1) Key scientific questions are too general to be valuable by considering the uncertainties associated and previous studies published in the literature. The authors are encouraged to deeply think the issues.***

Thanks so much and done as suggested. To address the importance of science question regarding quantify one decade's spatial-temporal patterns of NH₃ concentration, dry deposition and driving factors, we have considerably added and revised the introduction section, which highlights recent knowledge gap in China shoes the importance of addressing above science questions. Please see changes from line 75 to 196.

***2) Figure 4, the size of data is too small by considering one decade observations, what happens?***

Thank you for your question. Figures 4a and 4b present the results of the correlation analysis between satellite-based and ground-based NH$_3$ concentrations at monthly averages, each scatter plot represents averages of all available observations for either urban or rural site, and some sites only have 1-2 years with the same overlap period of satellite. Figure 4c further compares the calibrated satellite NH$_3$ concentrations with the corresponding site-based monitoring concentration. Due to discrepancy in the observation institutions and monitoring periods across stations, with some stations providing valid data for only a single year, we adopted a monthly-scale analysis approach to ensure an adequate sample size given the limited data. This method involves extracting monthly average values from ground-based observations and corresponding satellite concentration, thereby decreasing the number of data points to 12 per year and significantly enhancing data reliability of the analysis.

We have revised caption of Figure 4 as: "Figure 4. (a) Comparison between CrIS satellite-based column average (from ground to ~1 km) NH$_3$ concentration and ground site based (~1.5 m) NH$_3$ observations before calibration; (b) comparison between CrIS satellite-based column average NH$_3$ concentration and ground site based NH$_3$ observations after calibration to ground level ; (c) Spatial distribution of calibrated satellite at with ground site based NH$_3$ concentrations in 2015 (Unit: ppb), note the calibration from CrIS satellite-based column average (ground to ~1 km) to ground level (~1.5 m) is by using the linear regression equation derived from panel a, each scatter plot represents monthly averages of all available observations for either urban or rural site."

*Modeling results always suffer from the errors. However, most of air quality modeling results in China well reproduce PM2.5 in approximately 1/3 days in each year. However, it is not case in other times because of the poor prediction of one or several meteorological conditions. The authors should select the 1/3 days with good prediction performance for machine learning.*

We agree that Modeling results always suffer from the errors and it's better to choose days with good prediction. Here the NH$_3$ observation were measured by passive sampling representing averages of one week. And to avoid the random errors from observations and simulations, monthly average was conducted for NH$_3$ concentration for machine learning. To make clarification, we have added more description as "Note previous modeling results (i.e. PM$_{2.5}$) always suffers from bias in 1/3 of modeling days and it's better to choose days with good predictions. And in this study for NH$_3$ observations, they were measured by passive sampler, representing averages of one week instead of hourly or daily scales. Therefore, to avoid the random errors from observations and simulations, monthly average was conducted for NH$_3$ concentration for machine learning." on lines 390-394.

*4) Line 399-400, "Elevated temperatures further enhance volatilization from manure and urban waste, intensifying atmospheric NH$_3$ levels.". The reviewer has much concern on the statement, i.e., the authors might not know what exactly happen for agriculture emissions of NH$_3$ in China? With a large population moving from the country land to the city in the last decade, the sources are negligible.*

Done as suggested, we have added more clarification in this part as "Elevated temperatures further enhance volatilization from manure of agricultural area and urban waste in cities, intensifying atmospheric NH$_3$ concentration. Although urbanization has increased over the past decade, many system-scale farms continue to be used for agricultural production." on lines 487-490.

***Minor comments:***

***1)The effective number through the manuscript are total off and needs to be corrected. Principally, it should be consistent with the analytic error, i.e., 5-10% analytic errors correspond two effective numbers.***

Don as recommended, we have revised with the two effective numbers regarding errors throughout this MS.

***2)Abstract, lines 37, "our results", what does it means? Modeling results? Observations from CrIS, AMoN-China or NNDMN?***

Here "Our results" in line 37 means observation from CrIS, we have revised it as "The CrIS observations results show that column-averaged (averages from ground to ~1 km) NH$_3$ concentrations were the highest in the North China Plain (>10 ppb), with notable annual and seasonal increasing trends"

***3)Abstract, line 40-42, "Dry deposition fluxes exhibited a clear east-west gradient, with maxima in the North China Plain and Sichuan Basin. " The sentence is problematic. Sichuan Basin should be located in southwestern China, correct?***

Done as suggested, we have revised this sentence as "The NH$_3$ dry deposition fluxes exhibited a clear east-west gradient, with maxima in the North China Plain, and another hotpot region is also observed in the Sichuan Basin, southwestern China."

---

## Author Response (AR2)

The authors have comprehensively addressed most of my previous comments through a revised title and an enhanced discussion of the relationship between $NH_3$ concentration, deposition, and their emission contributions. To further strengthen the manuscript, the following minor revisions are required:

We sincerely appreciate and thank the reviewer 1 for the second round of valuable suggestions and comments. We have revised this MS based all comments, more details are replied below.

1. Line 143 and 561: explain the terms "ABL" and "CAGR method" is

Done as suggested, the term "ABL" is atmospheric boundary layer which has been explained on line 99 "the atmospheric boundary layer (hereafter ABL)", where it first appeared; and the "CAGR method" is "compound annual growth rate". To make clarification, we have added corresponding explanation on line 561 as "We used compound annual growth rate (CAGR) method".

2. Line 266-268: Is there a reference for this linear regression approach?

Done as suggested, we added two references for the linear regression approach on line 264 as "(Hu et al., 2017; Liu et al., 2024)".

Liu C, Huang J, Hu C. et al. Sensitivity of surface downward longwave radiation to aerosol optical depth over the Lake Taihu region, China, Atmospheric Research, 305 (2024) 107444

Hu, C, Wang Y, Wang W, et al. Trends in evaporation of a large subtropical lake. Theoretical and Applied Climatology, 2017, 129, 159–170, https://doi.org/10.1007/s00704-016-1768-z.

3. Line 348-350: add references

Done as suggested, we have added two references and citations as of "Crippa et al., 2019; Liu et al., 2024" on line 349.

Crippa, M., Janssens-Maenhout, G., Guizzardi, D., Van Dingenen, R., and Dentener, F.: Contribution and uncertainty of sectorial and regional emissions to regional and global $PM_{2.5}$ health impacts, Atmos. Chem. Phys., 19, 5165–5186, https://doi.org/10.5194/acp-19-5165-2019, 2019.

Liu, H., Hu, C., Xiao, Q., Zhang, J., Sun, F., Shi, X., Chen, X., Yang, Y., and Xiao, W.: Analysis of anthropogenic $CO_2$ emission uncertainty and influencing factors at city scale in Yangtze River Delta region: One of the world's largest emission hotspots, Atmospheric Pollution Research, 15, 102281, https://doi.org/10.1016/j.apr.2024.102281, 2024

4. Line 614: The statement "y = 0.35 + 0.16 was applied to all scatter plots" is unclear. If you applied a multiplicative scaling (i.e., $y_i = a \times y\_pred$), both $R^2$ and RMSE would change, as the variance of residuals relative to y changes nonlinearly. If you instead applied an additive shift (bias correction), $R^2$ might remain almost the same (since $R^2$ is based on variance, not the mean) or slightly increase. Could you clarify which correction was applied? In addition, I believe what you are doing is a form of "K-theory" (gradient diffusion theory), based on the well-mixed assumption in the ABL. It assumes that transport flux can be represented analogously to molecular diffusion, where fluxes are

proportional to the mean gradient of the transported quantity.

Thanks so much for this suggestion, yes, the approach we used here is applying an additive shift (bias correction), where $R^2$ remain almost the same, as mentioned in this comment. It's also based on "K-theory" (gradient diffusion theory) with the well-mixed assumption in the ABL. It assumes that transport flux can be represented analogously to molecular diffusion, where fluxes are proportional to the mean gradient of the transported quantity.

To make clarification, we also added more explanation on lines 611-616 as "The approach we used is applying an additive shift (bias correction), where $R^2$ remain almost the same (Figures 4a-b ), it's also based on "K-theory" (gradient diffusion theory) with the well-mixed assumption in the ABL. This method assumes that transport flux can be represented analogously to molecular diffusion, where fluxes are proportional to the mean gradient of the transported quantity.".

5. Line 789: Should the contribution results correspond to fig. 10b?

Thanks so much and we have double checked the results on line 789, the contribution results correspond to figure 10a, and we feel sorry of lacking corresponding descriptions. To avoid misunderstanding, we have added more explanation of how the relative contributions were calculated on lines 817-830 as "The above relative contributions are calculated by the method that using the emissions and meteorological-hydrological factors from 2013 as the baseline (more details in Method Section 2.6.1), we first simulated the $NH_3$ concentration for 2022 using the emissions and meteorological-hydrological factors from that year. The simulated concentration was 3.08 ppb, which was consistent with the satellite-observed concentration for 2022, yielding a relative error of 0.1%. Subsequently, we replaced the emissions data with those from 2013 while keeping the 2022 meteorological-hydrological factors constant, resulting in a simulated concentration of 3.14 ppb. We then replaced the meteorological-hydrological factors with those from 2013 while keeping the 2022 emissions constant, leading to a simulated concentration of 3.10 ppb. By subtracting the two simulated concentrations from the 2022 $NH_3$ concentration simulation, we quantified the effects of changes in emissions and meteorological-hydrological factors on $NH_3$ concentration. Finally, the results were normalized, revealing that the relative contributions of emissions and meteorological-hydrological factors to the concentration changes were 77.4% and 22.6%, respectively. ".

6. Fig. 4: adjust the y-label or include it in the title of plot (a) and (b) to clarify their differences.

Done as suggested, we have revised the y-label of figure 4b with "Corrected satellite $NH_3$ concentration (ppb)".

[Figure]

7. Table S9: Specify whether the values represent annual emission reductions.

Yes, the values in Tables S9 represent annual emission reductions, to make clarification, we have revised the label of Table S9 as "Table S9. Average annual reduction (Tg yr$^{-1}$), annual reduction rates (% yr$^{-1}$), and relative rates between 2013 and 2023 (%) of $SO_2$, $NO_x$, and $NH_3$."

8. Figure S2: The figure does not display properly (appears black).

Thanks so much for pointing it out, it may be caused by transformation from word version to pdf version, we have replaced this figure in new version.

Referee #2

The manuscript attempts to explore decade variations atmospheric ammonia (NH₃) in terms of methodological integration, spatiotemporal resolution, and data-processing framework. It represents an important contribution to understanding NH3 pollution and dry deposition dynamics across China. Therefore, the manuscript can be accepted after a minor revision.

However, to further enhance the paper's impact and reliability, the authors should strengthen and deepen several aspects, particularly regarding uncertainty quantification and potential changes in NH3 sinks.

We sincerely appreciate and thank reviewer 2 for the second round of valuable suggestions and comments. We have revised this MS based all comments, especially added more explanation and discussions on the uncertainty quantification and potential changes in NH₃ dry deposition and sinks, more details are replied below.

1.Although the authors empirically converted CrIS "near-surface" concentrations (0–1 km) to "ground-level" concentrations (1–1.5 m), they did not sufficiently validate potential systematic biases under varying thermal contrast and boundary-layer conditions, especially in high-altitude or low-temperature regions. Based on reading of this review, vertical profiles of atmospheric NH3 likely changed little in the height below 300 m in today's China. This linear extrapolation neglects the strong vertical gradient of NH₃, which could lead to overestimation in high-concentration regions and underestimation in low-concentration regions. The authors should quantify how such bias may affect the overall uncertainty of their results.

Thanks so much for pointing it out, we agree that there may be spatial changes of relationship between "near-surface" concentrations (0–1 km) and "ground-level" concentrations (1–1.5 m), where the variations of scatter plots in Figure 4a-b also reflects such uncertainty. We further calculated the uncertainty extent of regression slope, which were 0.45±0.04, indicating the above mentioned potential systematic biases under varying thermal contrast and boundary-layer conditions can be 9%, calculated by dividing uncertainty extent of 0.04 with 0.45. These uncertainties can be decreased by conducting more ground site based NH₃ observations in different regions.

Furthermore, we have added more clarifications and discussions on lines 624-633 as "It is important to acknowledge that spatial-temporal uncertainties or potential systematic biases may exist in the relationship between ground-based and satellite-derived NH₃ observations across different regions and under varying thermal contrast and boundary-layer conditions. As demonstrated in the scatter plots of Figure 4a-b, which exhibit significant variability. Nevertheless, the regression slope and associated uncertainty were 0.45 ± 0.04, indicating that the potential systematic biases mentioned above could result in an error of approximately 9% when deriving ground-level NH₃ concentrations and dry deposition rates. This error was calculated by dividing the uncertainty extent (0.04) by the regression slope (0.45). These uncertainties can be mitigated by increasing the number of ground-based NH₃ observations in diverse regions in future studies."

2.The manuscript mentions the combined use of the GEOS-Chem and random forest (RF) models but does not clearly describe the RF model's input variables or the feature-importance ranking. These details should be added to ensure methodological transparency and reproducibility.

Done as suggested, we have added the figure of the feature-importance ranking below and

discussions in Supplementary file, and more details have also been added as "The feature-importance ranking figure illustrates the relative importance of eight driving factors in predicting NH₃ concentrations using the Random Forest model (Figure S15, *SI*). Among the emission and meteorological-hydrological factors, the latter plays a more prominent role in explaining the spatial and temporal variability of NH₃ concentrations. Within the meteorological-hydrological factors, the 10-meter wind speed (20.3%), 2-meter temperature (14.9%), and boundary layer height (13.1%) are the most influential variables affecting the NH₃ concentration simulation. These variables collectively reflect the role of atmospheric diffusion capacity and volatilization conditions in regulating the distribution of NH₃ concentrations. Total precipitation (11.0%) and surface soil moisture content (13.6%) contribute to the removal of NH₃ from the atmosphere, though their relative importance is lower. Among the emission factors, NH₃ emissions (16.4%) are the most significant, followed by NOx (11.0%) and SO₂ (5.1%) emissions. This suggests that, in addition to direct emissions, precursor chemical processes also have an indirect influence on the distribution of NH₃ concentrations. ".

[Figure]

**Figure S15.** Feature-importance ranking figure illustrates the relative importance of eight driving factors in predicting NH₃ concentrations using the Random Forest model.

3.The conclusion states that "77.4% of the trend was explained by anthropogenic sources," yet the structure of variables, characteristics of the training set, and the attribution approach are not clearly explained. The change in NH3 sinks might not support the statement, based on a simple comparison between the decrease in particulate NH4+ and corresponding increase in atmospheric NH3 in US as well as the decrease in particulate NH4+in China.

Done as suggested, we have provided more details and discussions regarding how these values were calculated and potential uncertainty on lines 799-812 as "The above relative contributions are calculated by the method that using the emissions and meteorological-hydrological factors from 2013 as the baseline, we first simulated the NH₃ concentration for 2022 using the emissions and meteorological-hydrological factors from that year. The simulated concentration was 3.08 ppb, which was consistent with the satellite-observed concentration for 2022, yielding a relative error of 0.1%. Subsequently, we replaced the emissions data with those from 2013 while keeping the 2022 meteorological-hydrological factors constant, resulting in a simulated concentration of 3.14 ppb.

We then replaced the meteorological-hydrological factors with those from 2013 while keeping the 2022 emissions constant, leading to a simulated concentration of 3.10 ppb. By subtracting the two simulated concentrations from the 2022 $NH_3$ concentration simulation, we quantified the effects of changes in emissions and meteorological-hydrological factors on $NH_3$ concentration. Finally, the results were normalized, revealing that the relative contributions of emissions and meteorological-hydrological factors to the concentration changes were 77.4% and 22.6%, respectively." .

4.The language could benefit from additional professional editing or AI-assisted polishing to improve fluency and conciseness, particularly by simplifying lengthy or complex sentences.
Done as suggested, three co-authors are native speaker of English and also editor for many journals, they have provided professional editing during the draft written and revision process.

5. The effective number are total off through the manuscript, and should be corrected.
Done as suggested, we have double checked through this MS, and the relative changes of $NH_3$ concentration values are displayed with one decimal place, concentration values are displayed with two decimal places, and the trend analysis are displayed with three decimal places considering these values are relatively small.

6. Lines 485-487 "In China, summer fertilization for maize cultivation—often involving both mineral and organic fertilizers—contributes to the observed summer peak (Paulot et al., 2014)." This is not true in today's China, on basis of the reviewer's summer traveling experience therein.
Thanks so much for pointing it out, the main agricultural crops in China contains maize and rice paddy, corn and wheat, which is applied with fertilized in summer. We have revised this sentence as: "In China, summer fertilization is applied for the key agricultural crops as rice paddy, maize, corn and wheat—often involving both mineral and organic fertilizers—contributes to the observed summer peak (Paulot et al., 2014; Luo et al., 2025)"